# LINK PREDICTION WITH RELATIONAL HYPERGRAPHS

## ABSTRACT

Link prediction with knowledge graphs has been thoroughly studied in graph machine learning, leading to a rich landscape of graph neural network architectures with successful applications. Nonetheless, it remains challenging to transfer the success of these architectures to inductive link prediction with *relational hypergraphs*, where the task is over *k-ary relations*, substantially harder than link prediction on knowledge graphs with binary relations only. In this paper, we propose a framework for link prediction with relational hypergraphs, empowering applications of graph neural networks on *fully relational* structures. Theoretically, we conduct a thorough analysis of the expressive power of the resulting model architectures via corresponding relational Weisfeiler-Leman algorithms and also via logical expressiveness. Empirically, we validate the power of the proposed model architectures on various relational hypergraph benchmarks. The resulting model architectures substantially outperform every baseline for inductive link prediction, and also lead to state-of-the-art results for transductive link prediction.

## 1 INTRODUCTION

Knowledge graphs consist of facts (or, edges) representing different relations between *pairs of nodes*. Knowledge graphs are inherently incomplete (Ji et al., 2020; Wang et al., 2017) which motivated a large literature on link prediction with knowledge graphs (Wang et al., 2014; Schlichtkrull et al., 2018; Sun et al., 2019; Teru et al., 2020; Vashishth et al., 2020; Liu et al., 2021a; Zhu et al., 2021). This task amounts to predicting missing facts in the knowledge graph and has led to a rich landscape of graph neural network architectures (Schlichtkrull et al., 2018; Teru et al., 2020; Vashishth et al., 2020; Zhu et al., 2021). Our understanding of these architectures is supported by theoretical studies quantifying their expressive power (Barceló et al., 2022; Zhang et al., 2021; Huang et al., 2023; Qiu et al., 2024).

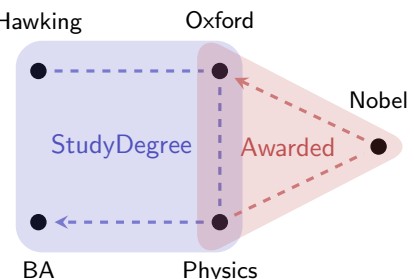

Figure 1: A relational hypergraph over the relations StudyDegree and Awarded. The facts StudyDegree(Hawking, Oxford, Physics, BA) and Awarded(Physics, Nobel, Oxford) are *ordered* hyperedges, where the order of entities in each fact is denoted by dashed arrows.

In this work, we are interested in link prediction on *fully relational* data, where every relation is between $k$ nodes, for any *relation-specific* choice of $k$. Relational data can encode rich relationships between entities; e.g., consider a relationship between *four* entities: "Hawking went to Oxford to study Physics and received a BA degree". This can be represented with a fact StudyDegree(Hawking, Oxford, Physics, BA). Clearly, relational data can be represented via relational hypergraphs, where each *ordered, relational hyperedge* in the hypergraph corresponds to a relational fact (see Figure 1).

**Motivation.** Given the prevalence of relational data, link prediction with relational hypergraphs has been widely studied in the context of shallow embedding models (Wen et al., 2016; Abboud et al., 2020; Fatemi et al., 2020; 2023), where the idea is to generalize knowledge graph embedding methods to relational hypergraphs. The key limitation of these approaches is that they are all *transductive*: they cannot be directly used to make predictions

between nodes that are not seen during training. The same limitation has motivated the development of graph neural network architectures for *inductive* link prediction on knowledge graphs — enabling for predictions between nodes that are not seen during training (Teru et al., 2020) — which eventually led to very strong architectures such as NBFNets (Zhu et al., 2021).

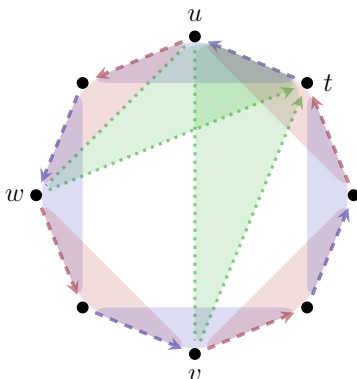

Figure 2: Unary encoders cannot distinguish the query facts $r(u, v, t)$ and $r(u, w, t)$, drawn in green.

In the same spirit, graph neural networks have been extended for inductive link prediction on relational hypergraphs (Yadati, 2020; Zhou et al., 2023), but these approaches do not enjoy the same level of success. This can be attributed to multiple, related factors. In essence, link prediction with relational hypergraphs is a $k$-ary task (for $k$ varying depending on the relation), which is much more challenging than a binary prediction task and requires dedicated approaches. On the other hand, existing proposals are simple extensions of relational graph neural networks (such as RGCNs (Schlichtkrull et al., 2018)), which cannot adequately capture $k$-ary tasks. In fact, these architectures are *unary* encoders that are used for $k$-ary predictions, which is known to be a fundamental limitation (Zhang et al., 2021; Huang et al., 2023). To make these points concrete, let us consider the example shown in Figure 2. In this example, regardless of the choice of the *unary* encoder, it is not possible to distinguish between the query facts $r(u, w, t)$ and $r(u, v, t)$, because the nodes $w$ and $v$ are isomorphic in the hypergraph. However, an appropriate *ternary* encoder can easily differentiate these facts using the information that the distance between the nodes $u$ and $w$ differs from the distance between the nodes $u$ and $v$. These limitations along with a lack of an established theory motivates our study.

**Approach.** We first investigate the expressive power of existing graph neural networks proposed for relational hypergraphs — such as G-MPNNs (Yadati, 2020) and RD-MPNNs (Zhou et al., 2023) — to rigorously identify their limitations. This is achieved by studying the framework of *hypergraph relational message passing neural networks* (HR-MPNNs) which subsumes these architectures. To address the limitations of HR-MPNNs, we introduce *hypergraph conditional message passing neural networks* (HC-MPNNs) as a framework for inductive link prediction inspired by the conditional message passing paradigm studied for knowledge graphs (Zhu et al., 2021; Huang et al., 2023). We conduct a systematic expressiveness study showing that HC-MPNNs can compute richer properties of nodes — dependent on $k$ other nodes — when compared to HR-MPNNs. Specifically, our study for expressive power answers the following questions:

1. *Which nodes can be distinguished by an architecture?* To answer this question, we generalize existing results given for graph neural networks on knowledge graphs (Barceló et al., 2022; Huang et al., 2023) using Weisfeler-Leman algorithms designated for relational hypergraphs.
2. *What properties of nodes can be uniformly expressed by an architecture?* To answer this question, we investigate logical expressiveness which situates the class of node properties that can be expressed by an architecture within an appropriate logical fragment.

**Contributions.** Our main contributions can be summarized as follows:

- We rigorously identify the expressive power and limitations of HR-MPNNs that encompass most of the existing architectures for link prediction with relational hypergraphs (Section 4).
- We introduce the novel framework of HC-MPNNs, which includes more expressive architectures, such as HCNets, and addresses the core limitations of HR-MPNNs (Section 5).
- We present a detailed empirical analysis to validate our theoretical findings (Section 6). Experiments for inductive and transductive link prediction with relational hypergraphs show that a simple HC-MPNNs architecture surpasses all existing baselines leading to state-of-the-art results. Our ablation studies on different model components justify the importance of our model design choices. We supplement the real-world experiments with a synthetic experiment inspired by the example from Figure 2 to validate the expressive power of HC-MPNNs (Appendix M).

All proofs and additional experiments (including link prediction results on standard knowledge graphs) can be found in the appendix of this paper.

## 2    RELATED WORK

**Knowledge graphs.** Link prediction with knowledge graphs has been studied extensively in the literature. Early literature is dominated by knowledge graph embedding models including TransE (Bordes et al., 2013), RotatE (Sun et al., 2019), ComplEx (Trouillon et al., 2016), TuckER (Balazevic et al., 2019), and BoxE (Abboud et al., 2020), which are all restricted to the *transductive* regime. In the space of graph neural networks, RGCN (Schlichtkrull et al., 2018) and CompGCN (Vashishth et al., 2020) emerged as architectures extending standard message passing neural networks (Gilmer et al., 2017) to knowledge graphs using a relational message passing scheme. GraIL (Teru et al., 2020) is the first architecture explicitly designed to operate in the *inductive* regime, but it suffers from a high computational complexity. Zhu et al. (2021) proposed NBFNets as an architecture that subsumes previous methods such as NeuralLP (Yang et al., 2017) and DRUM (Sadeghian et al., 2019). NBFNets perform strongly and have better computational complexity thanks to their high parallelizability Zhu et al. (2021). Recently, A*Net (Zhu et al., 2023) is proposed to scale NBFNets further with the usage of a neural priority function. NBFNets are shown to fall under the framework of *conditional message passing neural networks* (C-MPNNs) (Huang et al., 2023), as they compute node representations "conditioned" pairwise on other node representations, making these architectures suitable for binary link prediction tasks and explaining their superior performance. The success of conditional message passing on knowledge graphs serves as a motivation for our work on relational hypergraphs.

**Relational hypergraphs.** Link prediction with relational hypergraphs has been widely studied in the context of shallow embedding models (Wen et al., 2016; Abboud et al., 2020; Fatemi et al., 2020; 2023). To score facts of the form $r(u_1, \cdots, u_k)$, some methods extend the scoring function (i.e., decoder) of existing knowledge graph embedding methods to consider multiple entities. For example, m-TransH (Wen et al., 2016) is an extension of TransH (Wang et al., 2014) designed to handle multiple entities jointly. Similarly, GETD (Liu et al., 2020) builds on the bilinear embedding method TuckER (Balazevic et al., 2019) to handle higher-arity relations. Fatemi et al. (2020) proposed HSimplE and HypE that disentangle the position and relation embedding. BoxE (Abboud et al., 2020) is an embedding model that encodes each relation using box embeddings, and naturally applies to $k$-ary relations (using $k$ boxes) while achieving strong results on transductive benchmarks. Fatemi et al. (2023) explores the connection between relational algebra and relational hypergraph embeddings and proposes ReAlE. In the space of graph neural networks, Feng et al. (2018) and Yadati et al. (2019) leverage message-passing methods on *undirected* hypergraphs. The first approach that is tailored to relational hypergraphs is G-MPNN (Yadati, 2020), which operates by relational message passing. RD-MPNNs (Zhou et al., 2023) builds on this approach and additionally incorporates the positional information of entities in their respective relations during message passing, which is critical for relational facts since the order of nodes in each edge clearly matters. G-MPNN and RD-MPNNs represent closest related works to the present study and we show that these architectures are instances of HR-MPNNs and hence are subject to the same limitations.

**Hyper-relational knowledge graphs and beyond.** We carefully distinguish between link prediction with relational hypergraphs with hyper-relational knowledge graphs (Galkin et al., 2020), which are knowledge graphs where each edge is additionally augmented with additional information: a multiset of "qualifier-value" pairs, and $n$-ary relational graphs (Guan et al., 2019) relying on *unordered* hypergraphs. We focus on link prediction with relational hypergraphs in this work but note that in practice, we can convert one form of hypergraphs to another with certain transformations. See detailed discussion of in Appendix B.

## 3    LINK PREDICTION WITH RELATIONAL HYPERGRAPHS

**Relational hypergraphs.** A *relational hypergraph* $G = (V, E, R, c)$ consists of a set $V$ of *nodes*, a set $E$ of *hyperedges* (or simply *edges* or *facts*) of the form $e = r(u_1, \ldots, u_k) \in E$ where $r \in R$ is a *relation type*, $u_1, \ldots, u_k \in V$ are nodes, and $k = \mathtt{ar}(r)$ is the arity of the relation $r$. We consider labeled hypergraphs, where the labels are given by a coloring function on nodes $c : V \to D$. If the range of this coloring satisfies $D = \mathbb{R}^d$, we say $c$ is a $d$-dimensional *feature map* and use the notation $\boldsymbol{x}$. We write $\rho(e)$ as the relation $r \in R$ of the hyperedge $e \in E$, and $e(i)$ to refer to the node in the $i$-th arity position of the hyperedge $e$. We define $E(v) = \{(e, i) \mid e(i) = v, e \in E, 1 \leq i \leq \mathtt{ar}(\rho(e))\}$ as the set of edge-position pairs of a node $v$. Intuitively, this set captures all occurrences of node $v$ in

different hyperedges and arity positions. We also define the *positional neighborhood* of a hyperedge $e$ with respect to a position $i$ as $\mathcal{N}_i(e) = \{(e(j), j) \mid j \neq i, 1 \leq j \leq \mathtt{ar}(\rho(e))\}$. This set represents all nodes that co-occur with the node at position $i$ in a hyperedge $e$, along with their positions. A *knowledge graph* is a relational hypergraph where for all $r \in R$, $\mathtt{ar}(r) = 2$.

**Link prediction on hyperedges.** Given a relational hypergraph $G = (V, E, R, c)$, and a *query* $q(u_1, ..., u_{t-1}, ?, u_{t+1}..., u_k)$, where $q \in R$ is the query relation and "?" is the querying position, *link prediction* is the problem of scoring all the hyperedges obtained by substituting nodes $v \in V$ in place of "?". We denote a $k$-tuple $(u_1, \dots, u_k)$ by $\boldsymbol{u}$ and the tuple $(u_1, \dots, u_{t-1}, u_{t+1}, \dots, u_k)$ by $\tilde{\boldsymbol{u}}$. For convenience, we commonly write a query as a tuple $\boldsymbol{q} = (q, \tilde{\boldsymbol{u}}, t)$.

**Isomorphisms.** An *isomorphism* from a relational hypergraph $G = (V, E, R, c)$ to a relational hypergraph $G' = (V', E', R, c')$ is a bijection $f : V \to V'$ such that $c(v) = c'(f(v))$ for all $v \in V$, and $r(u_1, \cdots, u_k) \in E$ if and only if $r(f(u_1), \cdots, f(u_k)) \in E'$, for all $r \in R$ and $u_1, \cdots, u_k \in V$.

**Invariants.** For $k \geq 1$, we define a *$k$-ary relational hypergraph invariant* as a function $\xi$ associating with each relational hypergraph $G = (V, E, R, c)$ a function $\xi(G)$ with domain $V^k$ such that for all relational hypergraphs $G, G'$, all isomorphisms $f$ from $G$ to $G'$, and for all $k$-tuples of nodes $\boldsymbol{u} \in V^k$, we have $\xi(G)(\boldsymbol{u}) = \xi(G')(f(\boldsymbol{u}))$.

**Refinements.** Given two relational hypergraph invariants $\xi$ and $\xi'$, we say a function $\xi(G) : V^k \to D$ *refines* a function $\xi'(G) : V^k \to D$, denoted as $\xi(G) \preceq \xi'(G)$, if for all $\boldsymbol{u}, \boldsymbol{u}' \in V^k$, $\xi(G)(\boldsymbol{u}) = \xi(G)(\boldsymbol{u}')$ implies $\xi'(G)(\boldsymbol{u}) = \xi'(G)(\boldsymbol{u}')$. In addition, we call such functions *equivalent*, denoted as $\xi(G) \equiv \xi'(G)$, if $\xi(G) \preceq \xi'(G)$ and $\xi'(G) \preceq \xi(G)$. A $k$-ary relational hypergraph invariant $\xi$ *refines* a $k$-ary relational hypergraph invariant $\xi'$, if $\xi(G)$ refines $\xi'(G)$ for all relational hypergraphs $G$. Similarly for equivalence.

## 4 HYPERGRAPH RELATIONAL MPNNS

We first introduce HR-MPNNs, which capture existing architectures tailored for relational hypergraphs, such as G-MPNN (Yadati, 2020) and RD-MPNN (Zhou et al., 2023) (Appendix C.1).

Let $G = (V, E, R, \boldsymbol{x})$ be a relational hypergraph, where $\boldsymbol{x}$ is a feature map that yields the initial features $\boldsymbol{x}_v = \boldsymbol{x}(v)$ for all nodes $v \in V$. For $\ell \geq 0$, an HR-MPNN iteratively computes a sequence of feature maps $\boldsymbol{h}^{(\ell)} : V \mapsto \mathbb{R}^{d(\ell)}$, where the representations $\boldsymbol{h}_v^{(\ell)} := \boldsymbol{h}^{(\ell)}(v)$ are given by:

$$\boldsymbol{h}_v^{(0)} = \boldsymbol{x}_v,$$

$$\boldsymbol{h}_v^{(\ell+1)} = \text{UP}\Big(\boldsymbol{h}_v^{(\ell)}, \text{AGG}\big(\boldsymbol{h}_v^{(\ell)}, \{\!\{\text{MSG}_{\rho(e)}\big(\{(\boldsymbol{h}_w^{(\ell)}, j) \mid (w, j) \in \mathcal{N}_i(e)\}\big) \mid (e, i) \in E(v)\}\!\}\big)\Big),$$

where UP, AGG, and $\text{MSG}_{\rho(e)}$ are differentiable, *update*, *aggregation*, and relation-specific *message* functions, respectively. These functions are layer-specific, but we omit the superscript $(\ell)$ for brevity. An HR-MPNN has a fixed number of layers $L \geq 0$ and the final representations of nodes are given by the function $\boldsymbol{h}^{(L)} : V \to \mathbb{R}^{d(L)}$. We can then use a $k$-ary decoder $\text{DEC}_q : \mathbb{R}^{d(L) \times k} \to \mathbb{R}$, to produce a score for the likelihood of $q(\boldsymbol{u})$ for $q \in R, \boldsymbol{u} \in V^k$.

HR-MPNNs trivially contain architectures designed for single-relational, undirected hypergraphs, such as HGNN (Feng et al., 2018) and HyperGCN (Yadati et al., 2019) (see Appendix C.2 for details). Furthermore, HR-MPNNs capture relational message passing neural networks on knowledge graphs (Huang et al., 2023), as a special case (see Appendix C.3 for a proof).

MPNNs are well-understood both in terms of their ability to distinguish graph nodes (Morris et al., 2019; Xu et al., 2019) and in terms of their capacity to capture logical node properties (Barceló et al., 2020). This line of work has been extended to relational architectures (Barceló et al., 2022; Huang et al., 2023). In the next subsections, we provide similar characterizations for HR-MPNNs.

### 4.1 A WEISFEILER-LEMAN TEST FOR HR-MPNNS

We formally characterize the ability of HR-MPNNs to distinguish nodes in relational hypergraphs via a variant of the 1-dimensional Weisfeiler-Leman test, namely the *hypergraph relational 1-WL test*, denoted by $\text{hrwl}_1$. The test $\text{hrwl}_1$ is a natural generalization of $\text{rwl}_1$ (Barceló et al., 2022) to

relational hypergraphs. Given a relational hypergraph $G = (V, E, R, c)$, for $\ell \geq 0$, $\mathsf{hrwl}_1$ updates the node colorings as:

$$\mathsf{hrwl}_1^{(0)}(v) = c(v),$$
$$\mathsf{hrwl}_1^{(\ell+1)}(v) = \tau\Big(\mathsf{hrwl}_1^{(\ell)}(v), \{\!\{\big(\{(\mathsf{hrwl}_1^{(\ell)}(w), j) \,|\, (w, j) \in \mathcal{N}_i(e)\}, \rho(e)\big) \,|\, (e, i) \in E(v)\}\!\}\Big).$$

The function $\tau$ is an injective mapping that maps the above pair to a unique color that has not been used in previous iterations: $\mathsf{hrwl}_1^{(\ell)}$ defines a valid node invariant on relational hypergraphs for all $\ell \geq 0$.

As it turns out, $\mathsf{hrwl}_1$ has the same expressive power as HR-MPNNs in terms of distinguishing nodes over relational hypergraphs:

**Theorem 4.1.** *Let $G = (V, E, R, c)$ be a relational hypergraph, then the following statements hold:*

1. *For all initial feature maps $\boldsymbol{x}$ with $c \equiv \boldsymbol{x}$, all* HR-MPNNs *with $L$ layers, and for all $0 \leq \ell \leq L$, it holds that $\mathsf{hrwl}_1^{(\ell)} \preceq \boldsymbol{h}^{(\ell)}$.*

2. *For all $L \geq 0$, there is an initial feature map $\boldsymbol{x}$ with $c \equiv \boldsymbol{x}$ and an* HR-MPNN *with $L$ layers, such that for all $0 \leq \ell \leq L$, we have $\mathsf{hrwl}_1^{(\ell)} \equiv \boldsymbol{h}^{(\ell)}$.*

Intuitively, item (1) states that $\mathsf{hrwl}_1$ upper bounds the power of any HR-MPNN: if the test cannot distinguish two nodes, then HR-MPNNs cannot either. On the other hand, item (2) states that HR-MPNNs can be as expressive as $\mathsf{hrwl}_1$: for any $L$, there is an HR-MPNN that simulates $L$ iterations of the test. In our proof, we explicitly construct this HR-MPNN using a simple architecture: the proof requires a very delicate construction to ensure the HR-MPNN synthetizes the information around a node $v$ (given by its neighborhood $E(v)$), in the same way $\mathsf{hrwl}_1$ does (see Appendix D).

### 4.2 LOGICAL EXPRESSIVENESS OF HR-MPNNS

The previous WL characterization of HR-MPNNs is *non-uniform* in the sense that it holds for a given relational hypergraph $G$. We now turn our attention to a *uniform* analysis of the power of HR-MPNNs and study the problem of which *(node) properties* can be expressed as HR-MPNNs, which is well-suited for the *inductive* setup. Following Barceló et al. (2020), we investigate *logical* classifiers, i.e., those that can be defined in the formalism of first-order logic (FO). Briefly, a first-order formula $\phi(x)$ with one free variable $x$ defines a logical classifier that assigns value true to node $u$ in relational hypergraph $G$ whenever $G \models \phi(u)$. A logical classifier $\phi(x)$ is *captured* by a HR-MPNN $\mathcal{A}$ if for every relational hypergraph $G$ the nodes $u$ that are classified as true by $\phi$ and $\mathcal{A}$ are the same.

**Graded modal logic on hypergraphs.** Barceló et al. (2020) showed that a logical classifier is captured by an MPNN over single-relational undirected graphs if and only if it can be expressed in *graded modal logic* (de Rijke, 2000; Lutz et al., 2001). This result is extended to knowledge graphs by Huang et al. (2023). We consider a variant of graded modal logic for hypergraphs. Fix a set of relation types $R$ and a set of node colors $\mathcal{C}$. The *hypergraph graded modal logic* (HGML) is the fragment of FO containing the following unary formulas. Firstly, $a(x)$ for $a \in \mathcal{C}$ is a formula. Secondly, if $\varphi(x)$ and $\varphi'(x)$ are HGML formulas, then $\neg\varphi(x)$ and $\varphi(x) \wedge \varphi'(x)$ also are. Thirdly, for $r \in R$, $1 \leq i \leq \mathsf{ar}(r)$ and $N \geq 1$:

$$\exists^{\geq N} \tilde{\boldsymbol{y}}\Big(r(y_1, \ldots, y_{i-1}, x, y_{i+1}, \ldots, y_{\mathsf{ar}(r)}) \wedge \Psi(\tilde{\boldsymbol{y}})\Big)$$

is a HGML formula, where $\tilde{\boldsymbol{y}} = (y_1, \ldots, y_{i-1}, y_{i+1}, \ldots, y_{\mathsf{ar}(r)})$ and $\Psi(\tilde{\boldsymbol{y}})$ is a Boolean combination of HGML formulas having free variables from $\tilde{\boldsymbol{y}}$. Intuitively, the formula expresses that $x$ participates in at least $N$ edges $e$ at position $i$, where the remaining nodes in $e$ satisfy condition $\Psi$.

*Example* 4.2. Consider the set of relations from Figure 1 and the property: "$x$ is a person who obtained a degree $y$ of a subject $z$ at a university $m$ that has been awarded less than two prices $p$ of some subject $w$." This can be expressed as the following formula:

$$\phi(x) = \text{Person}(x) \wedge \exists y, z, m\Big(\text{StudyDegree}(x, y, z, m) \wedge \neg\exists^{\geq 2} p, w\,(\text{Awarded}(w, p, m))\Big)$$

It is easy to verify that $\phi(x)$ is indeed a HGML formula. $\diamond$

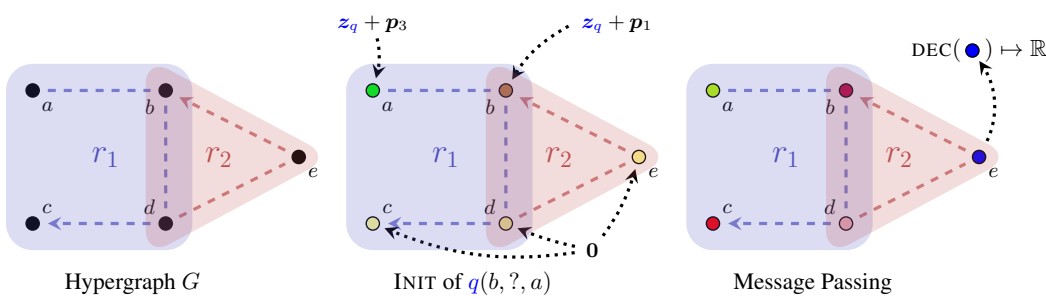

Figure 3: Given a relational hypergraph $G$ with $V = \{a, b, c, d, e\}$, $E = \{r_1(a, b, d, c), r_2(d, e, b)\}$, $R = \{r_1, r_2\}$ and a query $q(b, ?, a)$, HCNet conditions on the nodes $a$ and $b$ and then applies message passing to compute the score for $q(b, e, a)$. Here, $\boldsymbol{z}_q$ is the learnable relation vector for query relation, and $\boldsymbol{p}_i$ is the positional encoding of the $i$-th arity position.

For any property expressed in HGML, such as $\phi(x)$, does there exist an HR-MPNN that captures this property on *all* relational hypergraphs with a shared relational vocabulary $R$ and node colors $\mathcal{C}$? Indeed, we show that HR-MPNNs are as powerful as HGML:

**Theorem 4.3.** *Each hypergraph graded modal logic classifier is captured by a* HR-MPNN.

For the proof, we first show a simple normal form for HGML formulas, and then carefully translate formulas of this form into HR-MPNNs. See Appendix E for further discussion regarding HGML.

## 5 HYPERGRAPH CONDITIONAL MPNNS

In this section, we propose *hypergraph conditional message passing networks* (HC-MPNNs), a generalization of C-MPNNs (Huang et al., 2023) to relational hypergraphs.

Let $G = (V, E, R, \boldsymbol{x})$ be a relational hypergraph, where $\boldsymbol{x}$ is a feature map. Given a query $\boldsymbol{q} = (q, \tilde{\boldsymbol{u}}, t)$, for $\ell \geq 0$, an HC-MPNN computes a sequence of feature maps $\boldsymbol{h}_{v|\boldsymbol{q}}^{(\ell)}$ as follows:

$$\boldsymbol{h}_{v|\boldsymbol{q}}^{(0)} = \text{INIT}(v, \boldsymbol{q}),$$

$$\boldsymbol{h}_{v|\boldsymbol{q}}^{(\ell+1)} = \text{UP}\Big(\boldsymbol{h}_{v|\boldsymbol{q}}^{(\ell)}, \text{AGG}\big(\boldsymbol{h}_{v|\boldsymbol{q}}^{(\ell)}, \{\!\!\{\text{MSG}_{\rho(e)}\big(\{(\boldsymbol{h}_{w|\boldsymbol{q}}^{(\ell)}, j) \mid (w, j) \in \mathcal{N}_i(e)\}, \boldsymbol{q}\big) \mid (e, i) \in E(v)\}\!\!\}\big)\Big),$$

where INIT, UP, AGG, and $\text{MSG}_{\rho(e)}$ are differentiable *initialization*, *update*, *aggregation*, and relation-specific *message* functions, respectively. An HC-MPNN has a fixed number of layers $L \geq 0$, and the final conditional node representations are given by $\boldsymbol{h}_{v|\boldsymbol{q}}^{(L)}$. We denote by $\boldsymbol{h}_{\boldsymbol{q}}^{(\ell)} : V \to \mathbb{R}^{d(\ell)}$ the function $\boldsymbol{h}_{\boldsymbol{q}}^{(\ell)}(v) := \boldsymbol{h}_{v|\boldsymbol{q}}^{(\ell)}$.

To ensure that HC-MPNNs compute $k$-ary representations (see Appendix I), we impose a generalized version of *target node distinguishability* proposed by Huang et al. (2023). An initialization function satisfies *generalized target node distinguishability* if for all $\boldsymbol{q} = (q, \tilde{\boldsymbol{u}}, t)$:

$$\text{INIT}(u, \boldsymbol{q}) \neq \text{INIT}(v, \boldsymbol{q}), \forall u \in \tilde{\boldsymbol{u}}, v \notin \tilde{\boldsymbol{u}} \quad \text{and} \quad \text{INIT}(u_i, \boldsymbol{q}) \neq \text{INIT}(u_j, \boldsymbol{q}), \forall u_i, u_j \in \tilde{\boldsymbol{u}}, u_i \neq u_j$$

Differently from message passing on *simple* hypergraphs, we need to consider the relation type of each edge (multi-relational) and the relative position of each node (directed) in the edges on relational hypergraphs. Hence, the message function $\text{MSG}_{\rho(e)}$ needs to be relation-specific while also keeping track of the positions $j$ of nodes $w$ in their respective neighborhoods $\mathcal{N}_i(e)$. We can then obtain the scores of query $\boldsymbol{q}$ applying a unary decoder DEC on $\boldsymbol{h}_{v|\boldsymbol{q}}^{(L)}$.

## 5.1 Hypergraph conditional networks

We define a *basic* HC-MPNN, which we call *hypergraph conditional networks* (HCNets). For a query $\boldsymbol{q} = (q, \tilde{\boldsymbol{u}}, t)$, an HCNet computes the following representations for all $\ell \geq 0$:

$$\boldsymbol{h}_{v|\boldsymbol{q}}^{(0)} = \sum_{i \neq t} \mathbb{1}_{v = u_i} * (\boldsymbol{p}_i + \boldsymbol{z}_q),$$

$$\boldsymbol{h}_{v|\boldsymbol{q}}^{(\ell+1)} = \sigma \Big( \boldsymbol{W}^{(\ell)} \Big[ \boldsymbol{h}_{v|\boldsymbol{q}}^{(\ell)} \Big\| \sum_{(e,i) \in E(v)} g_{\rho(e),q}^{(\ell)} \Big( \odot_{j \neq i} \, (\alpha^{(\ell)} \boldsymbol{h}_{e(j)|\boldsymbol{q}}^{(\ell)} + (1 - \alpha^{(\ell)}) \boldsymbol{p}_j) \Big) \Big] + \boldsymbol{b}^{(\ell)} \Big),$$

where $g_{\rho(e),q}^{(\ell)}$ is learnable message function, $\sigma$ is an activation function, $\boldsymbol{W}^{(\ell)}$ is a learnable weight matrix, $\boldsymbol{b}^{(\ell)}$ as learnable bias term per layer, $\boldsymbol{z}_q$ is the learnable query vector for $q \in R$, and $\mathbb{1}_C$ is the indicator function that returns 1 if condition $C$ is true, and 0 otherwise. As usual, $*$ is scalar multiplication, and $\odot$ is element-wise multiplication of vectors. We write $\alpha$ to refer to a learnable scalar and $\boldsymbol{p}_i$ to refer to the positional encoding at position $i$, which is sinusoidal positional encoding (Vaswani et al., 2017).

In particular, we set $g_{\rho(e),q}^{(\ell)}$ to be a *query-dependent* diagonal linear map $\mathrm{Diag}(\boldsymbol{W}_r \boldsymbol{z}_q)$ where $\boldsymbol{W}_r$ is a learnable matrix for each relation $r$. Alternatively, we can adopt a *query-independent* map by replacing $\boldsymbol{W}_r \boldsymbol{z}_q$ with learnable vector $\boldsymbol{w}_r$ for each relation $r$.

Intuitively, the model initialization ensures that all *source nodes* (i.e., nodes that appear in $\tilde{\boldsymbol{u}}$) are initialized to their respective positions in the query edge, and all other nodes are initialized as the zero vector $\boldsymbol{0}$ satisfying generalized target node distinguishability, shown in Figure 3.

## 5.2 A Weisfeiler-Leman test for HC-MPNNs

To analyze the expressive power of HC-MPNNs for distinguishing nodes, we can still use the $\mathsf{hrwl}_1$ test provided we restrict ourselves to initial colorings $c$ that respect the given query $\boldsymbol{q}$. Formally, given a query $\boldsymbol{q} = (q, \tilde{\boldsymbol{u}}, t)$ on a relational hypergraph $G = (V, E, R, c)$, we say that the coloring $c$ satisfies *generalized target node distinguishability with respect to $\boldsymbol{q}$* if:

$$c(u) \neq c(v) \quad \forall u \in \tilde{\boldsymbol{u}}, v \notin \tilde{\boldsymbol{u}} \qquad \text{and} \qquad c(u_i) \neq c(u_j) \quad \forall u_i, u_j \in \tilde{\boldsymbol{u}}, u_i \neq u_j.$$

Note that initial colorings satisfying this property are equivalent to the initializations of HC-MPNNs. As a direct consequence of Theorem 4.1 we obtain:

**Theorem 5.1.** *Let $G = (V, E, R, c)$ be a relational hypergraph and $\boldsymbol{q} = (q, \tilde{\boldsymbol{u}}, t)$ be a query such that $c$ satisfies generalized target node distinguishability with respect to $\boldsymbol{q}$. Then the following statements hold:*

1. *For all HC-MPNNs with $L$ layers and initialization INIT with INIT $\equiv c$, $0 \leq \ell \leq L$, we have $\mathsf{hrwl}_1^{(\ell)} \preceq \boldsymbol{h}_{\boldsymbol{q}}^{(\ell)}$.*

2. *For all $L \geq 0$, there is an HC-MPNN with $L$ layers s.t. $0 \leq \ell \leq L$, $\mathsf{hrwl}_1^{(\ell)} \equiv \boldsymbol{h}_{\boldsymbol{q}}^{(\ell)}$ holds.*

Theorem 5.1 tells us that HC-MPNNs are stronger than HR-MPNNs due to the initialization: HC-MPNNs can initialize nodes differently based on the query $\boldsymbol{q}$, whereas HR-MPNNs always assign the same initialization for all queries. In fact, the ternary edges from Figure 2 cannot be distinguished by HR-MPNNs but they can be distinguished by HC-MPNNs.

## 5.3 Logical expressiveness of HC-MPNNs

We remark that Theorem 4.3 can be translated to HC-MPNNs by slightly modifying the logic. We consider *symbolic* queries $\boldsymbol{q} = (q, \tilde{\boldsymbol{b}}, t)$, where now each $b \in \tilde{\boldsymbol{b}}$ is a constant symbol. Our vocabulary contains relation types $r \in R$ and node colors $\mathcal{C}$, as before, and additionally the constants $b \in \tilde{\boldsymbol{b}}$. We define *hypergraph graded modal logic with constants* (HGML$_c$) as HGML but, as atomic cases, we additionally have formulas of the form $\varphi(x) = (x = b)$ for some constant $b$ (see Appendix G for details). This allows us to *identify* variables with individual constants.

*Example* 5.2. Now that we have a richer vocabulary with constants, we can now represent more formulas "conditioned" on the constants appearing in the query. For instance, given a symbolic query with $\tilde{\boldsymbol{b}} = (\text{Physics}, \text{BA})$, we can express a more complex formula $\psi(x)$ that represents "$x$ is a person with a BA degree of Physics at some University $m$, where less than two prizes $p$ in total have been awarded in Physics." as follows:

$$\psi(x) = \text{Person}(x) \wedge \exists y, z, m \Big( \text{StudyDegree}(x, y, z, m) \wedge (z = \text{Physics}) \wedge (y = \text{BA})$$

$$\wedge \neg \left( \exists^{\geq 2} p, w \left( \text{Awarded}(w, p, m) \wedge (w = \text{Physics}) \right) \right) \Big)$$

Note that this formula $\psi(x)$ cannot be expressed as an HGML formula but it can be as an HGML$_c$ formula, due to the additional introduction of constants. ◇

We prove the following result showing that HC-MPNNs can capture richer $k$-ary node properties:

**Theorem 5.3.** *Each HGML$_c$ classifier can be captured by a* HC-MPNN *over valid relational hypergraphs.*

## 6 EXPERIMENTAL EVALUATION

We evaluate HCNet on a broad range of experiments, some of which are reported in the appendix:

- **Inductive experiments** (Section 6.1): We evaluate HCNet for inductive link prediction with relational hypergraphs and report very substantial improvements reflecting on our theoretical findings.
- **Transductive experiments** (Section 6.2): We evaluate HCNet for transductive link prediction with relational hypergraphs and report multiple state-of-the-art results.
- **Ablation on initialization and positional encoding** (Section 6.3): We conduct ablation studies on the choice of initialization and positional encoding in HCNets.
- **Knowledge graph experiments** (Appendix K): HCNet can handle knowledge graphs as a special case and our evaluation shows that it can match the performance of models such as NBFNets.
- **Expressiveness evaluation** (Appendix M): We conduct a synthetic experiment on HyperCycle dataset, building on the counter-example in Figure 2 to showcase the expressivity differences between HR-MPNNs and HC-MPNNs.

**Experimental setups.** In all experiments, we consider a 2-layer MLP as the decoder and adopt layer normalization and dropout in all layers before applying ReLU activation and skip-connection. During the training, we remove edges that are currently being treated as positive tuples to prevent overfitting for each batch. We choose the best checkpoint based on its evaluation of the validation set. In terms of evaluation, we adopt *filtered ranking protocol*. For each test edge $q(u_1, \ldots, u_k)$ where $k = \text{ar}(q)$, and for each position $t \in \{1, ..., k\}$, we replace the $t$-th entities by all other possible entities such that the query after replacement is not in the graph. We consider the query-independent message function for all datasets except WikiPeople. We report Mean Reciprocal Rank (MRR), Hits@1, and Hits@3 for inductive experiments and additionally Hits@10 for transductive experiments as evaluation metrics and provide averaged results of *five* runs on different seeds. We reported standard deviations and execution time & memory used along with all other experiment details in Appendix Q. Furthermore, we provide a detailed discussion of computational complexity between HR-MPNNs and HC-MPNNs in Appendix J. We ran all experiments on a single NVIDIA V100 GPU. The code for experiments is provided in https://anonymous.4open.science/r/HCNet.

### 6.1 INDUCTIVE EXPERIMENTS

**Datasets.** Yadati (2020) constructed three inductive datasets, WP-IND, JF-IND, and MFB-IND from existing transductive datasets on relational hypergraphs: WikiPeople (Guan et al., 2019), JF17K (Wen et al., 2016), and M-FB15K (Fatemi et al., 2020), with their statistics in Table 11.

**Baselines.** We compare with the baseline models HGNN (Feng et al., 2018), HyperGCN (Yadati et al., 2019), and three variants of G-MPNN (Yadati, 2020) with different aggregation functions. Since HGNN and HyperGCN are designed for simple hypergraphs, Yadati (2020) tested them on transformed relational hypergraphs where the relations are ignored. In addition, Yadati (2020) initialized nodes with given node features, whereas we ignore the node feature and initialize each node

Table 1: Results of inductive link prediction experiments. We report MRR, Hits@1, and Hits@3 (higher is better) on test sets.

| | WP-IND | | | JF-IND | | | MFB-IND | | |
|---|---|---|---|---|---|---|---|---|---|
| | MRR | Hits@1 | Hits@3 | MRR | Hits@1 | Hits@3 | MRR | Hits@1 | Hits@3 |
| HGNN | 0.072 | 0.045 | 0.112 | 0.102 | 0.086 | 0.128 | 0.121 | 0.076 | 0.114 |
| HyperGCN | 0.075 | 0.049 | 0.111 | 0.099 | 0.088 | 0.133 | 0.118 | 0.074 | 0.117 |
| G-MPNN-sum | 0.177 | 0.108 | 0.191 | 0.219 | 0.155 | 0.236 | 0.124 | 0.071 | 0.123 |
| G-MPNN-mean | 0.153 | 0.096 | 0.145 | 0.112 | 0.039 | 0.116 | 0.241 | 0.162 | 0.257 |
| G-MPNN-max | 0.200 | 0.125 | 0.214 | 0.216 | 0.147 | 0.240 | 0.268 | 0.191 | 0.283 |
| RD-MPNN | 0.304 | 0.238 | 0.328 | 0.402 | 0.308 | 0.453 | 0.122 | 0.082 | 0.125 |
| HCNet | **0.414** | **0.352** | **0.451** | **0.435** | **0.357** | **0.495** | **0.368** | **0.223** | **0.417** |

Table 2: Results of transductive link prediction experiments on FB-AUTO and WikiPeople.

| | FB-AUTO | | | | WikiPeople | | | |
|---|---|---|---|---|---|---|---|---|
| | MRR | Hits@1 | Hits@3 | Hits@10 | MRR | Hits@1 | Hits@3 | Hits@10 |
| RAE | 0.703 | 0.614 | 0.764 | 0.854 | 0.253 | 0.118 | 0.343 | 0.463 |
| NaLP | 0.672 | 0.611 | 0.712 | 0.774 | 0.338 | 0.272 | 0.362 | 0.466 |
| tNaLP+ | 0.729 | 0.645 | 0.748 | 0.826 | 0.339 | 0.269 | 0.369 | 0.473 |
| HINGE | 0.678 | 0.630 | 0.706 | 0.765 | 0.333 | 0.259 | 0.361 | 0.477 |
| NeuInfer | 0.737 | 0.700 | 0.755 | 0.805 | 0.351 | 0.274 | 0.381 | 0.467 |
| BERT | 0.776 | 0.735 | 0.802 | 0.850 | - | - | - | - |
| HypE | 0.804 | 0.774 | 0.823 | 0.856 | 0.263 | 0.127 | 0.355 | 0.486 |
| RAM | 0.830 | 0.803 | 0.851 | 0.876 | 0.363 | 0.271 | 0.405 | 0.500 |
| S2S | - | - | - | - | 0.364 | 0.273 | 0.402 | 0.503 |
| BoxE | 0.844 | 0.814 | 0.863 | 0.898 | - | - | - | - |
| HyperMLN | 0.831 | 0.803 | 0.851 | 0.877 | - | - | - | - |
| HyConvE | 0.847 | 0.820 | 0.872 | 0.901 | 0.362 | 0.275 | 0.388 | 0.501 |
| ReAIE | 0.861 | 0.836 | 0.877 | 0.908 | - | - | - | - |
| RD-MPNN | 0.810 | 0.714 | 0.880 | 0.888 | - | - | - | - |
| HCNet | **0.871** | **0.842** | **0.892** | **0.922** | **0.421** | **0.344** | **0.457** | **0.565** |

with the respective initialization defined in HCNets. We modify RD-MPNNs (Zhou et al., 2023) by replacing learned entity embeddings to be all one vector $\mathbf{1}^d$ to enable inductive link prediction. We adopt the *batching trick* (Zhu et al., 2021) on MFB-IND. Hyper-parameters are reported in Table 13.

**Results.** We report the inductive experiments results in Table 1, and observe that HCNet outperforms all the existing baseline methods by a large margin, doubling the metric on WP-IND and JF-IND and substantially increasing on MFB-IND. Notably, we emphasize that HCNet does not utilize the provided node features whereas other baseline models do, further highlighting the effectiveness of HCNet in generalizing to entirely new graphs in the absence of node features. This is because HCNet is more expressive by computing query-dependent $k$-ary invariants instead of query-agnostic unary invariants in HR-MPNNs such as RD-MPNNs and G-MPNNs with different aggregation functions. Overall, these results perfectly align with the main theoretical findings presented in this paper.

## 6.2 TRANSDUCTIVE EXPERIMENTS

**Datasets & Baselines.** We evaluate HCNets on the link prediction task with relational hypergraphs, namely the publicly available FB-AUTO (Fatemi et al., 2020) and WikiPeople (Guan et al., 2021). These datasets include facts of different arities up to 9. We have taken the results of embedding methods RAE from Zhang et al. (2018), NaLP from Guan et al. (2019), tNaLP+ from Guan et al. (2021), HINGE from Rosso et al. (2020), NeuInfer from Guan et al. (2020), BERT from Devlin et al. (2019), HypE from Fatemi et al. (2020), BoxE from Abboud et al. (2020), RAM from Liu et al. (2021b), S2S from Di et al. (2021), HyperMLN from Chen et al. (2022), HyConvE from Wang et al. (2023), ReAIE from Fatemi et al. (2023), and GNN method RD-MPNN from Zhou et al.

Table 3: Ablation on Initialization

| INIT | | WP-IND | | JF-IND | |
|---|---|---|---|---|---|
| $z_q$ | $p_i$ | MRR | Hits@3 | MRR | Hits@3 |
| - | - | 0.388 | 0.421 | 0.390 | 0.451 |
| ✓ | - | 0.387 | 0.421 | 0.392 | 0.447 |
| - | ✓ | 0.394 | 0.430 | 0.393 | 0.456 |
| ✓ | ✓ | **0.414** | **0.451** | **0.435** | **0.495** |

Table 4: Ablation on Positional Encoding

| PE | WP-IND | | JF-IND | |
|---|---|---|---|---|
| | MRR | Hits@3 | MRR | Hits@3 |
| Constant | 0.393 | 0.426 | 0.356 | 0.428 |
| One-hot | 0.395 | 0.428 | 0.368 | 0.432 |
| Learnable | 0.396 | 0.425 | 0.416 | 0.480 |
| Sinusoidal | **0.414** | **0.451** | **0.435** | **0.495** |

(2023). The statistics of the datasets are reported in Table 12, and the hyper-parameter choices in Table 14. The full table with additional baselines is in Table 18.

**Results.** We summarize the results for the transductive link prediction tasks and report them in Table 9. HCNet obtains the best results in FB-AUTO and WikiPeople on all metrics. This demonstrates the effectiveness of HCNets also on transductive datasets by outperforming all existing embedding methods specifically designed for transductive link prediction tasks.

## 6.3 ABLATION STUDIES ON THE IMPACT OF INITIALIZATION AND POSITIONAL ENCODING

To assess the contribution of each model component, we conduct ablation studies mainly on different choices of positional encodings and initialization functions on WP-IND and JF-IND datasets with the same empirical setup described in Section 6.1. Complete results are reported in Appendix Q.

**Initialization.** We conduct experiments to validate the impact of different initialization by evaluating all combinations of whether including positional encoding $p_i$ or learnable query vectors $z_q$. From Table 3, we observe that both positional encoding $p_i$ and the relation $z_q$ are essential in the initialization, as removing either of them worsens the overall performance of HCNet. A closer look reveals that the removal of the positional encoding is more detrimental compared to removing relational embedding since the model could deduce the relation types based on implicit information such as the arity of the query relation.

**Positional encoding.** We also examine the importance of the choice of positional encodings, which serves as an indicator of which position the given entities lie in a hyperedge. We provide experiments on multiple choices of positional encodings and report the results in Table 4. Empirically, we notice that the sinusoidal positional encoding produces the best results due to its ability to measure sequential dependency between neighboring entities, compared with one-hot positional encoding which assumes orthogonality among each position. We also notice that learnable embeddings do not produce better results since it is generally hard to learn a suitable embedding that respects the order of the nodes in a relation based on random initialization. Finally, constant embedding evidently performs the worst as it pays no respect to position information and treats all hyperedges with the same set of nodes in the same way regardless of the order of the nodes in these edges.

## 7 SUMMARY, DISCUSSIONS, AND LIMITATIONS

We investigated two frameworks of relational message-passing neural networks on the task of link prediction with relational hypergraphs, namely HR-MPNNs and HC-MPNNs. Furthermore, we studied the expressive power of these two frameworks in terms of relational WL and logical expressiveness. We then proposed a simple yet powerful model instance of HC-MPNNs called HCNet and presented its superior performance on inductive link prediction tasks, which is further supported by additional transductive link prediction and synthetic experiments. One limitation lies in the potentially high computational complexity of our approach when applied to large relational hypergraphs. Our approach is also limited to link prediction and a potential future avenue is to investigate complex query answering on fully relational data. Our study extends the success of link prediction with knowledge graphs to relational hypergraphs where higher arity relations can be effectively modeled with GNNs, advancing applications of graph neural networks to fully relational structures.

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

## A  R-MPNNs AND C-MPNNs

In this section, we follow Huang et al. (2023) and define *relational message passing neural networks* (R-MPNNs) and *conditional message passing neural networks* (C-MPNNs). For ease of presentation, we omit the discussion regarding history functions and readout functions from Huang et al. (2023).

**R-MPNNs.** Let $G = (V, E, R, \boldsymbol{x})$ be a knowledge graph, where $\boldsymbol{x}$ is a feature map. A *relational message passing neural network* (R-MPNN) computes a sequence of feature maps $\boldsymbol{h}^{(\ell)} : V \to \mathbb{R}^{d(\ell)}$, for $\ell \geq 0$. For simplicity, we write $\boldsymbol{h}_v^{(\ell)}$ instead of $\boldsymbol{h}^{(\ell)}(v)$. For each node $v \in V$, the representations $\boldsymbol{h}_v^{(\ell)}$ are iteratively computed as:

$$\boldsymbol{h}_v^{(0)} = \boldsymbol{x}_v$$
$$\boldsymbol{h}_v^{(\ell+1)} = \text{UP}\left(\boldsymbol{h}_v^{(\ell)}, \text{AGG}(\{\!\!\{\text{MSG}_r(\boldsymbol{h}_w^{(\ell)})\mid w \in \mathcal{N}_r(v), r \in R\}\!\!\})\right),$$

where UP, AGG, and $\text{MSG}_r$ are differentiable *update*, *aggregation*, and relation-specific *message* functions, respectively, $\mathcal{N}_r(v) := \{u \mid r(u, v) \in E\}$ is the *neighborhood* of a node $v \in V$ relative to a relation $r \in R$. An R-MPNN has a fixed number of layers $L \geq 0$, and then, the final node representations are given by the map $\boldsymbol{h}^{(L)} : V \to \mathbb{R}^{d(L)}$. The final representations can be used for node-level predictions. For link-level tasks, we use a binary decoder $\text{DEC}_q : \mathbb{R}^{d(L)} \times \mathbb{R}^{d(L)} \to \mathbb{R}$, which produces a score for the likelihood of the fact $q(u, v)$, for $q \in R$.

**C-MPNNs.** Let $G = (V, E, R, \boldsymbol{x})$ be a knowledge graph, where $\boldsymbol{x}$ is a feature map. A *conditional message passing neural network* (C-MPNN) iteratively computes pairwise representations, relative to a fixed query $q \in R$ and a fixed node $u \in V$, as follows:

$$\boldsymbol{h}_{v|u,q}^{(0)} = \text{INIT}(u, v, q)$$
$$\boldsymbol{h}_{v|u,q}^{(\ell+1)} = \text{UP}\left(\boldsymbol{h}_{v|u,q}^{(\ell)}, \text{AGG}(\{\!\!\{\text{MSG}_r(\boldsymbol{h}_{w|u,q}^{(\ell)}, \boldsymbol{z}_q)\mid w \in \mathcal{N}_r(v), r \in R\}\!\!\})\right),$$

where INIT, UP, AGG, and $\text{MSG}_r$ are differentiable *initialization*, *update*, *aggregation*, and relation-specific *message* functions, respectively.

We denote by $\boldsymbol{h}_q^{(\ell)} : V \times V \to \mathbb{R}^{d(\ell)}$ the function $\boldsymbol{h}_q^{(\ell)}(u, v) := \boldsymbol{h}_{v|u,q}^{(\ell)}$, and denote $\boldsymbol{z}_q$ to be a learnable vector representing the query $q \in R$. A C-MPNN has a fixed number of layers $L \geq 0$, and the final pair representations are given by $\boldsymbol{h}_q^{(L)}$. To decode the likelihood of the fact $q(u, v)$ for some $q \in R$, we simply use a unary decoder $\text{DEC} : \mathbb{R}^{d(L)} \to \mathbb{R}$, parameterized by a 2-layer MLP. In addition, we require $\text{INIT}(u, v, q)$ to satisfy *target node distinguishability*: for all $q \in R$ and $v \neq u \in V$, it holds that $\text{INIT}(u, u, q) \neq \text{INIT}(u, v, q)$.

## B  ON REPRESENTATIONS OF HIGH-ARITY FACTS

We carefully distinguish between the task setting of relational hypergraphs (also known as *knowledge hypergraphs* in Fatemi et al. (2020; 2023), or *multi-relational ordered hypergraphs* in Yadati (2020)), hyper-relational knowledge graphs, and $n$-ary relational graphs.

- **Relational hypergraphs.** A relational hypergraph is $G = (V, E, R)$, where each facts in $E$ is represented as $k$-ary tuple $r(u_1, \cdots, u_k)$ for $u_1, \cdots, u_k \in V$ and $r \in R$. As of this work, many works (Fatemi et al., 2020; Yadati, 2020; Zhou et al., 2023) have considered this form of representation.

- **Hyper-relational knowledge graphs.** A hyper-relational knowledge graph $G = (V, E, R)$, where $R$ is a set of relation and each fact in $E$ is represented as a tuple $(u, r, v, \{\!\!\{(qr_i, qv_i) \mid 1 \leq i \leq n\}\!\!\})$ where $u, v \in V$ with $r \in R$ is the main triplet, and $\{\!\!\{(qr_i, qv_i) \mid 1 \leq i \leq n\}\!\!\} \in \mathcal{P}(R \times V)$ is a set of qualifier-value pairs. Note that qualifiers are also chosen from the set of relation $R$ and are used to describe the entities as the additional information stored in the knowledge graph for each triplet. Earlier research (Galkin et al., 2020; Xiong et al., 2023) mainly investigated this form of representation.

- $n$**-ary relational graphs.** A $n$-ary relational graph $G = (V, E, C)$, where each fact in $E$ is represented by a set of role-value tuple $\{\!\!\{(r_i, v_i) \mid 1 \leq i \leq n\}\!\!\} \in \mathcal{P}(C \times V)$, and $C$ is the set of

roles, which are unary relation defined over entity for each fact, acting as additional information. Earlier study (Guan et al., 2019) adopt this form of representation.

To further clarify the difference, we show an example with the following visualization for each type of graph in Figure 4. Given a high-arity fact "Hawking went to Oxford to study Physics and received a BA degree" to be captured:

- In relational hypergraphs, each fact is represented by a tuple (thus the ordering is fixed):

$$\text{StudyDegree(Hawking, Oxford, Physics, BA)}.$$

- In hyper-relational knowledge graphs, each fact is represented by a tuple of main triplet together with a set of qualifier-value pairs:

$$(\text{Hawking, Received, BA}, \{\!\{\text{University(Oxford), Subject(Physics)}\}\!\}).$$

- In a $n$-ary relational graphs, each fact is represented as a set (thus unordered):

$$\{\!\{\text{Person(Hawking), University(Oxford), Subject(Physics), Degree(BA)}\}\!\}.$$

Thus, notice that link prediction with relational hypergraphs is a more general problem setup, where no roles and qualifiers are provided as extra information. This differs from the problem setup of link prediction with hyper-relational knowledge graphs or $n$-ary relational graphs, where the unary relations (qualifiers/roles) are also within the relation vocabulary.

**On transformation between relational hypergraphs and other forms.** Note that relational hypergraphs can transformed into hyper-relational knowledge graphs and $n$-ary relational graphs by generating a brand new qualifier per relation per position. However, note that the hyper-relational knowledge graphs ($n$-ary relational graphs) generated this way are restricted: they must satisfy the property that qualifiers can only appear together with their corresponding relation in the main triplet, i.e., transforming $r(u_1, u_2, u_3, u_4)$ to $(u_1, r, u_4, \{r_2 : u_2, r_3 : u_3\})$ will enforce the newly introduced qualifiers $r_2$ and $r_3$ to appear together with each other and with the main relation $r$. They are a very general form of representation of high-arity facts.

On the other hand, hyper-relational knowledge graphs and $n$-ary relational graphs can be transformed into relational hypergraphs injectively by hashing the relation and qualifiers as a new relation type. However, empirically it is difficult to view these datasets as relational hypergraphs due to the explosion in the number of relations: any combination of existing relations and qualifiers would result in a brand new relation type in a relational hypergraph, which is impractical. Another type of transformation can be applied by directly dropping qualifiers and treating the relation on the main triplet as high-arity relations. Such transformation will lose essential qualifier information and is not injective, which is a significantly difficult and different task.

We also highlight the evaluation differences as experiments on hyper-relational knowledge graphs only corrupt entities in the main triplets, whereas, in link prediction with relational hypergraphs setting, all entities mentioned at all positions are corrupted. We thus opt out datasets of hyper-relational knowledge graphs such as WD50K (Galkin et al., 2024) and focus only on the datasets designed for relational hypergraphs to verify our theoretical expressiveness results.

## C   HR-MPNNs SUBSUME EXISTING MODELS

In this section, we provide further details on how the proposed framework HR-MPNNs subsumes existing models as claimed.

### C.1   HR-MPNNs SUBSUME G-MPNNs AND RD-MPNNs

To see why HR-MPNNs subsume RD-MPNNs (Zhou et al., 2023) and G-MPNNs (Yadati, 2020), which are prominent examples of message passing model on relational hypergraphs in the literature, it suffices to instantiate some components of HR-MPNNs with particular functions.

An RD-MPNN can be seen as an instance of an HR-MPNN that uses summation as AGG, and a relation-specific message function $\text{MSG}_r$ which, for each relation $r$, applies summation followed

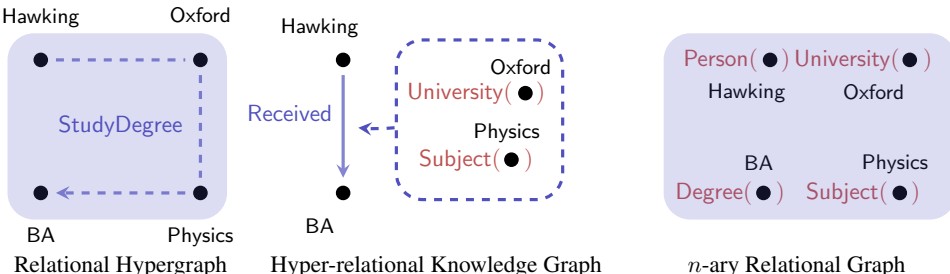

Figure 4: Different ways to represent high-arity fact "Hawking went to Oxford to study Physics and received a BA degree" as hyper-edges.

by a linear map with non-linearity. The update function UP is a one-layer Multi-layer Perceptron (MLP).

Similarly, a G-MPNN instance can be seen as an HR-MPNN that uses either summation, mean, or max as AGG, and a message function $\text{MSG}_r$ which, for each relation $r$, applies a *Hadamard product* of the relational embedding.

## C.2 HR-MPNNs SUBSUMING HGNNS AND HYPERGCNS

To see why HR-MPNNs generalize HGNNs (Feng et al., 2018) and HyperGCNs (Yadati et al., 2019) on simple, undirected hypergraph, first note that (i) these models are single-relational - no relation types - so they are a special case in this sense and (ii) the hyperedges in these undirected hypergraphs are unordered.

To recover HGNN, we can set the message function $\text{MSG}_r$ to be mean, ignoring the relation types $r$, and ignore the relative position in the formula (as there is no ordering in simple, undirected hypergraph). Then, we can choose the AGG function to be symmetrically normalized mean, similar to the aggregation in GCN (Kipf & Welling, 2017).

To recover HyperGCN, we set AGG to be the symmetrically normalized mean, and $\text{MSG}_r$ function to be $w_{i,j} * \arg\max_{\boldsymbol{h}_j} |\boldsymbol{h}_i - \boldsymbol{h}_j|$, with some weight $w_{i,j}$ (again ignoring the relation $r$ and position $i$), provided that the message function has access to the feature of considered node $\boldsymbol{h}_i$.

## C.3 HR-MPNNs SUBSUME R-MPNNs

We formally show that the R-MPNNs framework is subsumed by the HR-MPNNs framework when applied to the knowledge graph.

**Theorem C.1.** *Let $G = (V, E, R, \boldsymbol{x})$ be a knowledge graph, then given any R-MPNN instance $\mathcal{A}$ with $L$ layer parameterized by $\text{AGG}_{\mathcal{A}}^{(\ell)}$, $\text{UP}_{\mathcal{A}}^{(\ell)}$, and $\text{MSG}_{\mathcal{A}r}$ for $0 < \ell \leq L$, $r \in R$, there exists a HR-MPNN instance $\mathcal{B}$ with $L$ layer, parameterized by $\text{AGG}_{\mathcal{B}}^{(\ell)}$, $\text{UP}_{\mathcal{B}}^{(\ell)}$, and $\text{MSG}_{\mathcal{B}r}$, such that for all $v \in V$, we have $\boldsymbol{h}_{\mathcal{A},G}^{(\ell)}(v) = \boldsymbol{h}_{\mathcal{B},G}^{(\ell)}(v)$ for all $0 \leq \ell \leq L$.*

*Proof.* Given an R-MPNN instance $\mathcal{A}$ with $L$ layer, we can have that for $0 \leq \ell \leq L$, we have

$$\boldsymbol{h}_{\mathcal{A},G}^{(0)}(v) = \boldsymbol{x}(v)$$

$$\boldsymbol{h}_{\mathcal{A},G}^{(\ell+1)}(v) = \text{UP}_{\mathcal{A}}^{(\ell)}\left(\boldsymbol{h}_{\mathcal{A},G}^{(\ell)}(v), \text{AGG}_{\mathcal{A}}^{(\ell)}(\{\!\!\{\text{MSG}_{\mathcal{A}r}(\boldsymbol{h}_{\mathcal{A},G}^{(\ell)}(w))|\ w \in \mathcal{N}_r(v), r \in R\}\!\!\})\right),$$

Note that we can now rewrite the updating formula in the following form:

$$\boldsymbol{h}_{\mathcal{A},G}^{(\ell+1)}(v) = \text{UP}_{\mathcal{A}}^{(\ell)}\Big(\boldsymbol{h}_{\mathcal{A},G}^{(\ell)}(v),$$

$$\text{AGG}_{\mathcal{A}}^{(\ell)}\big(\{\!\!\{\text{MSG}_{\mathcal{A}\rho(e)}(\{(\boldsymbol{h}_{\mathcal{A},G}^{(\ell)}(w), j)|(w, j) \in \mathcal{N}_i(e)\})|(e, i) \in E(v), i = 2\}\!\!\}\big)\Big)$$

We then parameterize a HR-MPNN instance $\mathcal{B}$ with $L$ layer of the following form:

$$\boldsymbol{h}_{\mathcal{B},G}^{(0)}(v) = \boldsymbol{x}(v)$$

$$\boldsymbol{h}_{\mathcal{B},G}^{(\ell+1)}(v) = \mathrm{UP}_{\mathcal{B}}^{(\ell)}\Big(\boldsymbol{h}_{\mathcal{B},G}^{(\ell)}(v), \mathrm{AGG}_{\mathcal{B}}^{(\ell)}\big(\boldsymbol{h}_{\mathcal{B},G}^{(\ell)}(v),$$

$$\{\!\!\{\mathrm{MSG}_{\mathcal{B}\rho(e)}\big(\{(\boldsymbol{h}_{\mathcal{B},G}^{(\ell)}(w),j) \,|\, (w,j) \in \mathcal{N}_i(e)\}\big) \,|\, (e,i) \in E(v)\}\!\!\}\big)\Big)$$

where we have for all $0 < \ell \leq L$, $r \in R$, $\mathrm{UP}_{\mathcal{B}}^{(\ell)} := \mathrm{UP}_{\mathcal{A}}^{(\ell)}$, $\mathrm{AGG}_{\mathcal{B}}^{(\ell)}(\boldsymbol{x}, S) := \mathrm{AGG}_{\mathcal{A}}^{(\ell)}(S)$, for some vector $\boldsymbol{x}$ and some (multi-)set $S$, and

$$\mathrm{MSG}_{\mathcal{B}\rho(e)}\big(\{(\boldsymbol{h}^{(\ell)}(w),j) \,|\, (w,j) \in \mathcal{N}_i(e)\}\big) := \mathrm{MSG}_{\mathcal{A}\rho(e)}\big(\{(\boldsymbol{h}^{(\ell)}(w),j) \,|\, (w,j) \in \mathcal{N}_i(e), j=1\}\big)$$

We argue that $\mathrm{MSG}_{\mathcal{B}\rho(e)}$ can be achieved by applying a filtering function on each element of the set to check if the second argument of the tuple is 1 or not.

Now we are ready to prove the theorem by induction. First notice that the base case $\ell = 0$ trivially holds. For the inductive case, assume that for all $v \in V$, we have $\boldsymbol{h}_{\mathcal{A},G}^{(\ell)}(v) = \boldsymbol{h}_{\mathcal{B},G}^{(\ell)}(v)$. Then, notice that for $0 < \ell \leq L$:

$$\boldsymbol{h}_{\mathcal{A},G}^{(\ell+1)}(v) = \mathrm{UP}_{\mathcal{A}}^{(\ell)}\Big(\boldsymbol{h}_{\mathcal{A},G}^{(\ell)}(v),$$

$$\mathrm{AGG}_{\mathcal{A}}^{(\ell)}\big(\{\!\!\{\mathrm{MSG}_{\mathcal{A}\rho(e)}\big(\{(\boldsymbol{h}_{\mathcal{A},G}^{(\ell)}(w),j) \,|\, (w,j) \in \mathcal{N}_i(e)\}\big) \,|\, (e,i) \in E(v), i=2\}\!\!\}\big)\Big)$$

$$= \mathrm{UP}_{\mathcal{A}}^{(\ell)}\Big(\boldsymbol{h}_{\mathcal{A},G}^{(\ell)}(v),$$

$$\mathrm{AGG}_{\mathcal{A}}^{(\ell)}\big(\{\!\!\{\mathrm{MSG}_{\mathcal{A}\rho(e)}\big(\{(\boldsymbol{h}_{\mathcal{A},G}^{(\ell)}(w),j) \,|\, (w,j) \in \mathcal{N}_i(e), j=1\}\big) \,|\, (e,i) \in E(v), i=2\}\!\!\}\big)\Big)$$

$$= \mathrm{UP}_{\mathcal{B}}^{(\ell)}\Big(\boldsymbol{h}_{\mathcal{B},G}^{(\ell)}(v), \mathrm{AGG}_{\mathcal{B}}^{(\ell)}\big(\boldsymbol{h}_{\mathcal{B},G}^{(\ell)}(v),$$

$$\{\!\!\{\mathrm{MSG}_{\mathcal{B}\rho(e)}\big(\{(\boldsymbol{h}_{\mathcal{B},G}^{(\ell)}(w),j) \,|\, (w,j) \in \mathcal{N}_i(e)\}\big) \,|\, (e,i) \in E(v)\}\!\!\}\big)\Big)$$

$$= \boldsymbol{h}_{\mathcal{B},G}^{(\ell+1)}(v)$$

*Remark* C.2. Note that analogously we can show that HC-MPNNs subsumes C-MPNNs by noticing *generalized target node distinguishability* in HC-MPNNs degrades to *target node distinguishability* in the context of knowledge graph. See further detailed discussion in Appendix H.

$\square$

# D   PROOF OF THEOREM 4.1

**Theorem 4.1.** *Let $G = (V, E, R, c)$ be a relational hypergraph, then the following statements hold:*

1. *For all initial feature maps $\boldsymbol{x}$ with $c \equiv \boldsymbol{x}$, all HR-MPNNs with $L$ layers, and for all $0 \leq \ell \leq L$, it holds that $\mathsf{hrwl}_1^{(\ell)} \preceq \boldsymbol{h}^{(\ell)}$.*

2. *For all $L \geq 0$, there is an initial feature map $\boldsymbol{x}$ with $c \equiv \boldsymbol{x}$ and an HR-MPNN with $L$ layers, such that for all $0 \leq \ell \leq L$, we have $\mathsf{hrwl}_1^{(\ell)} \equiv \boldsymbol{h}^{(\ell)}$.*

*Proof.* First, for simplicity of notation, we define $\boldsymbol{m}_{e,i}^{(\ell)} = \mathrm{MSG}_{\rho(e)}\Big(\{(\boldsymbol{h}_w^{(\ell)}, j) \,|\, (w,j) \in \mathcal{N}_i(e)\}\Big)$ for edge $e$, position $1 \leq i \leq \mathsf{ar}(\rho(e))$, and $\ell \geq 0$.

To prove item (1), we first take an initial feature map $\boldsymbol{x}$ with $c \equiv \boldsymbol{x}$ and a HR-MPNN with $L$ layers. We apply induction on $\ell$. The base case where $\ell = 0$ follows directly as $\mathsf{hrwl}_1^{(0)} \equiv c \equiv \boldsymbol{x} \equiv \boldsymbol{h}^{(0)}$. For the inductive case, assume $\mathsf{hrwl}_1^{(\ell+1)}(u) = \mathsf{hrwl}_1^{(\ell+1)}(v)$ for some node pair $u, v \in V$ and for

some $\ell \geq 1$. By injectivity of $\tau$, it follows that $\mathsf{hrwl}_1^{(\ell)}(u) = \mathsf{hrwl}_1^{(\ell)}(v)$ and

$$\{\!\!\{(\{\!\!\{(\mathsf{hrwl}_1^{(\ell)}(w), j) \mid (w, j) \in \mathcal{N}_i(e)\}\!\!\}, \rho(e)) \mid (e, i) \in E(u)\}\!\!\} =$$
$$\{\!\!\{(\{\!\!\{(\mathsf{hrwl}_1^{(\ell)}(w), j) \mid (w, j) \in \mathcal{N}_{i'}(e')\}\!\!\}, \rho(e')) \mid (e', i') \in E(v)\}\!\!\}$$

By inductive hypothesis, we have $\boldsymbol{h}_u^{(\ell)} = \boldsymbol{h}_v^{(\ell)}$ and

$$\{\!\!\{(\{\!\!\{(\boldsymbol{h}_w^{(\ell)}, j) \mid (w, j) \in \mathcal{N}_i(e)\}\!\!\}, \rho(e)) \mid (e, i) \in E(u)\}\!\!\} =$$
$$\{\!\!\{(\{\!\!\{(\boldsymbol{h}_w^{(\ell)}, j) \mid (w, j) \in \mathcal{N}_{i'}(e')\}\!\!\}, \rho(e')) \mid (e', i') \in E(v)\}\!\!\}.$$

Thus we have

$$\{\!\!\{\mathrm{MsG}_{\rho(e)}^{(\ell)}\Big(\{(\boldsymbol{h}_w^{(\ell)}, j) \mid (w, j) \in \mathcal{N}_i(e)\}\Big) \mid (e, i) \in E(u)\}\!\!\} =$$
$$\{\!\!\{\mathrm{MsG}_{\rho(e')}^{(\ell)}\Big(\{(\boldsymbol{h}_w^{(\ell)}, j) \mid (w, j) \in \mathcal{N}_{i'}(e')\}\Big) \mid (e', i') \in E(v)\}\!\!\}$$

and then:

$$\{\!\!\{\boldsymbol{m}_{e,i}^{(\ell)} \mid (e, i) \in E(u)\}\!\!\} = \{\!\!\{\boldsymbol{m}_{e',i'}^{(\ell)} \mid (e', i') \in E(v)\}\!\!\}.$$

We thus conclude that

$$\boldsymbol{h}_u^{(\ell+1)} = \mathrm{UP}^{(\ell)}\Big(\boldsymbol{h}_u^{(\ell)}, \mathrm{AGG}\Big(\boldsymbol{h}_u^{(\ell)}, \{\!\!\{\boldsymbol{m}_{e,i}^{(\ell)} \mid (e, i) \in E(u)\}\!\!\}\Big)\Big)$$
$$= \mathrm{UP}^{(\ell)}\Big(\boldsymbol{h}_v^{(\ell)}, \mathrm{AGG}\Big(\boldsymbol{h}_v^{(\ell)}, \{\!\!\{\boldsymbol{m}_{e',i'}^{(\ell)} \mid (e', i') \in E(v)\}\!\!\}\Big)\Big)$$
$$= \boldsymbol{h}_v^{(\ell+1)}.$$

Now we proceed to show item (2). We use a model of HR-MPNN in the following form and show that any iteration of $\mathsf{hrwl}_1$ can be simulated by a specific layer of such instance of HR-MPNN:

$$\boldsymbol{h}_v^{(0)} = \boldsymbol{x}_v$$
$$\boldsymbol{h}_v^{(\ell+1)} = f^{(\ell)}\Big(\Big[\boldsymbol{h}_v^{(\ell)}\Big\| \sum_{(e,i) \in E(v)} g_{\rho(e)}^{(\ell)}\Big(\odot_{j \neq i}(\boldsymbol{h}_{e(j)}^{(\ell)} + \boldsymbol{p}_j)\Big)\Big]\Big).$$

Here, $f^{(\ell)}(\boldsymbol{z}) = \mathrm{sign}(\boldsymbol{W}^{(\ell)}\boldsymbol{z} - \boldsymbol{b})$ where $\boldsymbol{W}^{(\ell)}$ is a parameter matrix, $\boldsymbol{b}$ is the bias term, in this case the all-ones vector $\boldsymbol{b} = (1, \ldots, 1)^T$, and as non-linearity we use the sign function $\mathrm{sign}$. For a relation type $r \in R$, the function $g_r^{(\ell)}$ has the form $g_r^{(\ell)}(\boldsymbol{z}) = \boldsymbol{Y}_r^{(\ell)}\mathrm{sign}(\boldsymbol{W}_r^{(\ell)}\boldsymbol{z} - \boldsymbol{b})$, where $\boldsymbol{W}_r^{(\ell)}$ and $\boldsymbol{Y}_r^{(\ell)}$ are parameter matrices and $\boldsymbol{b}$ is the all-ones bias vector. Recall that $\odot$ denotes element-wise multiplication and $\boldsymbol{p}_j$ is the positional encoding at position $j$, which in this case is a parameter vector.

We shall use the following lemma shown in Morris et al. (2019)[Lemma 9]. The matrix $\boldsymbol{J}$ denotes the all-ones matrix (with appropriate dimensions).

**Lemma D.1** ((Morris et al., 2019)). *Let $\boldsymbol{B} \in \mathbb{N}^{s \times t}$ be a matrix whose columns are pairwise distinct. Then there is a matrix $\boldsymbol{X} \in \mathbb{R}^{t \times s}$ such that the matrix $\mathrm{sign}(\boldsymbol{X}\boldsymbol{B} - \boldsymbol{J}) \in \{-1, 1\}^{t \times t}$ is non-singular.*

For a matrix $\boldsymbol{B}$, we denote by $\boldsymbol{B}_i$ its $i$-th column. Let $n = |V|$ and without loss of generality assume $V = \{1, \ldots, n\}$. Let $m$ be the maximum arity over all edges of $G$. We will write feature maps $\boldsymbol{h} : V \to \mathbb{R}^d$ for $G = (V, E, R, c)$ also as matrices $\boldsymbol{H} \in \mathbb{R}^{d \times n}$, where the column $\boldsymbol{H}_v$ corresponds to the $d$-dimensional feature vector for node $v$.

Let $\boldsymbol{Fts}$ be the following $nm \times n$ matrix:

$$\boldsymbol{Fts} = \begin{bmatrix} -1 & -1 & \cdots & -1 & -1 \\ \vdots & \vdots & \vdots & \vdots & \vdots \\ -1 & -1 & \cdots & -1 & -1 \\ 1 & -1 & \cdots & -1 & -1 \\ \vdots & \vdots & \vdots & \vdots & \vdots \\ 1 & -1 & \cdots & -1 & -1 \\ \vdots & \ddots & \ddots & \ddots & \vdots \\ 1 & 1 & \cdots & 1 & -1 \\ \vdots & \vdots & \vdots & \vdots & \vdots \\ 1 & 1 & \cdots & 1 & -1 \end{bmatrix}$$

That is, $(\boldsymbol{Fts})_{ij} = -1$ if $m \times j \geq i$, and $(\boldsymbol{Fts})_{ij} = 1$ otherwise. We shall use the columns of $\boldsymbol{Fts}$ as node features in our simulation. The following lemma is a simple variation of Lemma A.5 from Huang et al. (2023), which in turn is a variation of Lemma D.1 above.

**Lemma D.2.** *Let $\boldsymbol{B} \in \mathbb{N}^{s \times t}$ be a matrix such that $t \leq n$, and all the columns are pairwise distinct and different from the all-zeros column. Then there is a matrix $\boldsymbol{X} \in \mathbb{R}^{nm \times s}$ such that the matrix $\mathrm{sign}(\boldsymbol{XB} - \boldsymbol{J}) \in \{-1, 1\}^{nm \times t}$ is precisely the sub-matrix of $\boldsymbol{Fts}$ given by its first $t$ columns.*

*Proof.* Let $\boldsymbol{z} = (1, k+1, (k+1)^2, \ldots, (k+1)^{s-1}) \in \mathbb{N}^{1 \times s}$, where $k$ is the largest entry in $\boldsymbol{B}$, and $\boldsymbol{b} = \boldsymbol{zB} \in \mathbb{N}^{1 \times t}$. By construction, the entries of $\boldsymbol{b}$ are positive and pairwise distinct. Without loss of generality, we assume that $\boldsymbol{b} = (b_1, b_2, \ldots, b_t)$ for $b_1 > b_2 > \cdots > b_t > 0$. As the $b_i$ are ordered, we can choose numbers $x_1, \ldots, x_t \in \mathbb{R}$ such that $b_i \cdot x_j < 1$ if $i \geq j$, and $b_i \cdot x_j > 1$ if $i < j$, for all $i, j \in \{1, \ldots, t\}$. Let $\boldsymbol{x} = (x_1, \ldots, x_t, 2/b_t, \ldots, 2/b_t)^T \in \mathbb{R}^{n \times 1}$. Note that $(2/b_t) \cdot b_i > 1$, for all $i \in \{1, \ldots, t\}$. Let $\boldsymbol{x}' = (\boldsymbol{x}_1, \ldots, \boldsymbol{x}_1, \boldsymbol{x}_2, \ldots, \boldsymbol{x}_2, \ldots, \boldsymbol{x}_n, \ldots, \boldsymbol{x}_n)^T \in \mathbb{R}^{nm \times 1}$ be the vector obtained from $\boldsymbol{x}$ by replacing each entry $\boldsymbol{x}_i$ with $m$ consecutive copies of $\boldsymbol{x}_i$. Then $\mathrm{sign}(\boldsymbol{x}'\boldsymbol{b} - \boldsymbol{J})$ is precisely the sub-matrix of $\boldsymbol{Fts}$ given by its first $t$ columns. We can choose $\boldsymbol{X} = \boldsymbol{x}'\boldsymbol{z} \in \mathbb{R}^{nm \times s}$. $\qquad\square$

We conclude item (2) by showing the following lemma:

**Lemma D.3.** *There exist a family of feature maps $\{\boldsymbol{h}^{(\ell)} : V \to \mathbb{R}^{nm} \mid 0 \leq \ell \leq L\}$, family of matrices $\{\boldsymbol{W}^{(\ell)} \mid 0 \leq \ell < L\}$ and $\{\{\boldsymbol{W}_r^{(\ell)}, \boldsymbol{Y}_r^{(\ell)}\} \mid 0 \leq \ell < L, r \in R\}$, and positional encodings $\{\boldsymbol{p}_j \mid 1 \leq j \leq m\}$ such that:*

- $\boldsymbol{h}^{(\ell)} \equiv \mathrm{hrwl}_1^{(\ell)}$ *for all $0 \leq \ell \leq L$.*

- $\boldsymbol{h}_v^{(\ell)} \in \mathbb{R}^{nm}$ *is a column of $\boldsymbol{Fts}$ for all $0 \leq \ell \leq L$ and $v \in V$.*

- $\boldsymbol{h}_v^{(\ell+1)} = f^{(\ell)}\left(\left[\boldsymbol{h}_v^{(\ell)} \,\middle\|\, \sum_{(e,i) \in E(v)} g_{\rho(e)}^{(\ell)}\left(\odot_{j \neq i}(\boldsymbol{h}_{e(j)}^{(\ell)} + \boldsymbol{p}_j)\right)\right]\right)$ *for all $0 \leq \ell < L$ and $v \in V$, where $f^{(\ell)}$ and $g_r^{(\ell)}$ are defined as above, i.e. $f^{(\ell)}(\boldsymbol{z}) = \mathrm{sign}(\boldsymbol{W}^{(\ell)}\boldsymbol{z} - \boldsymbol{b})$ and $g_r^{(\ell)}(\boldsymbol{z}) = \boldsymbol{Y}_r^{(\ell)}\,\mathrm{sign}(\boldsymbol{W}_r^{(\ell)}\boldsymbol{z} - \boldsymbol{b})$ (vector $\boldsymbol{b}$ is the all-ones vector).*

*Proof.* We proceed by induction on $\ell$. Suppose that the node coloring $\mathrm{hrwl}_1^{(0)} \equiv c$ with colors $1, \ldots, p$, for $p \leq n$. Then we choose $\boldsymbol{h}^{(0)}$ such that $\boldsymbol{h}_v^{(0)} = \boldsymbol{Fts}_{c(v)}$, i.e., $\boldsymbol{h}_v^{(0)}$ is the $c(v)$-th column of $\boldsymbol{Fts}$. Thus, $\boldsymbol{h}^{(0)}$ satisfies the required conditions.

For the inductive case, assume that $\boldsymbol{h}^{(\ell)} \equiv \mathrm{hrwl}_1^{(\ell)}$ for $0 \leq \ell < L$ and that $\boldsymbol{h}_v^{(\ell)}$ is a column of $\boldsymbol{Fts}$ for all $v \in V$. We shall define parameter matrices $\boldsymbol{W}^{(\ell)}$ and $\{\{\boldsymbol{W}_r^{(\ell)}, \boldsymbol{Y}_r^{(\ell)}\} \mid r \in R\}$ and positional encodings $\{\boldsymbol{p}_j \mid 1 \leq j \leq m\}$ such that the conditions of the lemma are satisfied.

For $1 \leq j \leq m$, the positional encoding $\boldsymbol{p}_j$ is independent of $\ell$. Let $\tilde{\boldsymbol{p}}_j = 4\boldsymbol{b} + 8\boldsymbol{e}_j \in \mathbb{R}^m$, where $\boldsymbol{b}$ is the $m$-dimensional all-ones vector and $\boldsymbol{e}_j$ is the $m$-dimensional one-hot encoding of $j$. In other

words, all entries of $\tilde{\boldsymbol{p}}_j$ are 4 except for the $j$-th entry which is 12. We define $\boldsymbol{p}_j = (\tilde{\boldsymbol{p}}_j, \ldots, \tilde{\boldsymbol{p}}_j) \in \mathbb{R}^{nm}$ to be the concatenation of $n$ copies of $\tilde{\boldsymbol{p}}_j$.

Let $r \in R$ and define $E_r^{pos} = \{(e, i) \mid e \in E, \rho(e) = r, 1 \leq i \leq \mathsf{ar}(r)\}$. For $(e, i) \in E_r^{pos}$, define

$$\boldsymbol{o}_{e,i}^{(\ell)} = \odot_{j \neq i}(\boldsymbol{h}_{e(j)}^{(\ell)} + \boldsymbol{p}_j) \qquad \widetilde{\mathsf{col}}_{e,i}^{(\ell)} = \{(\mathsf{hrwl}_1^{(\ell)}(w), j) \mid (w, j) \in \mathcal{N}_i(e)\}.$$

We claim that for $(e, i), (e', i') \in E_r^{pos}$, we have

$$\boldsymbol{o}_{e,i}^{(\ell)} = \boldsymbol{o}_{e',i'}^{(\ell)} \text{ if and only if } \widetilde{\mathsf{col}}_{e,i}^{(\ell)} = \widetilde{\mathsf{col}}_{e',i'}^{(\ell)}.$$

Suppose first that $\widetilde{\mathsf{col}}_{e,i}^{(\ell)} = \widetilde{\mathsf{col}}_{e',i'}^{(\ell)}$. By inductive hypothesis, we have

$$\{(\boldsymbol{h}_w^{(\ell)}, j) \mid (w, j) \in \mathcal{N}_i(e)\} = \{(\boldsymbol{h}_w^{(\ell)}, j) \mid (w, j) \in \mathcal{N}_{i'}(e')\}.$$

It follows that $\boldsymbol{o}_{e,i}^{(\ell)} = \boldsymbol{o}_{e',i'}^{(\ell)}$. Suppose now that $\widetilde{\mathsf{col}}_{e,i}^{(\ell)} \neq \widetilde{\mathsf{col}}_{e',i'}^{(\ell)}$. We consider two cases. Assume first $i \neq i'$. Then $\boldsymbol{o}_{e,i}^{(\ell)}$ and $\boldsymbol{o}_{e',i'}^{(\ell)}$ differ on the $i$-th coordinate, that is, $(\boldsymbol{o}_{e,i}^{(\ell)})_i \neq (\boldsymbol{o}_{e',i'}^{(\ell)})_i$. Indeed, note that the entries of vectors of the form $\boldsymbol{h}_w^{(\ell)} + \boldsymbol{p}_j$ are always prime numbers in $\{3, 5, 11, 13\}$ (the entries of $\boldsymbol{h}_w^{(\ell)}$ are always in $\{-1, 1\}$ by inductive hypothesis). The $i$-th coordinate of all the vector factors in the product $\boldsymbol{o}_{e,i}^{(\ell)} = \odot_{j \neq i}(\boldsymbol{h}_{e(j)}^{(\ell)} + \boldsymbol{p}_j)$ has value 3, and hence $(\boldsymbol{o}_{e,i}^{(\ell)})_i = 3^{\mathsf{ar}(r)-1}$. On the other hand, there exists a vector factor in the product $\boldsymbol{o}_{e',i'}^{(\ell)} = \odot_{j \neq i'}(\boldsymbol{h}_{e'(j)}^{(\ell)} + \boldsymbol{p}_j)$ (the factor $\boldsymbol{h}_{e'(i)}^{(\ell)} + \boldsymbol{p}_i$), whose $i$-th coordinate is 11. Hence $(\boldsymbol{o}_{e,i}^{(\ell)})_i$ and $(\boldsymbol{o}_{e',i'}^{(\ell)})_i$ have different prime factorizations and then they are distinct. Now assume $i = i'$. Since $\widetilde{\mathsf{col}}_{e,i}^{(\ell)} \neq \widetilde{\mathsf{col}}_{e',i'}^{(\ell)}$, there must be a position $j^*$ such that $\mathsf{hrwl}_1^{(\ell)}(e(j^*)) \neq \mathsf{hrwl}_1^{(\ell)}(e'(j^*))$. By inductive hypothesis, $\boldsymbol{h}_{e(j^*)}^{(\ell)} \neq \boldsymbol{h}_{e'(j^*)}^{(\ell)}$. Again by inductive hypothesis, we know that $\boldsymbol{h}_{e(j^*)}^{(\ell)}$ and $\boldsymbol{h}_{e'(j^*)}^{(\ell)}$ are columns of $\boldsymbol{Fts}$, say w.l.o.g. the $k$-th and $k'$-th columns, respectively, for $1 \leq k < k' \leq n$. By construction of $\boldsymbol{Fts}$, all the $m$ entries of $\boldsymbol{h}_{e(j^*)}^{(\ell)}$ from coordinates $\{km + 1, \ldots, km + m\}$ are 1, while these are $-1$ for $\boldsymbol{h}_{e'(j^*)}^{(\ell)}$. We claim that $\boldsymbol{o}_{e,i}^{(\ell)}$ and $\boldsymbol{o}_{e',i'}^{(\ell)}$ differ on the $(km + j^*)$-th coordinate. Consider the product $\boldsymbol{o}_{e,i}^{(\ell)} = \odot_{j \neq i}(\boldsymbol{h}_{e(j)}^{(\ell)} + \boldsymbol{p}_j)$. The $(km+j^*)$-th coordinate of the factor $\boldsymbol{h}_{e(j^*)}^{(\ell)} + \boldsymbol{p}_{j^*}$ is 13, while it is in $\{3, 5\}$ for the remaining factors. For the product $\boldsymbol{o}_{e',i'}^{(\ell)} = \odot_{j \neq i'}(\boldsymbol{h}_{e'(j)}^{(\ell)} + \boldsymbol{p}_j)$, the $(km+j^*)$-th coordinate of the factor $\boldsymbol{h}_{e'(j^*)}^{(\ell)} + \boldsymbol{p}_{j^*}$ is 11, while it is in $\{3, 5\}$ for the remaining factors. Hence $(\boldsymbol{o}_{e,i}^{(\ell)})_{km+j^*}$ and $(\boldsymbol{o}_{e',i'}^{(\ell)})_{km+j^*}$ have different prime factorizations and then they are distinct.

Let $r \in R$. It follows from the previous claim that if we interpret $\boldsymbol{o}^{(\ell)}$ and $\widetilde{\mathsf{col}}^{(\ell)}$ as colorings for $E_r^{pos}$, then these two colorings are equivalent (i.e., the produce the same partition). Let $s_r$ be the number of colors involved in these colorings, and let $\boldsymbol{o}_1, \ldots, \boldsymbol{o}_{s_r} \in \mathbb{R}^{nm}$ be an enumeration of the distinct vectors appearing in $\{\boldsymbol{o}_{e,i}^{(\ell)} \mid (e, i) \in E_r^{pos}\}$. Let $\boldsymbol{S}_r$ be the $(nm \times s_r)$-matrix whose columns are $\boldsymbol{o}_1, \ldots, \boldsymbol{o}_{s_r}$. Fix an enumeration $r_1, \ldots, r_{|R|}$ of $R$ and define $s = \sum_{r \in R} s_r$. Now we are ready to define our sought matrices $\boldsymbol{W}_r^{(\ell)}$ and $\boldsymbol{Y}_r^{(\ell)}$, for $r \in R$. We define $\boldsymbol{W}_r^{(\ell)}$ to be the $(s_r \times nm)$-matrix obtained from applying Lemma D.1 to the matrix $\boldsymbol{S}_r$. Let $\widetilde{\boldsymbol{Y}}_r^{(\ell)} \in \mathbb{R}^{s_r \times s_r}$ be the inverse matrix of $\mathsf{sign}(\boldsymbol{W}_r^{(\ell)}\boldsymbol{S}_r - \boldsymbol{J})$. Suppose $r = r_k$ for $1 \leq k \leq |R|$. Then, the matrix $\boldsymbol{Y}_r^{(\ell)}$ is the $(s \times s_r)$-matrix defined as the vertical concatenation of the following $|R|$ matrices: $\boldsymbol{N}_{r_1}, \ldots, \boldsymbol{N}_{r_{k-1}}, \widetilde{\boldsymbol{Y}}_r^{(\ell)}, \boldsymbol{N}_{r_{k+1}}, \ldots, \boldsymbol{N}_{r_{|R|}}$, where $\boldsymbol{N}_{r'}$ is the all-zeros $(s_{r'} \times s_r)$-matrix. By construction, $\boldsymbol{Y}_r^{(\ell)}\,\mathsf{sign}(\boldsymbol{W}_r^{(\ell)}\boldsymbol{S}_r - \boldsymbol{J})$ is the vertical concatenation of $\boldsymbol{N}_{r_1}, \ldots, \boldsymbol{N}_{r_{k-1}}$, $\boldsymbol{I}_r, \boldsymbol{N}_{r_{k+1}}, \ldots, \boldsymbol{N}_{r_{|R|}}$, where $\boldsymbol{I}_r$ is the $s_r \times s_r$ identity matrix. In particular, if we consider $g_r^{(\ell)}(\boldsymbol{z}) = \boldsymbol{Y}_r^{(\ell)}\,\mathsf{sign}(\boldsymbol{W}_r^{(\ell)}\boldsymbol{z} - \boldsymbol{b})$ as in the statement of the lemma, then for each $(e, i) \in E_r^{pos}$, the vector $\boldsymbol{m}_{e,i}^{(\ell)} = g_r^{(\ell)}(\boldsymbol{o}_{e,i}^{(\ell)})$ has the form $\boldsymbol{m}_{e,i}^{(\ell)} = (\boldsymbol{0}_{r_1}, \ldots, \boldsymbol{0}_{r_{k-1}}, \boldsymbol{c}_{e,i}^{(\ell)}, \boldsymbol{0}_{r_{k+1}}, \ldots, \boldsymbol{0}_{r_{|R|}})^T \in \{0, 1\}^s$, where $\boldsymbol{0}_{r'}$ is the all-zeros vector of dimension $s_{r'}$ and $\boldsymbol{c}_{e,i}^{(\ell)} \in \{0, 1\}^{s_r}$ is a one-hot encoding of edge

color $\boldsymbol{o}_{e,i}^{(\ell)}$, or equivalently, of edge color $\widetilde{\mathsf{col}}_{e,i}^{(\ell)}$. It follows that the vector

$$\boldsymbol{f}_v^{(\ell)} = \sum_{(e,i) \in E(v)} g_{\rho(e)}^{(\ell)}(\boldsymbol{o}_{e,i}^{(\ell)}) = \sum_{r \in R} \sum_{(e,i) \in E(v) \cap E_r^{pos}} g_r^{(\ell)}(\boldsymbol{o}_{e,i}^{(\ell)})$$

has the form $\boldsymbol{f}_v^{(\ell)} = (\boldsymbol{a}_{r_1}, \ldots, \boldsymbol{a}_{r_{|R|}})^T \in \mathbb{N}^s$, where $\boldsymbol{a}_r$ is the $s_r$-dimensional vector whose entry $(\boldsymbol{a}_r)_j$, for $1 \leq j \leq s_r$, is the number of elements $(e,i)$ in $E(v) \cap E_r^{pos}$ with color $j$, that is, such that $\boldsymbol{o}_{e,i}^{(\ell)} = \boldsymbol{o}_j$. In particular, $\boldsymbol{a}_r$ is an encoding of the multiset $\{\!\!\{\widetilde{\mathsf{col}}_{e,i}^{(\ell)} \mid (e,i) \in E(v) \cap E_r^{pos}\}\!\!\}$ and hence $\boldsymbol{f}_v^{(\ell)}$ is an encoding of the multiset $\{\!\!\{(\widetilde{\mathsf{col}}_{e,i}^{(\ell)}, \rho(e)) \mid (e,i) \in E(v)\}\!\!\}$. Note that this multiset is precisely the multiset $\{\!\!\{\mathsf{col}^{(\ell)}(e,i) \mid (e,i) \in E(v)\}\!\!\}$ from the definition of the update rule of the hypergraph relational 1-WL test. Hence, the feature map given by the concatenation $[\boldsymbol{h}_v^{(\ell)} \| \boldsymbol{f}_v^{(\ell)}]$, for all $v \in V$, is equivalent to $\mathsf{hrwl}_1^{(\ell+1)}$.

It remains to define the function $f^{(\ell)}$, given by the parameter matrix $\boldsymbol{W}^{(\ell)}$, so that the feature map $\boldsymbol{h}^{(\ell+1)}$ satisfies the conditions of the lemma. Since the columns of $\boldsymbol{Fts}$ are independent, there exists a matrix $\boldsymbol{M} \in \mathbb{R}^{n \times nm}$ such that $\boldsymbol{MFts}$ is the $n \times n$ identity matrix. Since each $\boldsymbol{h}_v^{(\ell)}$, with $v \in V$, is a column of $\boldsymbol{Fts}$, then $\boldsymbol{Mh}_v^{(\ell)} \in \{0,1\}^n$ corresponds to a one-hot encoding of the column or color $\boldsymbol{h}_v^{(\ell)}$. Let $\boldsymbol{M}'$ be the $(n+s) \times (nm+s)$ matrix with all entries 0 except for the upper-left $(n \times nm)$-submatrix which is $\boldsymbol{M}$, and the lower-right $(s \times s)$-submatrix which is the $(s \times s)$ identity matrix. By construction, we have $\boldsymbol{M}'[\boldsymbol{h}_v^{(\ell)} \| \boldsymbol{f}_v^{(\ell)}] = [\boldsymbol{Mh}_v^{(\ell)} \| \boldsymbol{f}_v^{(\ell)}] \in \mathbb{N}^{n+s}$. Let $\boldsymbol{z}_1, \ldots, \boldsymbol{z}_q$, with $q \leq n$, be the distinct vectors of the form $[\boldsymbol{Mh}_v^{(\ell)} \| \boldsymbol{f}_v^{(\ell)}]$ and let $\boldsymbol{B}$ be the $((n+s) \times q)$-matrix whose columns are precisely $\boldsymbol{z}_1, \ldots, \boldsymbol{z}_q$. We can apply Lemma D.2 to $\boldsymbol{B}$ to obtain a matrix $\boldsymbol{X} \in \mathbb{R}^{nm \times (n+s)}$ such that $\mathsf{sign}(\boldsymbol{XB} - \boldsymbol{J})$ is the matrix given by the first $q$ columns of $\boldsymbol{Fts}$. We define our sought matrix $\boldsymbol{W}^{(\ell)}$ to be $\boldsymbol{W}^{(\ell)} = \boldsymbol{XM}'$. $\qquad\square$

$\qquad\qquad\qquad\qquad\qquad\qquad\qquad\qquad\qquad\qquad\qquad\qquad\qquad\qquad\qquad\qquad\square$

# E HGML AND PROOF OF THEOREM 4.3

## E.1 HGML FORMULAS

Fix a set of relation types $R$ and a set of node colors $\mathcal{C}$. The *hypergraph graded modal logic* (HGML) is the fragment of FO containing the following unary formulas. Firstly, $a(x)$ for $a \in \mathcal{C}$ is a formula. Secondly, if $\varphi(x)$ and $\varphi'(x)$ are HGML formulas, then $\neg\varphi(x)$ and $\varphi(x) \wedge \varphi'(x)$ also are. Thirdly, for $r \in R$, $1 \leq i \leq \mathsf{ar}(r)$ and $N \geq 1$:

$$\exists^{\geq N} \tilde{\boldsymbol{y}} \left( r(y_1, \ldots, y_{i-1}, x, y_{i+1}, \ldots, y_{\mathsf{ar}(r)}) \wedge \Psi(\tilde{\boldsymbol{y}}) \right)$$

is a HGML formula, where $\tilde{\boldsymbol{y}} = (y_1, \ldots, y_{i-1}, y_{i+1}, \ldots, y_{\mathsf{ar}(r)})$ and $\Psi(\tilde{\boldsymbol{y}})$ is a boolean combination of HGML formulas having free variables from $\tilde{\boldsymbol{y}}$. Intuitively, the formula expresses that $x$ participates in at least $N$ edges $e$ at position $i$, such that the remaining nodes in $e$ satisfies $\Psi$.

Let $G = (V, E, R, c)$ be a relational hypergraph where the range of the node coloring $c$ is $\mathcal{C}$. Next, we define the semantics of HGML. We define when a node $v$ of $G$ satisfies a HGML formula $\varphi(x)$, denoted by $G \models \varphi(v)$, recursively as follows:

- if $\varphi(x) = a(x)$ for $a \in \mathcal{C}$, then $G \models \varphi(v)$ iff $a$ is the color of $v$ in $G$, i.e., $c(v) = a$.
- if $\varphi(x) = \neg\varphi'(x)$, then $G \models \varphi(v)$ iff $G \not\models \varphi'(v)$.
- if $\varphi(x) = \varphi'(x) \wedge \varphi''(x)$, then $G \models \varphi(v)$ iff $G \models \varphi'(v)$ and $G \models \varphi''(v)$.
- if $\varphi(x) = \exists^{\geq N} \tilde{\boldsymbol{y}} (r(y_1, \ldots, y_{i-1}, x, y_{i+1}, \ldots, y_{\mathsf{ar}(r)}) \wedge \Psi(\tilde{\boldsymbol{y}}))$ then $G \models \varphi(v)$ iff there exists at least $N$ tuples $(w_1, \ldots w_{i-1}, w_{i+1}, \ldots, w_{\mathsf{ar}(r)})$ of nodes of $G$ such that $r(w_1, \ldots, w_{i-1}, v, w_{i+1}, \ldots, w_{\mathsf{ar}(r)})$ holds in $G$ and the boolean combination $\Psi(w_1, \ldots w_{i-1}, w_{i+1}, \ldots, w_{\mathsf{ar}(r)})$ evaluates to true.

As an example, consider the set of relations from Figure 1, that is, relations $\{\text{Person}(x), \text{StudyDegree}(x, y, z, m), \text{Awarded}(w, p, m)\}$. Consider the property: "$x$ is a person who obtained a degree $y$ of a subject $z$ at a university $m$ that has been awarded less than two prices $p$ of some subject $w$." This can be expressed as the following HGML formula:

$$\phi(x) = \text{Person}(x) \wedge \exists y, z, m\Big(\text{StudyDegree}(x, y, z, m) \wedge \neg\exists^{\geq 2}p, w\,(\text{Awarded}(w, p, m))\Big)$$

Observe that HGML formulas have a restricted form and hence they are not able to represent all logical queries, which hints at the fundamental limitations of our studied models. For instance, formulas in HGML can only express local properties of nodes. That is, properties of the form "a node is connected (via hyper-edges) to other nodes satisfying other (local) properties". This is illustrated in the example above as the variables $y, z, m$ are forced to appear together with $x$ in the hyper-edge $\text{StudyDegree}(x, y, z, m)$. Another limitation of HGML is that once we quantify over the neighboring variables for $x$ (in the example $y, z, m$), we can only check (local) HGML properties separately for the neighboring variables and combine them via Boolean combinations. In the example above, we check the property "$m$ has been awarded less than two prices $p$ of some subject $w$" for university $m$ via the HGML formula $\alpha(m) = \neg\exists^{\geq 2}p, w\,(\text{Awarded}(w, p, m))$. In particular, we cannot check properties that involve simultaneously two or more neighboring variables, as these properties would not be HGML properties (they would not even be unary). As an example, consider the property "$x$ is a person who obtained a degree $y$ of a subject $z$ at a university $m$ that has been awarded less than two prices $p$ in subject $z$." Now we do not impose that $m$ has less than two prices in any subject, but less than two prices in the particular subject $z$ (the same related with person $x$). This can be expressed as:

$$\phi(x) = \text{Person}(x) \wedge \exists y, z, m\Big(\text{StudyDegree}(x, y, z, m) \wedge \neg\exists^{\geq 2}p\,(\text{Awarded}(z, p, m))\Big)$$

Note that this is not an HGML formula as $\beta(m, z) = \neg\exists^{\geq 2}p\,(\text{Awarded}(z, p, m))$ checks a condition that involves two neighboring variables ($m$ and $z$). This violates exactly the requirement discussed above.

### E.2 Proof of Theorem 4.3

Before showing Theorem 4.3, we need to prove an auxiliary result. We define a restriction of HGML, denoted by $\text{HGML}_r$, as follows. $\text{HGML}_r$ is defined as HGML, except for the inductive case

$$\exists^{\geq N}\tilde{\boldsymbol{y}}\left(r(y_1, \ldots, y_{i-1}, x, y_{i+1}, \ldots, y_{\text{ar}(r)}) \wedge \Psi(\tilde{\boldsymbol{y}})\right)$$

where now we impose $\Psi(\tilde{\boldsymbol{y}})$ to be a *conjunction* of HGML formulas with different free variables, that is,

$$\Psi(\tilde{\boldsymbol{y}}) = \varphi_1(y_1) \wedge \cdots \wedge \varphi_{i-1}(y_{i-1}) \wedge \varphi_{i+1}(y_{i+1}) \wedge \cdots \wedge \varphi_{\text{ar}(r)}(y_{\text{ar}(r)}).$$

We have that HGML is actually equivalent to $\text{HGML}_r$.

**Proposition E.1.** *Every HGML formula can be translated into an equivalent $HGML_r$ formula.*

*Proof.* We apply induction to the formulas in HGML. The only interesting case is when the formula has the form

$$\exists^{\geq N}\tilde{\boldsymbol{y}}\left(r(y_1, \ldots, y_{i-1}, x, y_{i+1}, \ldots, y_{\text{ar}(r)}) \wedge \Psi(\tilde{\boldsymbol{y}})\right)$$

for $r \in R$, $1 \leq i \leq \text{ar}(r)$, $N \geq 1$ and a boolean combination $\Psi(\tilde{\boldsymbol{y}})$ of HGML formulas. We can write $\Psi(\tilde{\boldsymbol{y}})$ in disjunctive normal form and since negation and conjunction are part of HGML, we can assume that $\Psi(\tilde{\boldsymbol{y}})$ has the form:

$$\Psi(\tilde{\boldsymbol{y}}) = \bigvee_{1 \leq k \leq q} \varphi_1^k(y_1) \wedge \cdots \wedge \varphi_{i-1}^k(y_{i-1}) \wedge \varphi_{i+1}^k(y_{i+1}) \wedge \cdots \wedge \varphi_{\text{ar}(r)}^k(y_{\text{ar}(r)}).$$

For $1 \leq k \leq d$ and a subset $T \subseteq \{1, \ldots, i-1, i+1, \ldots, \mathsf{ar}(r)\}$, we denote by $\phi_T^k$ the formula

$$\phi_T^k(y_1, \ldots, y_{i-1}, y_{i+1}, \ldots, y_{\mathsf{ar}(r)}) = \bigwedge_{a \in T} \neg \varphi_a^k(y_a) \wedge \bigwedge_{a \notin T} \varphi_a^k(y_a).$$

Note that $\phi_T^k$ expresses that for the $k$-th disjunct of $\Psi$, the conjuncts $\varphi_a^k(y_a)$ that are false are precisely those for which $a \in T$. In particular the $k$-th disjunct of $\Psi$ corresponds to $\phi_\emptyset^k$.

For $S \subseteq \{1, \ldots, d\}$, and a vector $\mathcal{T} = (T_k \subseteq \{1, \ldots, i-1, i+1, \ldots, \mathsf{ar}(r)\} : T_k \neq \emptyset, k \notin S)$, we denote by $\Psi_{S,\mathcal{T}}$ the formula:

$$\Psi_{S,\mathcal{T}}(y_1, \ldots, y_{i-1}, y_{i+1}, \ldots, y_{\mathsf{ar}(r)}) = \bigwedge_{k \in S} \phi_\emptyset^k \wedge \bigwedge_{k \notin S} \phi_{T_k}^k.$$

$\Psi_{S,\mathcal{T}}$ expresses that exactly the $k$-th disjuncts for $k \in S$ are true, and each of the remaining false disjuncts for $k \notin S$ are being falsified by making false precisely the conjuncts $\varphi_a^k(y_a)$, with $a \in T_k$. Since HGML contains negation and conjunction, we can write $\Psi_{S,\mathcal{T}}$ as a conjunction of HGML formulas with different free variables, that is:

$$\Psi_{S,\mathcal{T}}(y_1, \ldots, y_{i-1}, y_{i+1}, \ldots, y_{\mathsf{ar}(r)}) = \alpha_1(y_1) \wedge \cdots \wedge \alpha_{i-1}(y_{i-1}) \wedge \alpha_{i+1}(y_{i+1}) \wedge \cdots \wedge \alpha_{\mathsf{ar}(r)}(y_{\mathsf{ar}(r)}).$$

Define

$$\mathcal{F} := \{\Psi_{S,\mathcal{T}} \mid S \subseteq \{1, \ldots, d\}, S \neq \emptyset, \mathcal{T} = (T_k \subseteq \{1, \ldots, i-1, i+1, \ldots, \mathsf{ar}(r)\} \mid T_k \neq \emptyset, k \notin S)\}.$$

Then by construction, we have that $\Phi$ is true iff exactly one of the formulas in $\mathcal{F}$ is true. It follows that

$$\exists^{\geq N} \tilde{\boldsymbol{y}} \left( r(y_1, \ldots, y_{i-1}, x, y_{i+1}, \ldots, y_{\mathsf{ar}(r)}) \wedge \Psi(\tilde{\boldsymbol{y}}) \right)$$

is equivalent to the HGML$_r$ formula

$$\bigvee_{\substack{(N_{S,\mathcal{T}} \in \mathbb{N} | \Psi_{S,\mathcal{T}} \in \mathcal{F}) \\ \sum_{S,\mathcal{T}} N_{S,\mathcal{T}} = N}} \bigwedge_{\Psi_{S,\mathcal{T}} \in \mathcal{F}} \exists^{\geq N_{S,\mathcal{T}}} \tilde{\boldsymbol{y}} \left( r(y_1, \ldots, y_{i-1}, x, y_{i+1}, \ldots, y_{\mathsf{ar}(r)}) \wedge \widetilde{\Psi}_{S,\mathcal{T}}(\tilde{\boldsymbol{y}}) \right)$$

where

$$\widetilde{\Psi}_{S,\mathcal{T}}(y_1, \ldots, y_{i-1}, y_{i+1}, \ldots, y_{\mathsf{ar}(r)}) = \tilde{\alpha}_1(y_1) \wedge \cdots \wedge \tilde{\alpha}_{i-1}(y_{i-1}) \wedge \tilde{\alpha}_{i+1}(y_{i+1}) \wedge \cdots \wedge \tilde{\alpha}_{\mathsf{ar}(r)}(y_{\mathsf{ar}(r)}).$$

where $\tilde{\alpha}_a(y_a)$ is the translation to HGML$_r$ of the formula $\alpha_a(y_a)$, which we already have by induction. $\qquad \square$

Now we are ready to prove Theorem 4.3.

**Theorem 4.3.** *Each hypergraph graded modal logic classifier is captured by a* HR-MPNN.

*Proof.* We follow a similar strategy than the logic characterizations from Barceló et al. (2020); Huang et al. (2023). Let $\varphi(x)$ be a formula in HGML, where the vocabulary contains relation types $R$ and node colors $\mathcal{C}$. By Proposition E.1, we can assume that $\varphi(x)$ belongs to HGML$_r$. Let $\varphi_1, \ldots, \varphi_L$ be an enumeration of the subformulas of $\varphi$ such that if $\varphi_i$ is a subformula of $\varphi_j$, then $i \leq j$. In particular, $\varphi_L = \varphi$. We shall define an HR-MPNN $\mathcal{B}_\varphi$ with $L$ layers computing $L$-dimensional features in each layer. The idea is that at layer $\ell \in \{1, \ldots, L\}$, the $\ell$-th component of the feature $\boldsymbol{h}_v^{(\ell)}$ is computed correctly and corresponds to 1 if $\varphi_\ell$ is satisfied in node $v$, and 0 otherwise. We add an additional final layer that simply outputs the last component of the feature vector.

We use models of HR-MPNNs of the following form:

$$\boldsymbol{h}_v^{(\ell+1)} = f^{(\ell)} \left( \left[ \boldsymbol{h}_v^{(\ell)} \Big\| \sum_{(e,i) \in E(v)} g_{\rho(e)}^{(\ell)} \left( \odot_{j \neq i} (\boldsymbol{p}_j - \boldsymbol{h}_{e(j)}^{(\ell)}) \right) \right] \right).$$

Here, $f^{(\ell)}(z) = \sigma(W^{(\ell)} z + b)$ where $W^{(\ell)}$ is a parameter matrix, $b$ is the bias term and $\sigma$ is a non-linearity. For a relation type $r \in R$, the function $g_r^{(\ell)}$ has the form $g_r^{(\ell)}(z) = a_r - \sigma(W_r^{(\ell)} z)$, where $W_r^{(\ell)}$ is a parameter matrix and $a_r$ is a parameter vector. Recall that $\odot$ denotes element-wise multiplication and $p_j$ is the positional encoding at position $j$, which in this case is a parameter vector. The parameter matrix $W^{(\ell)}$ will be a $(L \times 2L)$-matrix of the form $W^{(\ell)} = [W_0^{(\ell)} \ I]$, where $W_0^{(\ell)}$ is a $(L \times L)$ parameter matrix and $I$ is the $(L \times L)$ identity matrix. The parameter matrices $W_0^{(\ell)}$ and $W_r^{(\ell)}$ are actually layer independent and hence we omit the superscripts. Therefore, our models are of the following form:

$$h_v^{(\ell+1)} = \sigma\Big( W_0 h_v^{(\ell)} + \sum_{r \in R} \sum_{\substack{(e,i) \in E(v) \\ \rho(e) = r}} \Big( a_r - \sigma(W_r \odot_{j \neq i} (p_j - h_{e(j)}^{(\ell)})) \Big) + b \Big).$$

For the non-linearity $\sigma$ we use the truncated ReLU function $\sigma(x) = \min(\max(0, x), 1)$. Let $m$ be the maximum arity of the relations in $R$. For $1 \leq j \leq m$, the positional encoding $p_j$ is defined as follows. The dimension of $p_j$ must be $L$ (the same as for feature vectors). We define a set of positions $I_j \subseteq \{1, \ldots, L\}$ as follows: $k \in I_j$ iff there exists a subformula of $\varphi$ of the form

$$\exists^{\geq N} \tilde{y} \Big( r(y_1, \ldots, y_{i-1}, x, y_{i+1}, \ldots, y_{\mathsf{ar}(r)}) \wedge \alpha_1(y_1)$$

$$\wedge \cdots \wedge \alpha_{i-1}(y_{i-1}) \wedge \alpha_{i+1}(y_{i+1}) \wedge \cdots \wedge \alpha_{\mathsf{ar}(r)}(y_{\mathsf{ar}(r)}) \Big).$$

such that $j \in \{1, \ldots, i-1, i+1, \ldots, \mathsf{ar}(r)\}$ and $\alpha_j$ is the $k$-th subformula in the enumeration $\varphi_1, \ldots, \varphi_L$. Then we define $p_j$ such that $(p_j)_k = 1$ if $k \in I_j$ and $(p_j)_k = 3$ otherwise.

Now we define the parameter matrices $W_0 \in \mathbb{R}^{L \times L}$ and $W_r \in \mathbb{R}^{L \times L}$, for $r \in R$, together with the bias vector $b$. For $0 \leq \ell < L$, the $\ell$-row of $W_0$ and $W_r$, and the $\ell$-th entry of $a_r$ and $b$ are defined as follows (omitted entries are 0):

1. If $\varphi_\ell(x) = a(x)$ for a color $a \in \mathcal{C}$, then $(W_0)_{\ell\ell} = 1$.

2. If $\varphi_\ell(x) = \neg\varphi_k(x)$ then $(W_0)_{\ell k} = -1$, and $b_\ell = 1$.

3. If $\varphi_\ell(x) = \varphi_j(x) \wedge \varphi_k(x)$ then $(W_0)_{\ell j} = 1$, $(W_0)_{\ell k} = 1$ and $b_\ell = -1$.

4. If

$$\varphi_\ell(x) = \exists^{\geq N} \tilde{y} \Big( r(y_1, \ldots, y_{i-1}, x, y_{i+1}, \ldots, y_{\mathsf{ar}(r)}) \wedge \varphi_{k_1}(y_1)$$

$$\wedge \cdots \wedge \varphi_{k_{i-1}}(y_{i-1}) \wedge \varphi_{k_{i+1}}(y_{i+1}) \wedge \cdots \wedge \varphi_{k_{\mathsf{ar}(r)}}(y_{\mathsf{ar}(r)}) \Big)$$

then $(W_r)_{\ell k_j} = 1$ for $j \in \{1, \ldots, i-1, i+1, \ldots, \mathsf{ar}(r)\}$ and $(a_r)_\ell = 1$ and $b_\ell = -N + 1$.

Let $G = (V, E, R, c)$ be a relational hypergraph with node colors from $\mathcal{C}$. In order to apply $\mathcal{B}_\varphi$ to $G$, we choose initial $L$-dimensional features $h_v^{(0)}$ such that $(h_v^{(0)})_\ell = 1$ if $\varphi_\ell = a(x)$ and $a$ is the color of $v$, and $(h_v^{(0)})_\ell = 0$ otherwise. In other words, the $L$-dimensional initial feature $h_v^{(0)}$ is a one-hot encoding of the color of $v$. To conclude the theorem we show by induction the following statement:

(†) *For all $1 \leq \ell \leq L$, all $1 \leq p \leq \ell$, all $v \in V$, we have $(h_v^{(\ell)})_p = 1$ if and only if $G \models \varphi_p(v)$.*

We start by showing the following:

(⋆) *For all $1 \leq \ell \leq L$, all $v \in V$, and all $1 \leq p \leq L$ such that $\varphi_p(x) = a(x)$ for some $a \in \mathcal{C}$, we have $(h_v^{(\ell)})_p = 1$ if and only if $G \models \varphi_p(v)$.*

We apply induction on $\ell$. For the base case assume $\ell = 1$. Take $v \in V$ and $1 \leq p \leq L$ such that $\varphi_p(x) = a(x)$ for some $a \in \mathcal{C}$. By construction, we have that:

$$(h_v^{(1)})_p = \sigma\Big( (h_v^{(0)})_p \Big) = (h_v^{(0)})_p.$$

By definition of $\boldsymbol{h}^{(0)}$, we obtain that $(\boldsymbol{h}_v^{(1)})_p = 1$ if and only if $G \models \varphi_p(v)$. For the inductive case, suppose $\ell > 1$ and take $v \in V$ and $1 \le p \le L$ such that $\varphi_p(x) = a(x)$ for some $a \in \mathcal{C}$. We have that:

$$(\boldsymbol{h}_v^{(\ell)})_p = \sigma\Big((\boldsymbol{h}_v^{(\ell-1)})_p\Big) = (\boldsymbol{h}_v^{(\ell-1)})_p.$$

By inductive hypothesis we know that $(\boldsymbol{h}_v^{(\ell-1)})_p = 1$ if and only if $G \models \varphi_p(v)$. It follows that $(\boldsymbol{h}_v^{(\ell)})_p = 1$ if and only if $G \models \varphi_p(v)$.

We now prove statement (†). We start with the base case $\ell = 1$. Take $v \in V$. It must be the case that $p = 1$ and hence $\varphi_p(x) = a(x)$ for some $a \in \mathcal{C}$. The result follows from ($\star$).

For the inductive case, take $\ell > 1$. Take $v \in V$ and $1 \le p \le \ell$. We consider several cases:

- Suppose $\varphi_p(x) = a(x)$ for some color $a \in \mathcal{C}$. Then the result follows from ($\star$).

- Suppose that $\varphi_p(x) = \neg\varphi_k(x)$. We have that:

$$(\boldsymbol{h}_v^{(\ell)})_p = \sigma\Big(-(\boldsymbol{h}_v^{(\ell-1)})_k + 1\Big) = -(\boldsymbol{h}_v^{(\ell-1)})_k + 1.$$

  We obtain that $(\boldsymbol{h}_v^{(\ell)})_p = 1$ iff $(\boldsymbol{h}_v^{(\ell-1)})_k = 0$. Since $k \le \ell-1$, we have by inductive hypothesis that $(\boldsymbol{h}_v^{(\ell-1)})_k = 1$ iff $G \models \varphi_k(v)$. It follows that $(\boldsymbol{h}_v^{(\ell)})_p = 1$ iff $G \models \varphi_p(v)$.

- Suppose that $\varphi_p(x) = \varphi_j(x) \wedge \varphi_k(x)$. Then:

$$(\boldsymbol{h}_v^{(\ell)})_p = \sigma\Big((\boldsymbol{h}_v^{(\ell-1)})_j + (\boldsymbol{h}_v^{(\ell-1)})_k - 1\Big).$$

  We obtain that $(\boldsymbol{h}_v^{(\ell)})_p = 1$ iff $(\boldsymbol{h}_v^{(\ell-1)})_j = 1$ and $(\boldsymbol{h}_v^{(\ell-1)})_k = 1$. Since $j, k \le \ell - 1$, we have by inductive hypothesis that $(\boldsymbol{h}_v^{(\ell-1)})_j = 1$ iff $G \models \varphi_j(v)$ and $(\boldsymbol{h}_v^{(\ell-1)})_k = 1$ iff $G \models \varphi_k(v)$. It follows that $(\boldsymbol{h}_v^{(\ell)})_p = 1$ iff $G \models \varphi_p(v)$.

- Suppose that

$$\varphi_p(x) = \exists^{\ge N}\tilde{\boldsymbol{y}}\,\Big(r(y_1,\ldots,y_{i-1},x,y_{i+1},\ldots,y_{\mathsf{ar}(r)}) \wedge \varphi_{k_1}(y_1)$$
$$\wedge\cdots\wedge \varphi_{k_{i-1}}(y_{i-1}) \wedge \varphi_{k_{i+1}}(y_{i+1}) \wedge \cdots \wedge \varphi_{k_{\mathsf{ar}(r)}}(y_{\mathsf{ar}(r)})\Big).$$

  Then:

$$(\boldsymbol{h}_v^{(\ell)})_p = \sigma\Big(\sum_{\substack{(e,q)\in E(v) \\ \rho(e)=r}} \Big(1 - \sigma\big(\sum_{j\neq i} \odot_{t\neq q}(\boldsymbol{p}_t - \boldsymbol{h}_{e(t)}^{(\ell-1)})_{k_j}\big)\Big) - N + 1\Big).$$

  We say that a pair $(e,q) \in E(v)$, with $\rho(e) = r$, is *good* if $q = i$ and $G \models \varphi_{k_j}(e(j))$ for all $j \in \{1,\ldots,i-1,i+1,\ldots,\mathsf{ar}(r)\}$. We claim that $\sum_{j\neq i}\odot_{t\neq q}(\boldsymbol{p}_t - \boldsymbol{h}_{e(t)}^{(\ell-1)})_{k_j} = 0$ if $(e,q)$ is good and $\sum_{j\neq i}\odot_{t\neq q}(\boldsymbol{p}_t - \boldsymbol{h}_{e(t)}^{(\ell-1)})_{k_j} > 1$ otherwise. Suppose $(e,q)$ is good. Then $q = i$. Take $j \neq i$. We have that $\odot_{t\neq i}(\boldsymbol{p}_t - \boldsymbol{h}_{e(t)}^{(\ell-1)})_{k_j} = 0$ since the factor $(\boldsymbol{p}_t - \boldsymbol{h}_{e(t)}^{(\ell-1)})_{k_j} = 0$ when $t = j$. Indeed, by construction, $(\boldsymbol{p}_j)_{k_j} = 1$. Also, since $k_j \le \ell - 1$, we have by inductive hypothesis that $(\boldsymbol{h}_{e(j)}^{(\ell-1)})_{k_j} = 1$ iff $G \models \varphi_{k_j}(e(j))$. Since $(e,q)$ is good, it follows that $(\boldsymbol{h}_{e(j)}^{(\ell-1)})_{k_j} = 1$. Hence $(\boldsymbol{p}_j - \boldsymbol{h}_{e(j)}^{(\ell-1)})_{k_j} = 0$. Suppose now that $(e,q)$ is not good. Assume first that $q = i$. Then there exists $j \neq i$ such that $G \not\models \varphi_{k_j}(e(j))$. We have that $\odot_{t\neq i}(\boldsymbol{p}_t - \boldsymbol{h}_{e(t)}^{(\ell-1)})_{k_j} > 1$. If $t = j$, then we have $(\boldsymbol{p}_t)_{k_j} = 1$. Since $k_j \le \ell - 1$, by inductive hypothesis we have that $(\boldsymbol{h}_{e(j)}^{(\ell-1)})_{k_j} = 1$ iff $G \models \varphi_{k_j}(e(j))$. It follows that $(\boldsymbol{p}_t - \boldsymbol{h}_{e(t)}^{(\ell-1)})_{k_j} = 1$ when $t = j$. If $t \notin \{i,j\}$, then $(\boldsymbol{p}_t)_{k_j} = 3$ and then $(\boldsymbol{p}_t - \boldsymbol{h}_{e(t)}^{(\ell-1)})_{k_j} > 1$. Hence $\odot_{t\neq i}(\boldsymbol{p}_t - \boldsymbol{h}_{e(t)}^{(\ell-1)})_{k_j} > 1$. Suppose now that $q \neq i$. Then we can choose $j = q$ and obtain that $\odot_{t\neq q}(\boldsymbol{p}_t - \boldsymbol{h}_{e(t)}^{(\ell-1)})_{k_j} > 1$.

Indeed, we have $(\boldsymbol{p}_t)_{k_q} = 3$ for all $t \neq q$. Hence all the factors of $\odot_{t \neq q}(\boldsymbol{p}_t - \boldsymbol{h}_{e(t)}^{(\ell-1)})_{k_q}$ are $> 1$ and then the product is $> 1$.

As a consequence of the previous claim, we have that:

$$(\boldsymbol{h}_v^{(\ell)})_p = \sigma\Big(\big|\{(e,i) \in E(v) \mid \rho(e) = r, (e,i) \text{ is good}\}\big| - N + 1\Big).$$

By definition $G \models \varphi_p(v)$ iff $\big|\{(e,i) \in E(v) \mid \rho(e) = r, (e,i) \text{ is good}\}\big| \geq N$. Hence $G \models \varphi_p(v)$ iff $(\boldsymbol{h}_v^{(\ell)})_p = 1$.

$\square$

## F  PROOF OF THEOREM 5.1

**Theorem 5.1.** *Let $G = (V, E, R, c)$ be a relational hypergraph and $\boldsymbol{q} = (q, \tilde{\boldsymbol{u}}, t)$ be a query such that $c$ satisfies target node distinguishability with respect to $\boldsymbol{q}$. Then the following statements hold:*

1. *For all* HC-MPNNs *with $L$ layers and initialization* INIT *with* INIT $\equiv c$, $0 \leq \ell \leq L$, *we have* $\mathsf{hrwl}_1^{(\ell)} \preceq \boldsymbol{h}_{\boldsymbol{q}}^{(\ell)}$.

2. *For all $L \geq 0$, there is an* HC-MPNN *with $L$ layers s.t. $0 \leq \ell \leq L$, $\mathsf{hrwl}_1^{(\ell)} \equiv \boldsymbol{h}_{\boldsymbol{q}}^{(\ell)}$ holds.*

*Proof.* Note that given $G$ and $\boldsymbol{q}$, each HC-MPNN $\mathcal{A}$ with $L$ layers can be translated into a HR-MPNN $\mathcal{B}$ with $L$ layers that produce the same node features in each layer: for $\mathcal{B}$ we choose as initial features, the features obtained from the initialization function of $\mathcal{A}$, and use the same architecture of $\mathcal{A}$ (functions UP, AGG, MSG). On the other hand, each HR-MPNN $\mathcal{B}$ with $L$ layers whose initial features define a coloring that satisfies generalized target node distinguishability with respect to $\boldsymbol{q}$ can be translated into a HC-MPNN $\mathcal{A}$ with $L$ layers that compute the same node features in each layer: we can define the initialization function of $\mathcal{A}$ so that we obtain the initial features of $\mathcal{B}$ and then use the same architecture of $\mathcal{B}$.

Item (1) is obtained by translating the given HC-MPNN into its correspondent HR-MPNN and then invoking Theorem 4.1. Similarly, item (2) is obtained by applying Theorem 4.1 to obtain an equivalent HR-MPNN and then translate it to a HC-MPNN. $\square$

## G  PROOF OF THEOREM 5.3

We consider *symbolic* queries $\boldsymbol{q} = (q, \tilde{\boldsymbol{b}}, t)$, where each $b \in \tilde{\boldsymbol{b}}$ is a constant symbol. We consider vocabularies containing relation types $r \in R$, node colors $\mathcal{C}$, and the constants $b \in \tilde{\boldsymbol{b}}$. In this case, we work with relational hypergraphs $G = (V, E, R, c, (v_b)_{b \in \tilde{\boldsymbol{b}}})$, where the range of the coloring $c$ is $\mathcal{C}$ and $v_b$ is the interpretation of constant $b$. We only focus on *valid* relational hypergraphs, that is, $G = (V, E, R, c, (v_b)_{b \in \tilde{\boldsymbol{b}}})$ such that for all $b, b' \in \tilde{\boldsymbol{b}}$, $b \neq b'$ implies $v_b \neq v_{b'}$.

We define *hypergraph graded modal logic with constants* (HGML$_c$) as HGML but, as atomic cases, we additionally have formulas of the form $\varphi(x) = (x = b)$ for some constant $b$. As expected, we have that HC-MPNNs can capture HGML$_c$ classifiers.

**Theorem 5.3.** *Each HGML$_c$ classifier can be captured by a HC-MPNNs over valid relational hypergraphs.*

*Proof.* The theorem follows by applying the same construction as in the proof of Theorem 4.3. Now we have extra base cases of the form $\varphi(x) = (x = b)$ but the same arguments apply. Note that now we need to define the initial features $\boldsymbol{h}^{(0)}$ via the initialization function of the HC-MPNN. Since we are focusing on valid relational hypergraphs, this can be easily done while satisfying generalized target node distinguishability. $\square$

# H    LINK PREDICTION WITH KNOWLEDGE GRAPHS

An interesting observation is that when we restrict relational hypergraphs to have hyperedges of arity exactly 2, we recover the class of knowledge graphs. C-MPNNs (Huang et al., 2023) are tailored for knowledge graphs and their expressive power has been recently studied extensively, with a focus on their capability for distinguishing *pairs of nodes* (for a formal definition see Appendix A). In this section, we compare HC-MPNNs and C-MPNNs, and hence we are interested in the expressive power of HC-MPNNs in terms of distinguishing pairs of nodes. Note however that, in principle, HC-MPNNs do not compute binary invariants. Indeed, for $q \in R$ and a pair of nodes $u, v$ we can obtain two final features depending on whether we pose the query $q(u, ?)$ or $q(?, v)$. As a convention, we shall define the final feature of the pair $u, v$ as the result of the query $q(u, ?)$. When a HC-MPNN computes binary invariants under this convention, we say the HC-MPNN is *restricted to tail predictions*.

We proceed to show that HC-MPNNs restricted to tail predictions have the same expressive power in terms of distinguishing pairs of nodes as the $\mathsf{rawl}_2^+$ test proposed in Huang et al. (2023). This test is an extension of $\mathsf{rawl}_2$, which in turn, matches the expressive power of C-MPNNs. It follows then that HC-MPNNs are strictly more powerful than C-MPNNs over knowledge graphs. We show this by first defining a variant of the relational WL test which upper bound the expressive power of HC-MPNNs restricting to tail predictions.

Given a knowledge graph $G = (V, E, R, c, \eta)$, where $\eta : V \times V \mapsto D$ is a pairwise coloring satisfying *target node distinguishability*, i.e. $\forall u \neq v, \eta(u, u) \neq \eta(u, v)$, we define a *relational hypergraph conditioned local 2-WL test*, denoted as $\mathsf{hcwl}_2$. $\mathsf{hcwl}_2$ iteratively updates binary coloring $\eta$ as follow for all $\ell \geq 0$:

$$\mathsf{hcwl}_2^{(0)} = \eta(u, v)$$

$$\mathsf{hcwl}_2^{(\ell+1)}(u, v) = \tau\Big(\mathsf{hcwl}_2^{(\ell)}(u, v), \{\!\!\{\big(\{(\mathsf{hcwl}_2^{(\ell)}(u, w), j) \,|\, (w, j) \in \mathcal{N}_i(e)\}, \rho(e)\big) \,|\, (e, i) \in E(v)\}\!\!\}\Big)$$

Note that indeed, $\mathsf{hcwl}_2^{(\ell)}$ computes a binary invariants for all $\ell \geq 0$. First, we show that HC-MPNN restricted on only tails prediction is indeed characterized by $\mathsf{hcwl}_2$. The proof idea is very similar to Theorem 5.1 in Huang et al. (2023).

**Theorem H.1.** *Let $G = (V, E, R, \boldsymbol{x}, \eta)$ be a knowledge graph where $\boldsymbol{x}$ is a feature map and $\eta$ is a pairwise node coloring satisfying* target node distinguishability. *Given a query with $\boldsymbol{q} = (q, \tilde{\boldsymbol{u}}, 2)$, then we have:*

1. *For all* HC-MPNNs *restricted on tails prediction with $L$ layers and initializations* INIT *with* INIT $\equiv \eta$, *and $0 \leq \ell \leq L$, we have $\mathsf{hcwl}_2^{(\ell)} \preceq \boldsymbol{h}_{\boldsymbol{q}}^{(\ell)}$*

2. *For all $L \geq 0$, there is an* HC-MPNN *restricted on tails prediction with $L$ layers such that for all $0 \leq \ell \leq L$, we have $\mathsf{hcwl}_2^{(\ell)} \equiv \boldsymbol{h}_{\boldsymbol{q}}^{(\ell)}$.*

*Proof.* We first rewrite the HC-MPNN restricted on tails predictions in the following form. Given a query $\boldsymbol{q} = (q, \tilde{\boldsymbol{u}}, t)$, we know that since $G$ is a knowledge graph, $\tilde{\boldsymbol{u}}$ only consists of a single node, which we denote as $u$. In addition, since we only consider the case of tail prediction, then we always have $t = 2$. With this restriction, we restate the HC-MPNN restricted on tails prediction on the knowledge graph as follows:

$$\boldsymbol{h}_{v|\boldsymbol{q}}^{(0)} = \text{INIT}(v, \boldsymbol{q}),$$

$$\boldsymbol{h}_{v|\boldsymbol{q}}^{(\ell+1)} = \text{UP}\Big(\boldsymbol{h}_{v|\boldsymbol{q}}^{(\ell)}, \text{AGG}\big(\boldsymbol{h}_{v|\boldsymbol{q}}^{(\ell)}, \{\!\!\{\text{MSG}_{\rho(e)}(\{(\boldsymbol{h}_{w|\boldsymbol{q}}^{(\ell)}, j) \,|\, (w, j) \in \mathcal{N}_i(e)\}), \,|\, (e, i) \in E(v)\}\!\!\}\big)\Big)$$

Now, we follow a similar idea in the proof of C-MPNN for binary invariants (Huang et al., 2023). Let $G = (V, E, R, c, \eta)$ be a knowledge graph where $\eta$ is a pairwise coloring. Construct the auxiliary knowledge graph $G^2 = (V \times V, E', R, c_\eta)$ where $E' = \{r((u, w), (u, v)) \,|\, r(w, v) \in E, r \in R\}$ and $c_\eta$ is the node coloring $c_\eta((u, v)) = \eta(u, v)$. Similar to Theorem 5.1, If $\mathcal{A}$ is a HC-MPNN and $\mathcal{B}$ is an HR-MPNN, we write $\boldsymbol{h}_{\mathcal{A},G}^{(\ell)}(u, v) := \boldsymbol{h}_{(q,(u),2)}^{(\ell)}(v)$ and $\boldsymbol{h}_{\mathcal{B},G^2}^{(\ell)}((u, v)) := \boldsymbol{h}^{(\ell)}((u, v))$ for the features computed by $\mathcal{A}$ and $\mathcal{B}$ over $G$ and $G^2$, respectively. We sometimes write $\mathcal{N}_r^G(e)$

and $E^G(v)$ to emphasize that the positional neighborhood within a hyperedge and set of hyperedges including node $v$ is taken over the knowledge graph $G$, respectively. Finally, we say that an initial feature map $\boldsymbol{y}$ for $G^2$ satisfies generalized target node distinguishability if $\boldsymbol{y}((u,u)) \neq \boldsymbol{y}((u,v))$ for all $u \neq v$. Note here that the generalized target node distinguishability naturally reduced to *target node distinguishability* proposed in Huang et al. (2023) since $\tilde{u}$ is a singleton. Thus, we have the following equivalence between HR-MPNN and HC-MPNN restricted on tail prediction on the knowledge graph.

**Proposition H.2.** *Let $G = (V, E, R, \boldsymbol{x}, \eta)$ be a knowledge graph where $\boldsymbol{x}$ is a feature map, and $\eta$ is a pairwise coloring. Let $q \in R$, then:*

1. *For every* HC-MPNN $\mathcal{A}$ *with $L$ layers, there is an initial feature map $\boldsymbol{y}$ for $G^2$ an* HR-MPNN $\mathcal{B}$ *with $L$ layers such that for all $0 \leq \ell \leq L$ and $u, v \in V$, we have $\boldsymbol{h}_{\mathcal{A},G}^{(\ell)}(u,v) = \boldsymbol{h}_{\mathcal{B},G^2}^{(\ell)}((u,v))$.*

2. *For every initial feature map $\boldsymbol{y}$ for $G^2$ satisfying generalized target node distinguishability and every* HR-MPNN $\mathcal{B}$ *with $L$ layers, there is a* HC-MPNN $\mathcal{A}$ *with $L$ layers such that for all $0 \leq \ell \leq L$ and $u, v \in V$, we have $\boldsymbol{h}_{\mathcal{A},G}^{(\ell)}(u,v) = \boldsymbol{h}_{\mathcal{B},G^2}^{(\ell)}((u,v))$.*

*Proof.* We proceed to show item (1) first. Consider the HR-MPNN $\mathcal{B}$ with the same relational-specific message $\text{MSG}_r$, aggregation $\text{AGG}$, and update functions $\text{UP}$ as $\mathcal{A}$ for all the $L$ layers. The initial feature map $\boldsymbol{y}$ is defined as $\boldsymbol{y}((u,v)) = \text{INIT}(v, (q, (u), 2))$, where $\text{INIT}$ is the initialization function of $\mathcal{A}$. Then, by induction on number of layer $\ell$, we have that for the base case $\ell = 0$, $\boldsymbol{h}_{\mathcal{A}}^{(0)}(u,v) = \text{INIT}(v, (q, (u), 2)) = \boldsymbol{y}((u,v)) = \boldsymbol{h}_{\mathcal{B}}^{(0)}((u,v))$. For the inductive case, assume $\boldsymbol{h}_{\mathcal{A}}^{(\ell)}(u,v) = \boldsymbol{h}_{\mathcal{B}}^{(\ell)}((u,v))$, then

$$
\begin{aligned}
\boldsymbol{h}_{\mathcal{A}}^{(\ell+1)}(u,v) &= \text{UP}\Big(\boldsymbol{h}_{\mathcal{A}}^{(\ell)}(u,v), \text{AGG}\big(\boldsymbol{h}_{\mathcal{A}}^{(\ell)}(u,v), \\
&\qquad \{\!\!\{\text{MSG}_{\rho(e)}\big(\{(\boldsymbol{h}_{\mathcal{A}}^{(\ell)}(u,w),j) \mid (w,j) \in \mathcal{N}_i^G(e)\}\big) \mid (e,i) \in E^G(v)\}\!\!\}\big)\Big) \\
&= \text{UP}\Big(\boldsymbol{h}_{\mathcal{B}}^{(\ell)}((u,v)), \text{AGG}\big(\boldsymbol{h}_{\mathcal{B}}^{(\ell)}((u,v)), \\
&\qquad \{\!\!\{\text{MSG}_{\rho(e)}\big(\{(\boldsymbol{h}_{\mathcal{B}}^{(\ell)}((u,w)),j) \mid (w,j) \in \mathcal{N}_i^{G^2}(e)\}\big) \mid (e,i) \in E^{G^2}(v)\}\!\!\}\big)\Big) \\
&= \boldsymbol{h}_{\mathcal{B}}^{(\ell+1)}((u,v)).
\end{aligned}
$$

To show item (2), we consider $\mathcal{A}$ with the same relational-specific message $\text{MSG}_r$, aggregation $\text{AGG}$, and update functions $\text{UP}$ as $\mathcal{B}$ for all the $L$ layers. We also take initialization function $\text{INIT}$ such that $\text{INIT}(v, (q, (u), 2)) = \boldsymbol{y}((u,v))$. Then, we can follow the same argument for the equivalence as item (1). $\qquad\square$

We then show the equivalence in terms of the relational WL algorithms:

**Proposition H.3.** *Let $G = (V, E, R, c, \eta)$ be a knowledge graph where $\eta$ is a pairwise coloring. For all $\ell \geq 0$ and $u, v \in V$, we have that $\text{hcwl}_2^{(\ell)}(u,v)$ computed over $G$ coincides with $\text{hrwl}_1^{(\ell)}((u,v))$ computed over $G^2 = (V \times V, E', R, c_\eta)$.*

*Proof.* For $\ell = 0$, we have $\text{hcwl}_2^{(0)}(G, u, v) = \eta(u,v) = c_\eta((u,v)) = \text{hrwl}_1^{(0)}(G^2, (u,v))$. For the inductive case, we have that

$$
\begin{aligned}
\text{hcwl}_2^{(\ell+1)}(G, u, v) &= \tau\Big(\text{hcwl}_2^{(\ell)}(G, u, v), \\
&\qquad \{\!\!\{(\{(\text{hcwl}_2^{(\ell)}(G, u, w), j) \mid (w,j) \in \mathcal{N}_i^G(e)\}, \rho(e)) \mid (e,i) \in E^G(v)\}\!\!\}\Big) \\
&= \tau\Big(\text{hrwl}_1^{(\ell)}(G^2, (u,v)), \\
&\qquad \{\!\!\{(\{(\text{hrwl}_1^{(\ell)}(G^2, (u,w)), j) \mid (w,j) \in \mathcal{N}_i^{G^2}(e)\}, \rho(e)) \mid (e,i) \in E^{G^2}(v)\}\!\!\}\Big) \\
&= \text{hrwl}_1^{(\ell+1)}(G^2, (u,v)).
\end{aligned}
$$

$\square$

Now we are ready to show the proof for Theorem H.1. For $G = (V, E, R, \boldsymbol{x}, \eta)$, we consider $G^2 = (V \times V, E', R, c_\eta)$. We start with item (1). Let $\mathcal{A}$ be a HC-MPNN with $L$ layers and initialization INIT satisfying INIT $\equiv \eta$ and let $0 \leq \ell \leq L$. Let $\boldsymbol{y}$ be an initial feature map for $G^2$ and $\mathcal{B}$ be an HR-MPNN with $L$ layers in Proposition H.2, item (1). For the initialization we have $\boldsymbol{y} \equiv c_\eta$ since $\boldsymbol{y}((u, v)) = \text{INIT}(v, (q, (u)), 2)$. Thus, we can proceed and apply Theorem 4.1, item (1) to $G^2$, $\boldsymbol{y}$, and $\mathcal{B}$ and show that $\mathsf{hrwl}_1^{(\ell)} \preceq \boldsymbol{h}_{\mathcal{B}, G^2}^{(\ell)}$, which in turns shows that $\mathsf{hcwl}_2^{(\ell)} \preceq \boldsymbol{h}_{\mathcal{A}, G}^{(\ell)}$.

We then proceed to show item (2). Let $L \geq 0$ be an integer representing a total number of layers. We apply Theorem 4.1, item (2) to $G^2$ and obtain an initial feature map $\boldsymbol{y}$ with $\boldsymbol{y} \equiv c_\eta$ and an HR-MPNN $\mathcal{B}$ with $L$ layer such that $\mathsf{hrwl}_1^{(\ell)} \equiv \boldsymbol{h}_{\mathcal{B}, G^2}^{(\ell)}$ for all $0 \leq \ell \leq L$. We stress again that $\boldsymbol{y}$ and $\eta$ both satisfied generalized target node distinguishability. Now, let $\mathcal{A}$ be the HC-MPNN from Proposition H.2, item (2). We finally have that $\mathsf{hcwl}_2^{(\ell)} \equiv \boldsymbol{h}_{\mathcal{A}, G}^{(\ell)}$ as required. Note that the item (2) again holds for HCNet.

$\square$

We are ready to prove the claim that HC-MPNN is more powerful than C-MPNN by showing the strict containment of their corresponding relational WL test, that is, $\mathsf{hcwl}_2$ and $\mathsf{rawl}_2$. In particular, we show that the defined $\mathsf{hcwl}_2$ is equivalent to $\mathsf{rawl}_2^+$ defined in Huang et al. (2023), via Theorem H.4. Then, by Proposition A.17 in Huang et al. (2023), we have that $\mathsf{rawl}_2^+ \prec \mathsf{rawl}_2$.

The intuition of Theorem H.4 is that for each updating step, $\mathsf{hcwl}_2$ aggregates over all the neighboring edges, which contain both incoming edges and outgoing edges. In addition, $\mathsf{hcwl}_2$ can differentiate between them via the position of the entities in the edge. This is equivalent to aggregating incoming relation and outgoing inversed-relation in $\mathsf{rawl}_2^+$.

**Theorem H.4.** *For all knowledge graph* $G = (V, E, R, c)$, *let* $\mathsf{hcwl}_2^{(0)}(G) \equiv \mathsf{rawl}_2^{+(0)}(G)$, *then* $\mathsf{hcwl}_2^{(\ell)}(G) \equiv \mathsf{rawl}_2^{+(\ell)}(G)$ *for all* $\ell \geq 0$.

*Proof.* First we restate the definition of $\mathsf{hcwl}_2(G)$ and $\mathsf{rawl}_2^+(G)$ for convenience. Given that the query is always a tail query, i.e., $k = 2$, and given a knowledge graph $G = (V, E, R, c)$, we have that the updating formula for $\mathsf{hcwl}_2(G)$ is

$$\mathsf{hcwl}_2^{(\ell+1)}(G, (u, v)) = \tau(\mathsf{hcwl}_2^{(\ell)}(G, (u, v)),$$
$$\{\!\{(\{\!\{(\mathsf{hcwl}_2^{(\ell)}(G, (u, w)), j) \mid (w, j) \in \mathcal{N}_i(e)\}\!\}, \rho(e)) \mid (e, i) \in E(v)\}\!\})$$
$$= \tau(\mathsf{hcwl}_2^{(\ell)}(G, (u, v)),$$
$$\{\!\{(\mathsf{hcwl}_2^{(\ell)}(G, (u, w)), j, \rho(e)) \mid (w, j) \in \mathcal{N}_i(e), (e, i) \in E(v)\}\!\})$$

Note here that the second equation comes from the fact that the maximum arity is always 2. Then, recall the definition of $\mathsf{rawl}_2$. Given a knowledge graph $G = (V, E, R, c, \eta)$, where $\eta$ is a pairwise coloring only, we have

$$\mathsf{rawl}_2^{(\ell+1)}(G, (u, v)) = \tau\big(\mathsf{rawl}_2^{(\ell)}(G, (u, v)), \{\!\{(\mathsf{rawl}_2^{(\ell)}(G, (u, w)), r) \mid w \in \mathcal{N}_r(v), r \in R\}\!\}\big)$$

where $\mathcal{N}_r(v)$ is the relational neighborhood with respect to relation $r \in R$, i.e., $w \in \mathcal{N}_r(v)$ if and only if $r(v, w) \in E$. Equivalently, we can rewrite $\mathsf{rawl}_2$ in the following form:

$$\mathsf{rawl}_2^{(\ell+1)}(G, (u, v)) = \tau\big(\mathsf{rawl}_2^{(\ell)}(G, (u, v)),$$
$$\{\!\{(\mathsf{rawl}_2^{(\ell)}(G, (u, w)), \rho(e)) \mid (w, j) \in \mathcal{N}_i(e), (e, i) \in E(v), i = 1\}\!\}\big)$$

since we only want to obtain the node $w$ as the tails entities in an edge, and thus the second argument of the (only) element in $\mathcal{N}_i(e)$ will always be 2.

For a test $\mathsf{T}$, we sometimes write $\mathsf{T}(G, \boldsymbol{u})$, or $\mathsf{T}(G, u, v)$ in case of binary tests, to emphasize that the test is applied over $G$, and $\mathsf{T}(G)$ for the pairwise/$k$-ary coloring given by the test. Let

$G = (V, E, R, c, \eta)$ be a knowledge graph. The, note that $G^+ = (V, E^+, R^+)$ is the *augmented knowledge graph* where $R^+$ is the disjoint union of $R$ and $\{r^- \mid r \in R\}$, and

$$E^- = \{r^-(v, u) \mid r(u, v) \in E, u \neq v\}$$

$$E^+ = E \cup E^-$$

We can then define

$$E(v) = \{(e, i) \mid e(i) = v, e \in E\}$$
$$E^+(v) = \{(e, i) \mid e(i) = v, e \in E^+\}$$
$$E^-(v) = \{(e, i) \mid e(i) = v, e \in E^-\}.$$

Finally, recall the definition of $\mathsf{rawl}_2^+(G, u, v) = \mathsf{rawl}_2(G^+, u, v)$. We can write this in the equivalent form:

$$\mathsf{rawl}_2^{+(\ell+1)}(G, (u, v)) = \tau\big(\mathsf{rawl}_2^{+(\ell)}(G, (u, v)),$$
$$\{\!\{(\mathsf{rawl}_2^{+(\ell)}(G, (u, w)), \rho(e)) \mid (w, j) \in \mathcal{N}_i(e), (e, i) \in E^+(v), i = 1\}\!\}\big)$$
$$= \tau\big(\mathsf{rawl}_2^{+(\ell)}(G, (u, v)),$$
$$\{\!\{(\mathsf{rawl}_2^{+(\ell)}(G, (u, w)), \rho(e)) \mid (w, j) \in \mathcal{N}_i(e), (e, i) \in E(v), i = 1\}\!\}$$
$$\cup \{\!\{(\mathsf{rawl}_2^{+(\ell)}(G, (u, w)), \rho(e)) \mid (w, j) \in \mathcal{N}_i(e), (e, i) \in E^-(v), i = 1\}\!\}\big)$$

Now we are ready to show the proof. First we show that $\mathsf{hcwl}_2^{(\ell)}(G) \equiv \mathsf{rawl}_2^{+(\ell)}(G)$. We prove by induction the number of layers $\ell$ by showing that for some $u, v \in V$ and for some $\ell$,

$$\mathsf{hcwl}_2^{(\ell+1)}(G, (u, v)) = \mathsf{hcwl}_2^{(\ell+1)}(G, (u', v')) \equiv \mathsf{rawl}_2^{+(\ell)}(G, (u, v)) = \mathsf{rawl}_2^{+(\ell)}(G, (u', v'))$$

By assumption, we know the base case holds. Assume that $\mathsf{hcwl}_2^{(\ell)}(G) \equiv \mathsf{rawl}_2^{+(\ell)}(G)$ for some $\ell \geq 0$, for a pair of node-pair $(u, v), (u', v') \in V^2$, Given that

$$\mathsf{hcwl}_2^{(\ell+1)}(G, (u, v)) = \mathsf{hcwl}_2^{(\ell+1)}(G, (u', v'))$$

By definition, we have that

$$\tau(\mathsf{hcwl}_2^{(\ell)}(G, (u, v)), \{\!\{(\mathsf{hcwl}_2^{(\ell)}(G, (u, w)), j, \rho(e)) \mid (w, j) \in \mathcal{N}_i(e), (e, i) \in E(v)\}\!\}) =$$
$$\tau(\mathsf{hcwl}_2^{(\ell)}(G, (u', v')), \{\!\{(\mathsf{hcwl}_2^{(\ell)}(G, (u', w)), j, \rho(e')) \mid (w, j) \in \mathcal{N}_i(e'), (e', i) \in E(v')\}\!\})$$

Conditioning on $i \in \{1, 2\}$, we can further decompose the set.

$$\tau(\mathsf{hcwl}_2^{(\ell)}(G, (u, v)), \{\!\{(\mathsf{hcwl}_2^{(\ell)}(G, (u, w)), j, \rho(e)) \mid (w, j) \in \mathcal{N}_i(e), (e, i) \in E(v), i = 1\}\!\},$$
$$\cup \{\!\{(\mathsf{hcwl}_2^{(\ell)}(G, (u, w)), j, \rho(e)) \mid (w, j) \in \mathcal{N}_i(e), (e, i) \in E(v), i = 2\}\!\}) =$$
$$\tau(\mathsf{hcwl}_2^{(\ell)}(G, (u', v')), \{\!\{(\mathsf{hcwl}_2^{(\ell)}(G, (u', w)), j, \rho(e')) \mid (w, j) \in \mathcal{N}_i(e'), (e', i) \in E(v'), i = 1\}\!\},$$
$$\cup \{\!\{(\mathsf{hcwl}_2^{(\ell)}(G, (u', w)), j, \rho(e')) \mid (w, j) \in \mathcal{N}_i(e'), (e', i) \in E(v'), i = 2\}\!\})$$

Assume $\tau$ is injective, the three arguments in $\tau$ must match, i.e., $\mathsf{hcwl}_2^{(\ell)}(G, (u, v)) = \mathsf{hcwl}_2^{(\ell)}(G, (u', v'))$, and

$$\{\!\{(\mathsf{hcwl}_2^{(\ell)}(G, (u, w)), j, \rho(e)) \mid (w, j) \in \mathcal{N}_i(e), (e, i) \in E(v), i = 1\}\!\} =$$
$$\{\!\{(\mathsf{hcwl}_2^{(\ell)}(G, (u', w)), j, \rho(e')) \mid (w, j) \in \mathcal{N}_i(e'), (e', i) \in E(v'), i = 1\}\!\}$$

We also have

$$\{\!\{(\mathsf{hcwl}_2^{(\ell)}(G, (u, w)), j, \rho(e)) \mid (w, j) \in \mathcal{N}_i(e), (e, i) \in E(v), i = 2\}\!\} =$$
$$\{\!\{(\mathsf{hcwl}_2^{(\ell)}(G, (u', w)), j, \rho(e')) \mid (w, j) \in \mathcal{N}_i(e'), (e', i) \in E(v'), i = 2\}\!\}$$

By inductive hypothesis, we have that $\mathsf{rawl}_2^{+(\ell)}(G,(u,v)) = \mathsf{rawl}_2^{+(\ell)}(G,(u',v'))$. Thus, we have that

$$\{\!\{(\mathsf{rawl}_2^{+(\ell)}(G,(u,w)), j, \rho(e)) \mid (w,j) \in \mathcal{N}_i(e), (e,i) \in E(v), i=1\}\!\} =$$
$$\{\!\{(\mathsf{rawl}_2^{+(\ell)}(G,(u',w)), j, \rho(e')) \mid (w,j) \in \mathcal{N}_i(e'), (e',i) \in E(v'), i=1\}\!\}$$

and also

$$\{\!\{(\mathsf{rawl}_2^{+(\ell)}(G,(u,w)), j, \rho(e)) \mid (w,j) \in \mathcal{N}_i(e), (e,i) \in E(v), i=2\}\!\} =$$
$$\{\!\{(\mathsf{rawl}_2^{+(\ell)}(G,(u',w)), j, \rho(e')) \mid (w,j) \in \mathcal{N}_i(e'), (e',i) \in E(v'), i=2\}\!\}$$

First, for the first equation, we notice that

$$\{\!\{(\mathsf{rawl}_2^{+(\ell)}(G,(u,w)), j, \rho(e)) \mid (w,j) \in \mathcal{N}_i(e), (e,i) \in E(v), i=1\}\!\} =$$
$$\{\!\{(\mathsf{rawl}_2^{+(\ell)}(G,(u',w)), j, \rho(e')) \mid (w,j) \in \mathcal{N}_i(e'), (e',i) \in E(v'), i=1\}\!\}$$

if and only if

$$\{\!\{(\mathsf{rawl}_2^{+(\ell)}(G,(u,w)), \rho(e)) \mid (w,j) \in \mathcal{N}_i(e), (e,i) \in E(v), i=1\}\!\} =$$
$$\{\!\{(\mathsf{rawl}_2^{+(\ell)}(G,(u',w)), \rho(e')) \mid (w,j) \in \mathcal{N}_i(e'), (e',i) \in E(v'), i=1\}\!\}$$

since the filtered set of pair $(w,j)$ are the same, and the $(\mathsf{rawl}_2^{+(\ell)}(G,(u,w)), \rho(e))$ and $(\mathsf{rawl}_2^{+(\ell)}(G,(u',w)), \rho(e'))$ matches if and only if $(\mathsf{rawl}_2^{+(\ell)}(G,(u,w)), 2, \rho(e))$ and $(\mathsf{rawl}_2^{+(\ell)}(G,(u',w)), 2, \rho(e'))$ matches. This is because we simply augment an additional position indicator 2 in the tuple as we fixed $i=1$, which does not break the equivalence of the statements.

Then, for the second equation, we note that

$$\{\!\{(\mathsf{rawl}_2^{+(\ell)}(G,(u,w)), j, \rho(e)) \mid (w,j) \in \mathcal{N}_i(e), (e,i) \in E(v), i=2\}\!\} =$$
$$\{\!\{(\mathsf{rawl}_2^{+(\ell)}(G,(u',w)), j, \rho(e')) \mid (w,j) \in \mathcal{N}_i(e'), (e',i) \in E(v'), i=2\}\!\}$$

if and only if

$$\{\!\{(\mathsf{rawl}_2^{+(\ell)}(G,(u,w)), \rho(e)) \mid (w,j) \in \mathcal{N}_i(e), (e,i) \in E^-(v), i=1\}\!\} =$$
$$\{\!\{(\mathsf{rawl}_2^{+(\ell)}(G,(u',w)), \rho(e')) \mid (w,j) \in \mathcal{N}_i(e'), (e',i) \in E^-(v'), i=1\}\!\}$$

since this time the filtered set of pair $(w,j)$ also matches, but for the inverse relation. For any edge $e \in E(v)$ where $(w,1) \in \mathcal{N}_e(v)$, the edge will be in form $\rho(e)(w,v)$ as $w$ is placed in the first position. Thus, there will be a corresponding reversed edge $\rho(e)^{-1}(v,w) \in E^-$ by definition. Then, by the same argument as in the second equation above, adding such an additional position indicator 1 on every tuple will not break the equivalence of the statement.

An important observation is that since the inverse relations are freshly created, we will never mix up these inverse edges in both tests. For $\mathsf{rawl}_2^+$, we can distinguish these edges by checking the freshly created relation symbols $r^{-1} \in R^+ \backslash R$, whereas in $\mathsf{hcwl}_2$, the neighboring nodes from these edges are identified with the position indicator 1 in the tuple.

Thus, we have that

$$\{\!\{(\mathsf{rawl}_2^{+(\ell)}(G,(u,w)), \rho(e)) \mid (w,j) \in \mathcal{N}_i(e), (e,i) \in E(v), i=1\}\!\} =$$
$$\{\!\{(\mathsf{rawl}_2^{+(\ell)}(G,(u',w)), \rho(e')) \mid (w,j) \in \mathcal{N}_i(e'), (e',i) \in E(v'), i=1\}\!\}$$

and also

$$\{\!\{(\mathsf{rawl}_2^{+(\ell)}(G,(u,w)), \rho(e)) \mid (w,j) \in \mathcal{N}_i(e), (e,i) \in E^-(v), i=1\}\!\}) =$$
$$\{\!\{(\mathsf{rawl}_2^{+(\ell)}(G,(u',w)), \rho(e')) \mid (w,j) \in \mathcal{N}_i(e'), (e',i) \in E^-(v'), i=1\}\!\})$$

Since $\tau$ is injective, this is equivalent to

$$\tau\big(\mathsf{rawl}_2^{+(\ell)}(G,(u,v)),\{\!\{(\mathsf{rawl}_2^{+(\ell)}(G,(u,w)),\rho(e))\mid(w,j)\in\mathcal{N}_i(e),(e,i)\in E(v),i=1\}\!\}$$

$$\cup\{\!\{(\mathsf{rawl}_2^{+(\ell)}(G,(u,w)),\rho(e))\mid(w,j)\in\mathcal{N}_i(e),(e,i)\in E^-(v),i=1\}\!\}\big)=$$

$$\tau\big(\mathsf{rawl}_2^{+(\ell)}(G,(u',v')),\{\!\{(\mathsf{rawl}_2^{+(\ell)}(G,(u',w)),\rho(e'))\mid(w,j)\in\mathcal{N}_i(e'),(e',i)\in E(v'),i=1\}\!\}$$

$$\cup\{\!\{(\mathsf{rawl}_2^{+(\ell)}(G,(u',w)),\rho(e'))\mid(w,j)\in\mathcal{N}_i(e'),(e',i)\in E^-(v'),i=1\}\!\}\big)$$

and thus, we have

$$\tau\big(\mathsf{rawl}_2^{+(\ell)}(G,(u,v)),\{\!\{(\mathsf{rawl}_2^{+(\ell)}(G,(u,w)),\rho(e))\mid(w,j)\in\mathcal{N}_i(e),(e,i)\in E^+(v),i=1\}\!\}\big)=$$

$$\tau\big(\mathsf{rawl}_2^{+(\ell)}(G,(u',v')),\{\!\{(\mathsf{rawl}_2^{+(\ell)}(G,(u',w)),\rho(e'))\mid(w,j)\in\mathcal{N}_i(e'),(e',i)\in E^+(v'),i=1\}\!\}\big)$$

and finally

$$\mathsf{rawl}_2^{+(\ell+1)}(G,(u,v))=\mathsf{rawl}_2^{+(\ell+1)}(G,(u',v'))$$

Note that since all arguments apply for both directions, the converse holds. □

*Remark* H.5. We remark that the idea of HC-MPNNs restricted to tail predictions can be extended to arbitrary relational hypergraphs in order to compute $k$-ary invariants for any $k$. See Appendix I for a discussion.

# I COMPUTING $k$-ARY INVARIANTS

In this section, we present a canonical way to construct a valid $k$-ary invariants. We start by introducing a construction of a valid $k$-ary invariants termed as *atomic types*, following the convention by Grohe (2021).

## I.1 ATOMIC TYPES

Given a relational hypergraph $G=(V,E,R,c)$ with $l$ labels and a tuple $\boldsymbol{u}=(u_1,...,u_k)\in V^k$, where $k>1$, we define the *atomic type* of $\boldsymbol{u}$ in $G$ as a vector:

$$\mathtt{atp}_k(G)(\boldsymbol{u})\in\{0,1\}^{lk+\binom{k}{2}+m^2+|R|k^m},$$

where $l$ is the number of colors and $m$ is the arity of the relation with maximum arity. We use the first $lk$ bits to represent the color of the $k$ nodes in $\boldsymbol{u}$, another $\binom{k}{2}$ bits to indicate whether node $u_i$ is identical to $u_j$. We then represent the order of these nodes using $m^2$ bits and finally represent the relation with additional $|R|k^m$ bits.

Atomic types are $k$-ary relational hypergraph invariants as they satisfy the property that $\mathtt{atp}_k(G)(\boldsymbol{u})=\mathtt{atp}_k(G')(\boldsymbol{u}')$ if and only if the mapping $u_1\mapsto u'_1$, ..., $u_k\mapsto u'_k$ is an isomorphism from the induced subgraph $G[\{u_1,\cdots,u_k\}]$ to $G'[\{u'_1,\cdots,u'_k\}]$.

## I.2 RELATIONAL HYPERGRAPH CONDITIONED LOCAL $k$-WL TEST

Now we are ready to show the $k$-ary invariants. Similarly to $\mathsf{hcwl}_2$, we can restrict HC-MPNN to only carry out a tail prediction with relational hypergraphs to make sure it directly computes $k$-ary invariants. Here, we introduce *Relational hypergraph conditioned local $k$-WL test*, dubbed $\mathsf{hcwl}_k$, which naturally generalized $\mathsf{hcwl}_2$ to relational hypergraph. Given $\tilde{\boldsymbol{u}}\in V^{k-1}$ and a relational hypergraph $G=(V,E,R,c,\zeta)$ where $\zeta:V^k\mapsto D$ is a $k$-ary coloring that satisfied generalized target node distinguishability, i.e.,

$$\zeta(\tilde{\boldsymbol{u}},u)\neq\zeta(\tilde{\boldsymbol{u}},v)\quad\forall u\in\tilde{\boldsymbol{u}},v\notin\tilde{\boldsymbol{u}},$$
$$\zeta(\tilde{\boldsymbol{u}},u_i)\neq\zeta(\tilde{\boldsymbol{u}},u_j)\quad\forall u_i,u_j\in\tilde{\boldsymbol{u}},u_i\neq u_j.$$

hcwl$_k$ updates $k$-ary coloring $\zeta$ for $\ell \geq 0$:

$$\text{hcwl}_k^{(0)} = \zeta(\tilde{\boldsymbol{u}}, v)$$

$$\text{hcwl}_k^{(\ell+1)}(\tilde{\boldsymbol{u}}, v) = \tau\Big(\text{hcwl}_k^{(\ell)}(\tilde{\boldsymbol{u}}, v), \{\!\!\{\big(\{(\text{hcwl}_k^{(\ell)}(\tilde{\boldsymbol{u}}, w), j)\,|\,(w, j) \in \mathcal{N}_i(e)\}, \rho(e)\big)\,|\,(e, i) \in E(v)\}\!\!\}\Big)$$

Again, we notice that hcwl$_k^{(\ell)}$ computes a valid $k$-ary invariants. We can also show that HC-MPNN restricted on tails prediction, i.e., for each query $\boldsymbol{q} = (q, \tilde{\boldsymbol{u}}, j)$ where $j = k$, is characterized by hcwl$_k$.

**Theorem I.1.** *Let $G = (V, E, R, \boldsymbol{x}, \zeta)$ be a relational hypergraphs where $\boldsymbol{x}$ is a feature map and $\zeta$ is a $k$-ary node coloring satisfying* generalized target nodes distinguishability. *Given a query with $\boldsymbol{q} = (q, \tilde{\boldsymbol{u}}, k)$, then we have that:*

1. *For all* HC-MPNNs *restricted on tails prediction with $L$ layers and initializations* INIT *with* INIT $\equiv \eta$, *and $0 \leq \ell \leq L$, we have* hcwl$_k^{(\ell)} \preceq \boldsymbol{h}_{\boldsymbol{q}}^{(\ell)}$

2. *For all $L \geq 0$, there is an* HC-MPNN *restricted on tails prediction with $L$ layers such that for all $0 \leq \ell \leq L$, we have* hcwl$_k^{(\ell)} \equiv \boldsymbol{h}_{\boldsymbol{q}}^{(\ell)}$.

*Proof.* The proof is very similar to that in Theorem H.1. Note that we sometimes write a $k$-ary tuple $\boldsymbol{v} = (u_1, \cdots, u_k) \in V^k$ by $(\boldsymbol{u}, u_k)$ where $\boldsymbol{u} = (u_1, \cdots, u_{k-1})$ with a slight abuse of notation. We build an auxiliary relational hypergraph $G^k = (V^k, E', R, c_\zeta)$ where $E' = \{r((\tilde{\boldsymbol{u}}, v_1), \cdots, (\tilde{\boldsymbol{u}}, v_m)) \mid r(v_1, \cdots, v_m) \in E, r \in R\}$, and $c_\zeta$ is a node coloring $c_\zeta((\tilde{\boldsymbol{u}}, v)) = \zeta(\tilde{\boldsymbol{u}}, v)$. If $\mathcal{A}$ is a HC-MPNN and $\mathcal{B}$ is an HR-MPNN, we write $\boldsymbol{h}_{\mathcal{A}, G}^{(\ell)}(\tilde{\boldsymbol{u}}, v) := \boldsymbol{h}_{\boldsymbol{q}}^{(\ell)}(v)$ and $\boldsymbol{h}_{\mathcal{B}, G^k}^{(\ell)}((\tilde{\boldsymbol{u}}, v)) := \boldsymbol{h}^{(\ell)}((\tilde{\boldsymbol{u}}, v))$ for the features computed by $\mathcal{A}$ and $\mathcal{B}$ over $G$ and $G^k$, respectively. Again, we write $\mathcal{N}_r^G(e)$ and $E(v)^G$ to emphasize that the positional neighborhood, as well as the hyperedges containing node $v$, is taken over the relational hypergraph $G$, respectively. Finally, we say that an initial feature map $\boldsymbol{y}$ for $G^k$ satisfies generalized target node distinguishability if

$$\boldsymbol{y}((\tilde{\boldsymbol{u}}, u)) \neq \boldsymbol{y}((\tilde{\boldsymbol{u}}, v)) \quad \forall u \in \tilde{\boldsymbol{u}}, v \notin \tilde{\boldsymbol{u}},$$
$$\boldsymbol{y}((\tilde{\boldsymbol{u}}, u_i)) \neq \boldsymbol{y}((\tilde{\boldsymbol{u}}, u_j)) \quad \forall u_i, u_j \in \tilde{\boldsymbol{u}}, u_i \neq u_j.$$

As a result, we have the following equivalence between HR-MPNN and HC-MPNN restricted on tail prediction with the relational hypergraph.

**Proposition I.2.** *Let $G = (V, E, R, \boldsymbol{x}, \zeta)$ be a knowledge graph where $\boldsymbol{x}$ is a feature map, and $\zeta$ is a $k$-ary nodes coloring. Let $q \in R$, then:*

1. *For every HC-MPNN $\mathcal{A}$ with $L$ layers, there is an initial feature map $\boldsymbol{y}$ for $G^k$ an HR-MPNN $\mathcal{B}$ with $L$ layers such that for all $0 \leq \ell \leq L$ and $u, v \in V$, we have $\boldsymbol{h}_{\mathcal{A}, G}^{(\ell)}(\tilde{\boldsymbol{u}}, v) = \boldsymbol{h}_{\mathcal{B}, G^2}^{(\ell)}((\tilde{\boldsymbol{u}}, v))$.*

2. *For every initial feature map $\boldsymbol{y}$ for $G^k$ satisfying generalized target node distinguishability and every HR-MPNN $\mathcal{B}$ with $L$ layers, there is a HC-MPNN $\mathcal{A}$ with $L$ layers such that for all $0 \leq \ell \leq L$ and $(\tilde{v}u, v) \in V^k$, we have $\boldsymbol{h}_{\mathcal{A}, G}^{(\ell)}(\tilde{\boldsymbol{u}}, v) = \boldsymbol{h}_{\mathcal{B}, G^k}^{(\ell)}((\tilde{\boldsymbol{u}}, v))$.*

*Proof.* We first show item (1). Consider the HR-MPNN $\mathcal{B}$ with the same relational-specific message MSG$_r$, aggregation AGG, and update functions UP as $\mathcal{A}$ for all the $L$ layers. The initial feature map $\boldsymbol{y}$ is defined as $\boldsymbol{y}((\tilde{\boldsymbol{u}}, v)) = \text{INIT}(v, \boldsymbol{q})$, where INIT is the initialization function of $\mathcal{A}$. Then, by induction on number of layer $\ell$, we have that for the base case $\ell = 0$, $\boldsymbol{h}_{\mathcal{A}}^{(0)}(\tilde{\boldsymbol{u}}, v) = \text{INIT}(v, \boldsymbol{q}) = \boldsymbol{y}((\tilde{\boldsymbol{u}}, v)) = \boldsymbol{h}_{\mathcal{B}}^{(0)}((\tilde{\boldsymbol{u}}, v))$.

For the inductive case, assume $\boldsymbol{h}_{\mathcal{A}}^{(\ell)}(\tilde{\boldsymbol{u}}, v) = \boldsymbol{h}_{\mathcal{B}}^{(\ell)}((\tilde{\boldsymbol{u}}, v))$, then

$$
\begin{aligned}
\boldsymbol{h}_{\mathcal{A}}^{(\ell+1)}(\tilde{\boldsymbol{u}}, v) = \text{UP}\Big( & \boldsymbol{h}_{\mathcal{A}}^{(\ell)}(\tilde{\boldsymbol{u}}, v), \text{AGG}\big(\boldsymbol{h}_{\mathcal{A}}^{(\ell)}(\tilde{\boldsymbol{u}}, v), \\
& \{\!\!\{ \text{MSG}_{\rho(e)}\Big(\{(\boldsymbol{h}_{\mathcal{A}}^{(\ell)}(\tilde{\boldsymbol{u}}, w), j) \mid (w, j) \in \mathcal{N}_i^G(e)\}\Big) \mid (e, i) \in E^G(v)\}\!\!\}\big)\Big) \\
= \text{UP}\Big( & \boldsymbol{h}_{\mathcal{B}}^{(\ell)}((\tilde{\boldsymbol{u}}, v)), \text{AGG}\big(\boldsymbol{h}_{\mathcal{B}}^{(\ell)}((\tilde{\boldsymbol{u}}, v)), \\
& \{\!\!\{ \text{MSG}_{\rho(e)}\Big(\{(\boldsymbol{h}_{\mathcal{B}}^{(\ell)}((\tilde{\boldsymbol{u}}, w)), j) \mid (w, j) \in \mathcal{N}_i^{G^k}(e)\}\Big) \mid (e, i) \in E^{G^k}(v)\}\!\!\}\big)\Big) \\
= & \boldsymbol{h}_{\mathcal{B}}^{(\ell+1)}((\tilde{\boldsymbol{u}}, v)).
\end{aligned}
$$

To show item (2), we consider $\mathcal{A}$ with the same relational-specific message $\text{MSG}_r$, aggregation $\text{AGG}$, and update functions $\text{UP}$ as $\mathcal{B}$ for all the $L$ layers. We also take initialization function $\text{INIT}$ such that $\text{INIT}(v, \boldsymbol{q}) = \boldsymbol{y}((\tilde{\boldsymbol{u}}, v))$. Then, we can follow the same argument for the equivalence as item (1). $\qquad\square$

Similarly, we can show the equivalence in terms of the relational WL algorithms with $\text{hcwl}_k$:

**Proposition I.3.** *Let $G = (V, E, R, c, \zeta)$ be a relational hypergraph where $\zeta$ is a $k$-ary node coloring. For all $\ell \geq 0$ and $(\tilde{\boldsymbol{u}}, v) \in V^k$, we have that $\text{hcwl}_k^{(\ell)}(\tilde{\boldsymbol{u}}, v)$ computed over $G$ coincides with $\text{hrwl}_1^{(\ell)}((\tilde{\boldsymbol{u}}, v))$ computed over $G^k = (V^k, E', R, c_\zeta)$.*

*Proof.* For $\ell = 0$, we have $\text{hcwl}_k^{(0)}(G, \tilde{\boldsymbol{u}}, v) = \zeta(\tilde{\boldsymbol{u}}, v) = c_\zeta((\tilde{\boldsymbol{u}}, v)) = \text{hrwl}_1^{(0)}(G^k, (\tilde{\boldsymbol{u}}, v))$.

For the inductive case, we have that

$$
\begin{aligned}
\text{hcwl}_k^{(\ell+1)}(G, \tilde{\boldsymbol{u}}, v) = \tau\Big( & \text{hcwl}_k^{(\ell)}(G, \tilde{\boldsymbol{u}}, v), \\
& \{\!\!\{ \big(\{(\text{hcwl}_2^{(\ell)}(G, \tilde{\boldsymbol{u}}, w), j) \mid (w, j) \in \mathcal{N}_i^G(e)\}, \rho(e)\big) \mid (e, i) \in E^G(v)\}\!\!\}\Big) \\
= \tau\Big( & \text{hrwl}_1^{(\ell)}(G^k, (\tilde{\boldsymbol{u}}, v)), \\
& \{\!\!\{ \big(\{(\text{hrwl}_1^{(\ell)}(G^k, (\tilde{\boldsymbol{u}}, w)), j) \mid (w, j) \in \mathcal{N}_i^{G^k}(e)\}, \rho(e)\big) \mid (e, i) \in E^{G^k}(v)\}\!\!\}\Big) \\
= & \text{hrwl}_1^{(\ell+1)}(G^k, (\tilde{\boldsymbol{u}}, v)).
\end{aligned}
$$

$\qquad\square$

Now we are ready to show the proof for Theorem I.1. For a relational hypergraph $G = (V, E, R, \boldsymbol{x}, \zeta)$, we consider $G^k = (V^k, E', R, c_\zeta)$ as defined earlier. We start with item (1). Let $\mathcal{A}$ be a HC-MPNN with $L$ layers and initialization $\text{INIT}$ satisfying $\text{INIT} \equiv \zeta$ and let $0 \leq \ell \leq L$. Let $\boldsymbol{y}$ be an initial feature map for $G^k$ and $\mathcal{B}$ be an HR-MPNN with $L$ layers in Proposition I.2, item (1). For the initialization we have $\boldsymbol{y} \equiv c_\zeta$ since $\boldsymbol{y}((\tilde{\boldsymbol{u}}, v)) = \text{INIT}(v, \boldsymbol{q})$. Thus, we can proceed and apply Theorem 4.1, item (1) to $G^k$, $\boldsymbol{y}$, and $\mathcal{B}$ and show that $\text{hrwl}_1^{(\ell)} \preceq \boldsymbol{h}_{\mathcal{B}, G^k}^{(\ell)}$, which in turns shows that $\text{hcwl}_k^{(\ell)} \preceq \boldsymbol{h}_{\mathcal{A}, G}^{(\ell)}$.

We then proceed to show item (2). Let $L \geq 0$ be an integer representing a total number of layers. We apply Theorem 4.1, item (2) to $G^k$ and obtain an initial feature map $\boldsymbol{y}$ with $\boldsymbol{y} \equiv c_\zeta$ and an HR-MPNN $\mathcal{B}$ with $L$ layer such that $\text{hrwl}_1^{(\ell)} \equiv \boldsymbol{h}_{\mathcal{B}, G^k}^{(\ell)}$ for all $0 \leq \ell \leq L$. We stress again that $\boldsymbol{y}$ and $\zeta$ both satisfy generalized target node distinguishability. Now, let $\mathcal{A}$ be the HC-MPNN from *Proposition I.2*, item (2). Thus, $\text{hcwl}_k^{(\ell)} \equiv \boldsymbol{h}_{\mathcal{A}, G}^{(\ell)}$ as required. Again, we note that the item (2) holds for HCNet.

$\qquad\square$

Table 5: Model asymptotic runtime complexities.

| Model | Complexity of a forward pass | Amortized complexity of a query |
|---|---|---|
| HR-MPNNs | $\mathcal{O}(L(m\|E\|d + \|V\|d^2))$ | $\mathcal{O}(L(\frac{m\|E\|d}{\|R\|\|V\|^2} + \frac{d^2}{\|R\|\|V\|} + d))$ |
| HC-MPNNs | $\mathcal{O}(L(m\|E\|d + \|V\|d^2))$ | $\mathcal{O}(L(\frac{m\|E\|d}{\|V\|} + d^2))$ |

## J  COMPLEXITY ANALYSIS

In this section, we discuss the asymptotic time complexity of HR-MPNN and HC-MPNN. For HC-MPNN, we consider the model instance of HCNet with $g_r^{(\ell)}$ being a *query-independent* diagonal linear map. For HR-MPNN, we consider the model instance with the same updating function UP and relation-specific message function MSG$_r$ as the considered HCNet model instance, referred to as HRNet:

$$\boldsymbol{h}_v^{(0)} = \mathbf{1}^d$$

$$\boldsymbol{h}_v^{(\ell+1)} = \sigma\Big(\boldsymbol{W}^{(\ell)}\Big[\boldsymbol{h}_v^{(\ell)}\Big\| \sum_{(e,i)\in E(v)} \Big(\odot_{j\neq i}\,(\alpha^{(\ell)}\boldsymbol{h}_{e(j)}^{(\ell)} + (1-\alpha^{(\ell)})\boldsymbol{p}_j)\odot\boldsymbol{w}_r^{(\ell)}\Big)\Big] + \boldsymbol{b}^{(\ell)}\Big).$$

**Notation.** Given a relational hypergraph $G = (V, E, R, c)$, we denote $|V|, |E|, |R|$ to be the size of vertices, edges, and relation types. $d$ is the hidden dimension and $m$ is the maximum arity of the edges. Additionally, we denote $L$ to be the total number of layers, and $k$ to be the arity of the query relation $q \in R$ in the query $\boldsymbol{q} = (q, \tilde{\boldsymbol{u}}, t)$.

**Analysis.** Given a query $\boldsymbol{q} = (q, \tilde{\boldsymbol{u}}, t)$, the runtime complexity of a single forward pass of HCNet is $O(L(m|E|d + |V|d^2))$ since for each message, we need $O(d)$ for the relation-specific transformation, and we have $m|E|$ total amount of message in each layer. During the updating function, we additionally need a linear transformation for each aggregated message as well as a self-transformation, which costs $O(d^2)$ for each node. Adding them up, we have $O(m|E|d + |V|d^2)$ cost for each layer, and thus $O(L(m|E|d + |V|d^2))$ in total.

Note that this is the same as the complexity of HRNet since the only differences lie in initialization methods, which is $O(|V|d)$ cost for HCNet. In terms of computing a single query, the amortized complexity of HCNet is $O(L(\frac{m|E|d}{|V|} + d^2))$ since in each forward pass, $|V|$ number of queries are computed at the same time. In contrast, HRNet computes $|V|^k$ query as once it has representations for all nodes in the relational hypergraph, it can compute all possible hyperedges by permuting the nodes and feeding them into the $k$-ary decoder. We summarize the complexity analysis in Table 5.

## K  EXPERIMENTS ON INDUCTIVE LINK PREDICTION WITH KNOWLEDGE GRAPHS

We carry out additional inductive experiments on knowledge graphs where each edge has its arity fixed to 2 and compare the results against the current state-of-the-art models.

**Setup.** We evaluate HCNet on 4 standard inductive splits of WN18RR (Bordes et al., 2013) and FB15k-237 (Dettmers et al., 2018), which was proposed in Teru et al. (2020). We provide the details of the datasets in Table 7. Contrary to the standard experiment setting (Zhu et al., 2021; 2023) on knowledge graph $G = (V, E, R, \boldsymbol{x})$ where for each relation $r(u, v) \in E$, an inverse-relation $r^{-1}$ is introduced as a fresh relation symbol and $r^{-1}(v, u)$ is added in the knowledge graph, in our setup we do *not* augment inverse edges for HCNet. This makes the task more challenging. We compare HCNet with models designed *only* for inductive binary link prediction task with knowledge graphs, namely GraIL (Teru et al., 2020), NeuralLP (Yang et al., 2017), DRUM (Sadeghian et al., 2019), NBFNet (Zhu et al., 2021), RED-GNN (Zhang & Yao, 2022), and A*Net (Zhu et al., 2023), and we take the results provided in Zhu et al. (2023) for comparison.

**Implementation.** We report the hyperparamter used in Table 8. For all models, we consider a 2-layer MLP as decoder and adopt layer-normalization with dropout in all layers before applying

Table 6: Binary inductive experiment on knowledge graph for Hits@10 result. The best result is in **bold**, and second/third best in underline.

| Method | FB15k-237 | | | | WN18RR | | | |
|---|---|---|---|---|---|---|---|---|
| | v1 | v2 | v3 | v4 | v1 | v2 | v3 | v4 |
| GraIL | 0.429 | 0.424 | 0.424 | 0.389 | 0.760 | 0.776 | 0.409 | 0.687 |
| NeuralLP | 0.468 | 0.586 | 0.571 | 0.593 | 0.772 | 0.749 | 0.476 | 0.706 |
| DRUM | 0.474 | 0.595 | 0.571 | 0.593 | 0.777 | 0.747 | 0.477 | 0.702 |
| NBFNet | 0.574 | **0.685** | **0.637** | 0.627 | **0.826** | 0.798 | **0.568** | 0.694 |
| RED-GNN | 0.483 | 0.629 | 0.603 | 0.621 | 0.799 | 0.780 | 0.524 | 0.721 |
| A*Net | **0.589** | 0.672 | 0.629 | **0.645** | 0.810 | **0.803** | 0.544 | **0.743** |
| HCNet | 0.566 | 0.646 | 0.614 | 0.610 | 0.822 | 0.790 | 0.536 | 0.724 |

Table 7: Dataset statistics for the inductive relation prediction experiments. **#Query\*** is the number of queries used in the validation set. In the training set, all triplets are used as queries.

| Dataset | | #Relation | Train & Validation | | | Test | | |
|---|---|---|---|---|---|---|---|---|
| | | | #Nodes | #Triplet | #Query* | #Nodes | #Triplet | #Query |
| WN18RR | $v_1$ | 9 | 2,746 | 5,410 | 630 | 922 | 1,618 | 188 |
| | $v_2$ | 10 | 6,954 | 15,262 | 1,838 | 2,757 | 4,011 | 441 |
| | $v_3$ | 11 | 12,078 | 25,901 | 3,097 | 5,084 | 6,327 | 605 |
| | $v_4$ | 9 | 3,861 | 7,940 | 934 | 7,084 | 12,334 | 1,429 |
| FB15k-237 | $v_1$ | 180 | 1,594 | 4,245 | 489 | 1,093 | 1,993 | 205 |
| | $v_2$ | 200 | 2,608 | 9,739 | 1,166 | 1,660 | 4,145 | 478 |
| | $v_3$ | 215 | 3,668 | 17,986 | 2,194 | 2,501 | 7,406 | 865 |
| | $v_4$ | 219 | 4,707 | 27,203 | 3,352 | 3,051 | 11,714 | 1,424 |

ReLU activation and skip-connection. We also adopt the sinusoidal positional encoding as described in the body of the paper. We discard all the edges in the training graph that are currently being treated as positive triplets in each batch to prevent overfitting. We additionally pass in the considered query representation $z_q$ to the decoder via concatenation to $h_{v|q}^{(L)}$. The best checkpoint for each model is selected based on its performance on the validation sets, and all experiments are performed on one NVIDIA A10 24GB GPU. For evaluation, we consider *filtered ranking protocol* (Bordes et al., 2013) with 32 negative samples per positive triplet, and report Hits@10 for each model.

**Results.** We report the results in Table 6. We observe that HCNets are highly competitive even compared with state-of-the-art models specifically designed for link prediction with knowledge graphs. HCNets reach the top 3 for 7 out of 8 datasets, and obtain a very close result for the final dataset. Note here that the top 2 models are NBFNet (Zhu et al., 2021) and A*Net (Zhu et al., 2023), which share a similar idea of HCNet and are all based on conditional message passing. The difference in results lies in the different message functions, which are further supported in Table 1 of Huang et al. (2023).

However, we highlight that HCNet does *not* augment with inverse relation edges, as described in the set-up of the experiment. HCNet can recognize the directionality of relational edges and pay respect to both incoming and outgoing edges during message passing. No current link prediction model based on message passing can explicitly take care of this without edge augmentation. In fact, Theorem H.4 implies that all current models based on conditional message passing, including NBFNets, need inverse relation augmentation to match the expressive power of HCNet. Theoretically speaking, this allows us to claim that HCNet is strictly more powerful than all other models in the baseline that are based on conditional message passing, assuming all considered models are expressive enough to match their corresponding relational Weisfeiler-Leman test.

Table 8: Hyperparameters for binary inductive experiments with HCNet.

| Hyperparameter | | WN18RR | FB15k-237 |
|---|---|---|---|
| **GNN Layer** | Depth($L$) | 6 | 6 |
| | Hidden Dimension | 32 | 32 |
| **Decoder Layer** | Depth | 2 | 2 |
| | Hidden Dimension | 64 | 64 |
| **Optimization** | Optimizer | Adam | Adam |
| | Learning Rate | 5e-3 | 5e-3 |
| **Learning** | Batch size | 32 | 32 |
| | #Negative Samples | 32 | 32 |
| | Epoch | 30 | 30 |
| | #Batch Per Epoch | — | — |
| | Adversarial Temperature | 0.5 | 0.5 |
| | Dropout | 0.2 | 0.2 |
| | Accumulation Iteration | 1 | 1 |

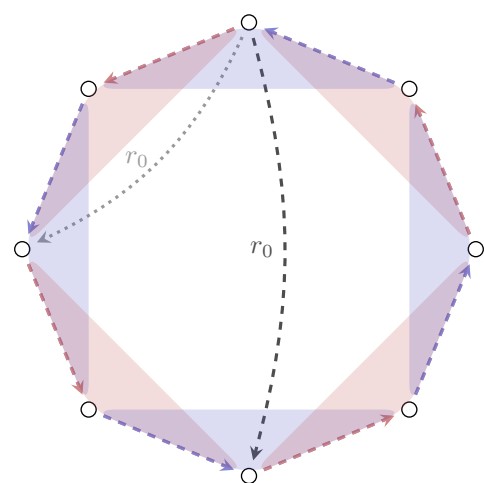

Figure 5: An example relational hypergraph of dataset HyperCycle, where $n = 8$ and $k = 3$. We colored $r_1$ as blue and $r_2$ as red. The goal is to predict the black edge as true but the gray edge as false. We claim that no HR-MPNNs can correctly solve this task, but HCNets can.

## L    EXPERIMENTS ON HYPER-RELATIONAL KNOWLEDGE GRAPHS

**Dataset & Baselines.** We conduct experiments on hyper-relational knowledge graphs, namely publicly available JF17K (Wen et al., 2016), while we note that there are some critical issues of this dataset such as redundant entries[1] and severe test leakages (Galkin et al., 2020), we still include this as it is one of the most common hyper-relational knowledge graphs datasets in the literature. To adapt HCNets and other methods only applicable on relational hypergraphs, we transformed JF17K by the conversion described in Appendix B. For baselines, we have taken the experiment results from r-SimplE, m-DistMult, m-CP, m-TransH from Fatemi et al. (2020), RAE from Zhang et al. (2018), NaLP from Guan et al. (2019), tNaLP+ from Guan et al. (2021), HINGE from Rosso et al. (2020), NeuInfer from Guan et al. (2020), RAM from Liu et al. (2021b), S2S from Di et al. (2021), and GNN method RD-MPNN from Zhou et al. (2023), and StarE (Galkin et al., 2020). We report the statistics of the datasets transformed into relational hypergraphs in Table 12, and the hyper-parameter in Table 14.

---

[1]https://www.site.uottawa.ca/ yymao/JF17K/

Table 9: Results of transductive link prediction experiments on JF17K.

| | JF17K | | |
| | MRR | Hits@1 | Hits@10 |
|---|---|---|---|
| r-SimplE | 0.102 | 0.069 | 0.168 |
| m-DistMult | 0.463 | 0.372 | 0.634 |
| m-CP | 0.391 | 0.298 | 0.563 |
| m-TransH | 0.444 | 0.370 | 0.581 |
| RAE | 0.392 | 0.312 | 0.561 |
| NaLP | 0.310 | 0.239 | 0.450 |
| tNaLP+ | 0.449 | 0.370 | 0.598 |
| HINGE | 0.517 | 0.436 | 0.675 |
| NeuInfer | 0.451 | 0.373 | 0.604 |
| SAS | 0.528 | 0.457 | 0.690 |
| RAM | 0.539 | 0.463 | 0.690 |
| G-MPNN | 0.501 | 0.425 | 0.660 |
| RD-MPNN | 0.512 | 0.445 | 0.685 |
| StarE | **0.542** | **0.454** | 0.685 |
| HCNet | 0.540 | 0.440 | **0.730** |

**Evaluation.** We report the MRR, Hits@1 and Hits@10 for all considered model. However, we highlight the differences in evaluation: in the evaluation of hyper-relational knowledge graphs, only the head and tail entities in the main triplet are corrupted. In our setup, we follow the evaluation convention of relational hypergraphs and corrupt all positions.

**Results.** The results of HCNet are better than existing baselines, including models designed for relational hypergraphs and models designed for hyper-relational knowledge graphs. In particular, HCNet marginally outperforms StarE according to Hits@10. StarE is one of the state-of-the-art models on link prediction with hyper-relational knowledge graphs, but StarE is a transductive method and is inherently limited to transductive datasets, whereas HCNets do not have such limitations. In this sense, HCNets also lift the capabilities of methods designed for hyper-relational knowledge graphs to the inductive setup, which is substantially more challenging.

## M  SYNTHETIC EXPERIMENTS

We carry out a synthetic experiment with a custom-built dataset HyperCycle to showcase that HC-MPNNs are more expressive than HR-MPNNs in the task of link prediction with relational hypergraphs.

**Dataset.** We construct HyperCycle, a synthetic dataset that consists of multiple relational hypergraphs with relation $R = \{r_0, r_1, r_2\}$. Each relational hypergraph $G$ is parameterized by 2 hyperparameters: the number of nodes $n$ which is always a multiple of $4$, and the arity of each edge $k$. Given such $(n, k)$ pair, we generate the relational hypergraph $G(n,k) = (V(n,k), E(n,k), R(n,k))$ where

$$V(n,k) = \{x_1, \cdots, x_n\}$$
$$E(n,k) = \{r_{(i \bmod 2)+1}(x_{(i+j) \bmod n} \mid 0 \le j < k) \mid 1 \le i \le n\}$$
$$R(n,k) = \{r_0, r_1, r_2\}$$

In short, there is a directed hyper-edge of arity $k$ with alternating relations between $r_1$ and $r_2$ for all $k$ consecutive nodes in this cycle. We present one example of such relational hypergraph in Figure 5, where $n = 8$ and $k = 3$. We generate the dataset by choosing $n = \{8, 12, 16, 20\}$ and $k = \{3, 4, 5, 6, 7\}$. We then randomly pick 70% of the generated graphs as the training set and the remaining 30% as the testing set.

**Objective.** The objective of this task is for each node to identify the node that is located at the "opposite point" in the cycle of the given node as true. Formally speaking, for a relational hypergraph $G(n,k)$, we want to predict a 2-ary (hyper-)edge of relation $r_0$ between any node $x_i$ and its "opposite point" $x_{(i+n/2 \bmod n)}$ for all $1 \le i \le n$, i.e., classify $r_0(x_i, x_{(i+n/2) \bmod n})$ as true. The

negative sample is generated by considering the $r_0$ relation (hyper-)edges that connect the "2-hop" neighboring node, i.e., classify $r_0(x_i, x_{(i+2) \bmod n})$ as false. Note that since $n \neq 4$, we will never have $(i + n/2) \bmod n = (i + 2) \bmod n$.

**Model architectures.** We considered two model architectures, namely an HC-MPNN instance HCNet:

$$\boldsymbol{h}_{v|\boldsymbol{q}}^{(0)} = \sum_{i \neq t} \mathbb{1}_{v=u_i} * (\boldsymbol{p}_i + \boldsymbol{z}_q)$$

$$\boldsymbol{h}_{v|\boldsymbol{q}}^{(\ell+1)} = \sigma\Big(\boldsymbol{W}^{(\ell)}\Big[\boldsymbol{h}_{v|\boldsymbol{q}}^{(\ell)}\Big\| \sum_{(e,i)\in E(v)} \Big(\odot_{j\neq i}\,(\alpha^{(\ell)}\boldsymbol{h}_{e(j)|\boldsymbol{q}}^{(\ell)} + (1-\alpha^{(\ell)})\boldsymbol{p}_j)\odot\boldsymbol{w}_r^{(\ell)}\Big)\Big] + \boldsymbol{b}^{(\ell)}\Big).$$

and a corresponding HR-MPNNs instance called HRNet that shares the same *update*, *aggregate*, and relation-specific *message* functions as in HCNet, defined as follow:

$$\boldsymbol{h}_v^{(0)} = \mathbf{1}^d$$

$$\boldsymbol{h}_v^{(\ell+1)} = \sigma\Big(\boldsymbol{W}^{(\ell)}\Big[\boldsymbol{h}_v^{(\ell)}\Big\| \sum_{(e,i)\in E(v)} \Big(\odot_{j\neq i}\,(\alpha^{(\ell)}\boldsymbol{h}_{e(j)}^{(\ell)} + (1-\alpha^{(\ell)})\boldsymbol{p}_j)\odot\boldsymbol{w}_r^{(\ell)}\Big)\Big] + \boldsymbol{b}^{(\ell)}\Big).$$

Note that $\sigma$ stands for the ReLU activation function in both models. We additionally use a binary MLP decoder for HRNet, which takes the concatenation of the final representation for each entity in the query, together with the learnable query vector $\boldsymbol{z}_q$ to obtain the final probability.

**Design.** We claim that HCNet can correctly predict all the testing triplets, whereas HRNet fails to learn this pattern and will only achieve 50% accuracy, which is no better than random guessing. This is exactly due to the lack of expressiveness of HR-MPNNs by relying on a $k$-ary decoder for link prediction. Theoretically, all nodes of the relational hypergraphs in HyperCycle, due to their rotational symmetry introduced by alternating relation types $r_1, r_2$, can be partitioned into two sets. Since the nodes within each set are isomorphic to each other, it is impossible for any HR-MPNNs to distinguish between these nodes by only computing its unary invariant. Thus HR-MPNNs cannot possibly solve this task, as whenever they classify the target "opposite point" node to be true, they also have to classify the "2-hop" node to be true, and vice versa.

However, HCNet can bridge this gap by introducing the relevant notion of "distance". As HCNet carries out message-passing after identifying the source node, the relative distance between the source node and the target "opposite point" node will be different than the one with the "2-hop" node. Thus, by keeping track of the distance from the source node, HCNet will compute a different embedding for the positive triplet and the negative triplet, effectively solving this task.

**Experimental details.** For both models, we use 7 layers, each with 32 hidden dimensions. We configure the learning rate to be 1e-3 for both models and train them for 100 epochs. Empirically, we observe that HCNet easily reaches 100% accuracy, solving this task completely, whereas HRNet always fails to learn anything meaningful, reaching an accuracy of 50%. The experiment results are consistent with our theory.

## N    SCALABILITY AND CUSTOM TRITON KERNEL

Scalability is generally a concern for inductive link prediction since link prediction between a given pair of nodes relies heavily on the structural properties of these nodes (due to the lack of node features) which necessitates strong encoders that go beyond the power of 1-WL. This is more dramatic for relational hypergraphs since the prediction now relies on the structural properties of $k$ nodes and any model will suffer from scalability issues if $k$ becomes large. With that being said, our approach remains feasible for the benchmark datasets, but we think it is important for future work to scale up these models for larger datasets, much like it has been done for classical GNNs (Hamilton et al., 2017; Zhu et al., 2023).

To resolve this empirically, we have included custom implementation via Triton kernel [2] in our codebase to account for the message passing process on relational hypergraphs, which on average halved the training times and dramatically reduced the space usage of the algorithm (5 times reduction on

---

[2]https://github.com/triton-lang/triton

Table 10: Average degree of relational hypergraphs in the experiments.

|  | WP-IND | JF-IND | MFB-IND | FB-AUTO | WikiPeople |
|---|---|---|---|---|---|
| Average degree | 1.03 | 1.36 | 104.5 | 2.16 | 6.06 |

average). The idea is to not materialize all the messages explicitly as in PyTorch geometric (Fey & Lenssen, 2019), but directly write the neighboring features into the corresponding memory addresses. Compared with materializing all hyperedge messages which takes $O(k|E|)$ where $k$ is the maximum arity, computing with Triton kernel only is $O(|V|)$ in memory. This will enable fast and scalable message passing on relational hypergraphs, both on HR-MPNNs and HC-MPNNs.

## O    ON ADDING NODE FEATURES

On the surface, it seems that HC-MPNNs does not directly take node features into account. This is because in the task of link prediction on relational hypergraphs, no node features are explicitly provided to begin with, and thus we did not assume the presence of node features in this particular task setting. However, it is relatively straightforward to account for node features by simply concatenating the node feature $\boldsymbol{x}_v$ on top of the current representation $\boldsymbol{h}_v$ to obtain $\boldsymbol{h}_v^* = [\boldsymbol{h}_v \| \boldsymbol{x}_v]$. Indeed, the only requirement for HC-MPNNs in the initialization is to satisfy generalized target node distinguishability, and thus concatenating node features will preserve this property. As a result, all theoretical results can be directly applied to HC-MPNNs with node features. It is worth noting that this concatenating technique has already been applied in Zhang et al. (2021) on knowledge graphs with node features and has proven to be successful. Additionally, this technique is also mentioned in Galkin et al. (2024) for link prediction with knowledge graphs using conditional message passing.

## P    IMPACT ON THE DENSITY OF THE RELATIONAL HYPERGRAPHS

To further analyze the impact on the structure and density, we present the average degree of (training) datasets in Table 10. Observe that even though MFB-IND is a very dense hypergraph, HCNets can still manage to double the metrics compared to existing models. Furthermore, we highlight the performance of HCNets in sparse hypergraph settings, which are more representative of many real-world scenarios. Remarkably, HCNets maintain competitive performance even under these challenging conditions, underscoring their adaptability and effectiveness across a wide range of graph density regimes. These findings highlight the versatility of HCNets in handling diverse hypergraph structures.

## Q    FURTHER EXPERIMENT DETAILS

We report the details of the experiment carried out in the body of the paper in this section. In particular, we report the dataset statistics of the inductive link prediction task in Table 11 and of the transductive link prediction task in Table 12. We also report the hyperparameter used for HCNet in the inductive link prediction task at Table 13 and transductive link prediction task at Table 14, respectively.

In addition, we report additionally the standard deviation for the main experiments in Table 15 and Table 18 along with extra baseline results on FB-AUTO, namely, r-SimplE, m-DistMult, m-CP, m-TransH, HSimplE from Fatemi et al. (2020) and GETD from Liu et al. (2020). We also show the complete tables for the ablation study mentioned in Table 16 and Table 17, the detailed definitions of initialization and positional encoding considered in Table 19 and Table 20, respectively.

Finally, we report the execution time and GPU usages for 1 epochs of HCNets on all datasets considered in the paper with corresponding hyperparameters in Table 21. See further discussion of scalability in Appendix N. For the RD-MPNNs training, we consider a learning rate of 0.1, a dimension of 200, and 10 negative samples for training on all inductive datasets. In the experiments, all

Table 11: Dataset statistics of inductive link prediction task with relational hypergraph.

| Dataset | # seen vertices | # train hyperedges | # unseen vertices | # relations | # features | # max arity |
|---|---|---|---|---|---|---|
| WP-IND | 4,363 | 4,139 | 100 | 32 | 37 | 4 |
| JF-IND | 4,685 | 6,167 | 100 | 31 | 46 | 4 |
| MFB-IND | 3,283 | 336,733 | 500 | 12 | 25 | 3 |

relational hypergraphs do not contain node features. We present a detailed discussion and strategy in Appendix O for HC-MPNNs to be applied on relational hypergraphs with node features.

We adopt the *partial completeness assumption* (Galárraga et al., 2013) on relational hypergraphs, where we randomly corrupt the $t$-th position of a $k$-ary fact $q(u_1, \cdots, u_k)$ each time for $1 \leq t \leq k$. HCNets minimize the negative log-likelihood of the positive fact presented in the training graph, and the negative facts due to corruption. We represent query $\boldsymbol{q} = (q, \tilde{\boldsymbol{u}}, t)$ as the fact $q(u_1, \cdots, u_k)$ given corrupting $t$-th position, and represent its conditional probability as $p(v|\boldsymbol{q}) = \sigma(f(\boldsymbol{h}_{v|\boldsymbol{q}}^{(L)}))$, where $v \in V$ is the considered entity in the $t$-th position, $L$ is the total number of layer, $\sigma$ is the sigmoid function, and $f$ is a 2-layer MLP. We then adopt *self-adversarial negative sampling* (Sun et al., 2019) by sampling negative triples from the following distribution:

$$\mathcal{L}(v \mid \boldsymbol{q}) = -\log p(v \mid \boldsymbol{q}) - \sum_{i=1}^{n} w_{i,\alpha} \log(1 - p(v_i' \mid \boldsymbol{q}))$$

where $\alpha$ is the adversarial temperature as part of the hyperparameter, $n$ is the number of negative samples for the positive sample and $v_i'$ is the $i$-th corrupted vertex of the negative sample. Finally, $w_i$ is the weight for the $i$-th negative sample, given by

$$w_{i,\alpha} := \text{Softmax}\left(\frac{\log(1 - p(v_i' \mid \boldsymbol{q}))}{\alpha}\right).$$

Table 12: Dataset statistics of transductive link prediction task with relational hypergraph on FB-AUTO and WikiPeople with respective arity.

| Dataset | FB-AUTO | WikiPeople | JF17K |
|---|---|---|---|
| $|V|$ | 3,410 | 47,765 | 29,177 |
| $|R|$ | 8 | 707 | 327 |
| #train | 6,778 | 305,725 | 61,104 |
| #valid | 2,255 | 38,223 | 15,275 |
| #test | 2,180 | 38,281 | 24,915 |
| # arity= 2 | 3,786 | 337,914 | 56,322 |
| # arity= 3 | 0 | 25,820 | 34,550 |
| # arity= 4 | 215 | 15,188 | 9,509 |
| # arity≥ 5 | 7,212 | 3,307 | 2,267 |

## R    SOCIAL IMPACT

This work mainly focused on link prediction with relational hypergraphs, which has a wide range of applications and thus many potential societal impacts. One potential negative impact is the enhancement of malicious network activities like phishing or pharming through the use of powerful link prediction models. We encourage further studies to mitigate these issues.

Table 13: Hyperparameters for inductive experiments of HCNet.

| Hyperparameter | | WP-IND | JF-IND | MFB-IND |
|---|---|---|---|---|
| **GNN Layer** | Depth($L$) | 5 | 5 | 4 |
| | Hidden Dimension | 128 | 256 | 32 |
| **Decoder Layer** | Depth | 2 | 2 | 2 |
| | Hidden Dimension | 128 | 256 | 32 |
| **Optimization** | Optimizer | Adam | Adam | Adam |
| | Learning Rate | 5e-3 | 1e-2 | 5e-3 |
| **Learning** | Batch size | 32 | 32 | 1 |
| | #Negative Sample | 10 | 10 | 10 |
| | Epoch | 20 | 20 | 10 |
| | #Batch Per Epoch | - | - | 10000 |
| | Adversarial Temperature | 0.5 | 0.5 | 0.5 |
| | Dropout | 0.2 | 0.2 | 0 |
| | Accumulation Iteration | 1 | 1 | 32 |

Table 14: Hyperparameters for transductive experiments of HCNet.

| Hyperparameter | | FB-AUTO | WikiPeople | JF17K |
|---|---|---|---|---|
| **GNN Layer** | Depth($L$) | 4 | 5 | 6 |
| | Hidden Dimension | 128 | 64 | 64 |
| **Decoder Layer** | Depth | 2 | 2 | 2 |
| | Hidden Dimension | 128 | 64 | 64 |
| **Optimization** | Optimizer | Adam | Adam | Adam |
| | Learning Rate | 1e-3 | 1e-3 | 5e-3 |
| **Learning** | Batch size | 32 | 16 | 1 |
| | #Negative Sample | 32 | 32 | 50 |
| | Epoch | 20 | 6 | 6 |
| | #Batch Per Epoch | — | 5000 | — |
| | Adversarial Temperature | 0.5 | 0.5 | 0.5 |
| | Dropout | 0.2 | 0.2 | 0.2 |
| | Accumulation Iteration | 1 | 1 | 32 |

Table 15: Results of inductive link prediction experiments. We report averaged MRR, Hits@1, and Hits@3 (higher is better) on test sets together with its standard deviation.

| | WP-IND | | | JF-IND | | | MFB-IND | | |
|---|---|---|---|---|---|---|---|---|---|
| | MRR | Hits@1 | Hits@3 | MRR | Hits@1 | Hits@3 | MRR | Hits@1 | Hits@3 |
| HGNN | 0.072 | 0.045 | 0.112 | 0.102 | 0.086 | 0.128 | 0.121 | 0.076 | 0.114 |
| HyperGCN | 0.075 | 0.049 | 0.111 | 0.099 | 0.088 | 0.133 | 0.118 | 0.074 | 0.117 |
| G-MPNN-sum | 0.177 | 0.108 | 0.191 | 0.219 | 0.155 | 0.236 | 0.124 | 0.071 | 0.123 |
| G-MPNN-mean | 0.153 | 0.096 | 0.145 | 0.112 | 0.039 | 0.116 | 0.241 | 0.162 | 0.257 |
| G-MPNN-max | 0.200 | 0.125 | 0.214 | 0.216 | 0.147 | 0.240 | 0.268 | 0.191 | 0.283 |
| RD-MPNN | 0.304 | 0.238 | 0.328 | 0.402 | 0.308 | 0.453 | 0.122 | 0.082 | 0.125 |
| HCNet | **0.414** | **0.352** | **0.451** | **0.435** | **0.357** | **0.495** | **0.368** | **0.223** | **0.417** |
| | ± | ± | ± | ± | ± | ± | ± | ± | ± |
| | 0.005 | 0.004 | 0.005 | 0.017 | 0.023 | 0.014 | 0.015 | 0.014 | 0.022 |

Table 16: Results of ablation study experiments on initialization. We report MRR, Hits@1, and Hits@3 (higher is better) on test sets.

| INIT | | WP-IND | | | JF-IND | | |
|---|---|---|---|---|---|---|---|
| $z_q$ | $p_i$ | MRR | Hits@1 | Hits@3 | MRR | Hits@1 | Hits@3 |
| - | - | 0.388 | 0.324 | 0.421 | 0.390 | 0.295 | 0.451 |
| ✓ | - | 0.387 | 0.321 | 0.421 | 0.392 | 0.302 | 0.447 |
| - | ✓ | 0.394 | 0.329 | 0.430 | 0.393 | 0.300 | 0.456 |
| ✓ | ✓ | **0.414** | **0.352** | **0.451** | **0.435** | **0.357** | **0.495** |

Table 17: Results of ablation study experiments on positional encoding. We report MRR, Hits@1, and Hits@3 (higher is better) on test sets.

| PE | WP-IND | | | JF-IND | | |
|---|---|---|---|---|---|---|
| | MRR | Hits@1 | Hits@3 | MRR | Hits@1 | Hits@3 |
| Constant | 0.393 | 0.328 | 0.426 | 0.356 | 0.247 | 0.428 |
| One-hot | 0.395 | 0.334 | 0.428 | 0.368 | 0.275 | 0.432 |
| Learnable | 0.396 | 0.335 | 0.425 | 0.416 | 0.335 | 0.480 |
| Sinusoidal | **0.414** | **0.352** | **0.451** | **0.435** | **0.357** | **0.495** |

Table 18: Full results of transductive link prediction experiments on FB-AUTO and WikiPeople.

| | FB-AUTO | | | | WikiPeople | | | |
|---|---|---|---|---|---|---|---|---|
| | MRR | Hits@1 | Hits@3 | Hits@10 | MRR | Hits@1 | Hits@3 | Hits@10 |
| r-SimplE | 0.106 | 0.082 | 0.115 | 0.147 | - | - | - | - |
| m-DistMult | 0.784 | 0.745 | 0.815 | 0.845 | - | - | - | - |
| m-CP | 0.752 | 0.704 | 0.785 | 0.837 | - | - | - | - |
| m-TransH | 0.728 | 0.727 | 0.728 | 0.728 | - | - | - | - |
| RAE | 0.703 | 0.614 | 0.764 | 0.854 | 0.253 | 0.118 | 0.343 | 0.463 |
| NaLP | 0.672 | 0.611 | 0.712 | 0.774 | 0.338 | 0.272 | 0.362 | 0.466 |
| tNaLP+ | 0.729 | 0.645 | 0.748 | 0.826 | 0.339 | 0.269 | 0.369 | 0.473 |
| HINGE | 0.678 | 0.630 | 0.706 | 0.765 | 0.333 | 0.259 | 0.361 | 0.477 |
| NeuInfer | 0.737 | 0.700 | 0.755 | 0.805 | 0.351 | 0.274 | 0.381 | 0.467 |
| HSimplE | 0.798 | 0.766 | 0.821 | 0.855 | - | - | - | - |
| BERT | 0.776 | 0.735 | 0.802 | 0.850 | - | - | - | - |
| HypE | 0.804 | 0.774 | 0.823 | 0.856 | 0.263 | 0.127 | 0.355 | 0.486 |
| GETD | 0.367 | 0.254 | 0.422 | 0.601 | - | - | - | - |
| RAM | 0.830 | 0.803 | 0.851 | 0.876 | 0.363 | 0.271 | 0.405 | 0.500 |
| S2S | - | - | - | - | 0.364 | 0.273 | 0.402 | 0.503 |
| BoxE | 0.844 | 0.814 | 0.863 | 0.898 | - | - | - | - |
| HyperMLN | 0.831 | 0.803 | 0.851 | 0.877 | - | - | - | - |
| HyConvE | 0.847 | 0.820 | 0.872 | 0.901 | 0.362 | 0.275 | 0.388 | 0.501 |
| ReAIE | 0.861 | 0.836 | 0.877 | 0.908 | - | - | - | - |
| RD-MPNN | 0.810 | 0.714 | 0.880 | 0.888 | - | - | - | - |
| HCNet | **0.871** ± 0.005 | **0.842** ± 0.007 | **0.892** ± 0.003 | **0.922** ± 0.004 | **0.421** ± 0.004 | **0.344** ± 0.004 | **0.457** ± 0.005 | **0.565** ± 0.007 |

Table 19: Definition of INIT in the ablation study of initialization. Here, $\boldsymbol{q} = (q, \tilde{\boldsymbol{u}}, t)$, and $d$ is the hidden dimension before passing to the first layer.

| INIT | | $\boldsymbol{h}_{v\mid\boldsymbol{q}}^{(0)}$ |
|:--:|:--:|:--:|
| $\boldsymbol{z}_q$ | $\boldsymbol{p}_i$ | |
| - | - | $\sum_{i \neq t} \mathbb{1}_{v=u_i} * \boldsymbol{1}^d$ |
| ✓ | - | $\sum_{i \neq t} \mathbb{1}_{v=u_i} * \boldsymbol{p}_i$ |
| - | ✓ | $\sum_{i \neq t} \mathbb{1}_{v=u_i} * \boldsymbol{z}_q$ |
| ✓ | ✓ | $\sum_{i \neq t} \mathbb{1}_{v=u_i} * (\boldsymbol{p}_i + \boldsymbol{z}_q)$ |

Table 20: Definition of $\boldsymbol{p}_i$ in the ablation study of positional encoding. Here, $\mathbb{I}_i^d$ is the one-hot vector of $d$ dimension where only the index $i$ has entry 1 and the rest 0. Note that $d$ is the hidden dimension before passing to the first layer. $\hat{\boldsymbol{p}}$ is a $d$-dimensional learnable vectors. $\boldsymbol{p}_{i,j}$ is the $j$-th index of position encoding $\boldsymbol{p}_i$, and $d$ is the dimension of the vector $\boldsymbol{p}_i$.

| PE | |
|:--:|:--:|
| Constant | $\boldsymbol{p}_i = \boldsymbol{1}^d$ |
| One-hot | $\boldsymbol{p}_i = \mathbb{I}_i^d$ |
| Learnable | $\boldsymbol{p}_i = \hat{\boldsymbol{p}}_i$ |
| Sinusoidal | $\boldsymbol{p}_{i,2j} = \sin\left(\frac{i}{10000^{2j/d}}\right); \boldsymbol{p}_{i,(2j+1)} = \cos\left(\frac{i}{10000^{2j/d}}\right)$ |

Table 21: Comparison of the execution time of 1 epoch for inductive and transductive link prediction task with relational hypergraph using a single A10 GPU. Note that we use batch size $= 1$ during the testing for all models, and 10k steps for MFB-IND during the training of HCNets.

| | WP-IND | | JF-IND | | MFB-IND | | FB-AUTO | | WikiPeople | |
|---|---|---|---|---|---|---|---|---|---|---|
| | Train | Test | Train | Test | Train | Test | Train | Test | Train | Test |
| RD-MPNN | 2sec | 3.5min | 2sec | 3min | 14min | 38min | 3sec | 35min - | - | |
| HCNet | 3.5min | 18sec | 8min | 10sec | 80min | 3.5min | 4.5min | 4min | 3hr | 2hr |

