# OpenReview forum: "Link Prediction with Relational Hypergraphs"
_ICLR.cc/2025/Conference — Submitted to ICLR 2025_

### Official Review · Reviewer_dikX · 2024-10-27

**Soundness:** 3
**Presentation:** 3
**Contribution:** 2
**Rating:** 6
**Confidence:** 3

**Summary:**

This work proposes HCNet, which extends state-of-the-art NBFNet and C-MPNN of conventional graph for link prediction to relational hypergraphs for hyper-relation predictions. It first follows the schema of conventional graph neural network works, defines the invariance under relational hypergraph and proves the expressivity of their proposal using proposed WL test for relational hypergraph. Then, it follows the schema of knowledge graph works, proves that the logical reasoning ability of HCNet is same as grade model logic on hypergraph. Besides, this work achieves better empirical results on relational hypergraph benchmarks.

**Strengths:**

- This works have concrete theoretical justifications on both graph representation and knowledege graph domains including WL test (for relational hypergraph) and power of logical reasoning.
- The empirical result looks pretty good in compare to related works on relational hypergraph baselines.

**Weaknesses:**

- Although this work made enough refinement to adpat NBFNet and C-MPNN design onto relational hypergraph, there isn't significiant novelty introduced in the refinement: they are simply directly replace the concepts from conventional graph to relational hypergraph.
- For logical reasoning, I believe it should focus more on explainability to people, but the concept of graded modal logic for hypergraph (the definition and examples) in this paper is too abstract for people to understand their power. I think more evidence of HGML should be provided (in appendix), e.g., is it able to represent all logics or answer all logical queries on relational hypergraph? If not, what's the limitation?
- Since this is a neural network work and link prediction work (where negative sampling is necessary), they will be a lot randomness in experiments, thus it is essential to provide confidence interval at least to your works (I understand that some baselines do not have confidence interval).

**Questions:**

- I am not familiar with the background of link prediction on relational hypergraph, thus I have questions to the necessity of this problem: It seems that the hyper relation prediction can be decomposed into prediction over a relational path or subgraph, and all three benchark you used seems to built from conventional knowledege graph, thus I am wondering is there a real-world dataset or task where only relational hypergraph methods can handle, and can not be decomposed into conventional relational graph methods?
- Since your method is transductive on relations, can we simply treat each relation as an independent node, and use the hyper relation position as edge attribute between a node and the relation node (e.g., $r(u_k)$ becomes $u_k \overset{k}{\leftrightarrow} r$), thus we can convert this task into a classifition task over multiple edges on conventional graph? Is there any baselines corresponding to this idea?
- Is the order essential in hyper relations? I think it is possible to permute some node positions while not hurts the hyper relation? If so, isn't your definition too strict?
- I think predicting the relation also falls into link prediction catalog, but your link prediciton only focuses on nodes, is it defined only for this work?
- In refinement, I think you want to define $\xi$ is more expressive than $\xi'$ (can distinguish more cases), thus isn't $\xi' \prec \xi$ better than $\xi \prec \xi'$ in representing which one is more expressive?
- How is your method runtime in real-world training and inference? NBFNet is already expensive since the graph convolution has to be redo if the query change, and in your case, hyperedge may introduce exponential increase on the amount of neighbors.

---

> ### Author Response · Authors · 2024-11-18
>
> > “W1. Although this work made enough refinement to adpat NBFNet and C-MPNN design onto relational hypergraph, there isn't significiant novelty introduced in the refinement: they are simply directly replace the concepts from conventional graph to relational hypergraph.”
>
> HC-MPNNs can be viewed as an adaptation of C-MPNNs to relational hypergraphs, but this adaptation is non-trivial and the presented theory requires very different techniques. One crucial ingredient behind HC-MPNNs is the positional encoding of the arity positions, which is not present in C-MPNNs. This choice is not merely conceptual and leads to technical differences. The precise relation between HC-MPNN and C-MPNN is discussed in **Appendix H**, and further supported by **Theorem H.4**. The fundamental take-away is the following: **HC-MPNNs remain more expressive than C-MPNNs even on knowledge graphs**. This holds because HC-MPNNs on knowledge graphs can emulate the power C-MPNNs have on inverse-augmented knowledge graphs. Our theory shows that there is no need for such augmentation if we use a positional encoding, as we do in the paper.
>
>
> > “W2. For logical reasoning, I believe it should focus more on explainability to people, but the concept of graded modal logic for hypergraph (the definition and examples) in this paper is too abstract for people to understand their power. I think more evidence of HGML should be provided (in appendix), e.g., is it able to represent all logics or answer all logical queries on relational hypergraph? If not, what's the limitation?”
>
> HGML is not able to represent all logical queries, which hints at the fundamental limitations of our studied models. For example, formulas in HGML can only express the local properties of nodes. That is, properties of the form “a node is connected (via hyper-edges) to other nodes satisfying other (local) properties”. This reflects the message-passing nature of our models. Following the suggestion of the reviewer, we will extend the discussion regarding HGML in the camera-ready version.
>
> > “W3. Since this is a neural network work and link prediction work (where negative sampling is necessary), they will be a lot randomness in experiments, thus it is essential to provide confidence interval at least to your works (I understand that some baselines do not have confidence interval).”
>
> While we do not directly provide the confidence interval, we indeed provide the standard deviation over five different runs for experiments presented in the main body of the paper, shown in **Table 13** and **Table 16** of the original submission.
>
> > “Q1. I am not familiar with the background of link prediction on relational hypergraph, thus I have questions to the necessity of this problem: It seems that the hyper relation prediction can be decomposed into prediction over a relational path or subgraph, and all three benchark you used seems to built from conventional knowledege graph, thus I am wondering is there a real-world dataset or task where only relational hypergraph methods can handle, and can not be decomposed into conventional relational graph methods?”
>
> All datasets were originally built from Wikidata and Freebase, which come in the format of knowledge graphs with additional qualifiers involving other entities. Thus, technically they are also forms of hypergraphs.  One could argue that any relational hypergraphs can be decomposed into predictions over relational paths or via reifications. In fact, earlier approaches such as [1] and [2] have considered these approaches and showcased that conversion from relational hypergraphs to knowledge graphs will distract the knowledge graph models from capturing the essential information for link prediction over high-arity facts. We are happy to elaborate on this topic.

---

> ### Author Response · Authors · 2024-11-18
>
> >“Q2. Since your method is transductive on relations, can we simply treat each relation as an independent node, and use the hyper relation position as edge attribute between a node and the relation node (e.g., $r(u_k)$ becomes $u_k \xleftrightarrow{k} r$), thus we can convert this task into a classifition task over multiple edges on conventional graph? Is there any baselines corresponding to this idea?”
>
> Unfortunately, the proposed conversion does not work because it will confuse two hyper-edges with the same relations and entities, but with the entities being in a different order. For instance, given a relational hypergraph $G = (V, E, R)$, assume we have a relation $r \in R$ and $x,y,z \in V$, and two hyper-edges $r(x,y,z)$ and $r(z,y,x) \in E$. According to the described transformation, we will obtain the following (undirected) edges: $x \xleftrightarrow{1} r, y \xleftrightarrow{2} r, z \xleftrightarrow{3} r, z \xleftrightarrow{1} r, y \xleftrightarrow{2} r, x \xleftrightarrow{3} r$. Now, note that another pair of hyper-edges, $r(x,y,x)$ and $r(z,y,z)$, will generate the same set of edges using the described transformation. This counter-example shows that the conversion described will lose a lot of information for individual hyper-edges, and thus cannot be used to convert the task of link prediction with relational hypergraph.
>
> >“Q3. Is the order essential in hyper relations? I think it is possible to permute some node positions while not hurts the hyper relation? If so, isn't your definition too strict?”
>
> Yes, the order is essential in hyper-relations. This is because, for each hyper-relation, each position stands for a different meaning. For instance, take a hyper-relational that describes Mr. Tahta’s math tutorship for Hawking: *Tutoring(Tahta, Maths, Hawking)*. Surely it is impossible to permute the node positions without changing the meaning of this fact. This is exactly the reason why we consider positional encodings.
>
> >“Q4. I think predicting the relation also falls into link prediction catalog, but your link prediciton only focuses on nodes, is it defined only for this work?”
>
> The task of predicting the missing relation between given entities is commonly referred to as *relation prediction* in the literature,  and while it is related to link prediction (on entities), it is out of the scope of this paper and thus we opt not to dive into this. The proposed HR-MPNNs and HC-MPNNs frameworks are only defined on link prediction on nodes settings.
>
> >“Q5. In refinement, I think you want to define $\xi$ is more expressive than $\xi’$ (can distinguish more cases), thus isn't $\xi’ \prec \xi$  better than $\xi \prec \xi’$ in representing which one is more expressive?”
>
> Here we followed the usual notation regarding partition refinements: for two partitions $P$ and $P’$, we have $P \prec P’$ if $P$ is a refinement of $P’$. Indeed, in our case $\xi \prec \xi’$ corresponds to the case when $\xi$  distinguishes more cases than $\xi’$, that is, when $\xi$ is “more expressive’’ than $\xi’$. If this is considered too confusing by the reviewer, we are open to update the notation accordingly in the final version.

---

> ### Author Response · Authors · 2024-11-18
>
> >“Q6. How is your method runtime in real-world training and inference? NBFNet is already expensive since the graph convolution has to be redo if the query change, and in your case, hyperedge may introduce exponential increase on the amount of neighbors.”
>
> We agree with the reviewer that HC-MPNNs in general suffer from scalability issues. We have provided the full table of runtime for real-world training and inference in **Table 19** of the original submission.  We have also provided an in-depth analysis of computational complexity in **Appendix J**. Indeed, any model will suffer from scalability issues if the number of arities becomes large. With that being said, our approach remains feasible for the benchmark datasets, but we think it is important for future work to scale up these models for larger datasets, much like it has been done for classical GNNs [3,4].
>
> **Potential strategies.** There are several strategies to alleviate the scalability issues. One potential strategy is to take $k$-hop subgraph samplings of source nodes and potential targets to limit the size of the input hypergraphs, thus allowing HC-MPNNs to run on larger data. Additionally, we have included custom implementation via Triton kernel in the submitted codebase to account for the message passing process on relational hypergraphs, which on average halved the training times and dramatically reduced the space usage of the algorithm (5 times reduction on average). The idea is to not materialize all the messages explicitly as in PyTorch geometric, but directly write the neighboring features into the corresponding memory addresses. Compared with materializing all hyperedge messages which takes $O(k|E|)$ where $k$ is the maximum arity, computing with Triton kernel only is $O(|V|)$ in memory. This will enable fast and scalable message passing on relational hypergraphs, both on HR-MPNNs and HC-MPNNs.
>
> [1] Rosso et al. Beyond Triplets: Hyper-Relational Knowledge Graph Embedding for Link Prediction. WWW 20
>
> [2] Wen et al., On the Representation and Embedding of Knowledge Bases Beyond Binary Relations. IJCAI 2016
>
> [3] Hamilton et al. Inductive representation learning on large graphs. NIPS 2017
>
> [4] Zhu et al. A*net: A scalable path-based reasoning approach for knowledge graphs. NeurIPS 2023

---

> ### Author Response · Authors · 2024-11-25
>
> We thank you for your time and for your initial review that allowed us to strengthen the paper. Since the author-reviewer discussion period is closing soon, we would highly appreciate your feedback on our response. We are keen to take advantage of the author-reviewer discussion period to clarify any additional concerns.

---

> ### Author Response · Authors · 2024-11-29
>
> We thank the reviewer for their initial review. In our rebuttal, we aimed to address all of the concerns of the reviewer. In particular, we have provided concrete queries that cannot be expressed in HGML in the appendix (as requested). Further to this, we have also shown that the link prediction problem on relational hypergraphs cannot be solved by the method proposed by the reviewer (further justifying our approach). In terms of the scalability, we included a detailed analysis and provided strategies on how to scale up the models.
>
> We would greatly appreciate the reviewers feedback on our rebuttal so that we can integrate this feedback to our paper.

---

### Official Review · Reviewer_mmja · 2024-11-04

**Soundness:** 3
**Presentation:** 3
**Contribution:** 3
**Rating:** 8
**Confidence:** 3

**Summary:**

The paper introduces HC-MPNNs, a novel GNN architecture for link prediction on relational hypergraphs, achieving state-of-the-art results in both inductive and transductive tasks. While highly expressive, the model faces scalability challenges due to its computational complexity.

**Strengths:**

- **Innovative Approach**: The paper introduces Hypergraph Conditional Message Passing Neural Networks (HC-MPNNs), which generalize existing GNN architectures and target the complex task of k-ary link prediction on relational hypergraphs, moving beyond traditional binary link prediction tasks.
- **Theoretical Rigor**: The authors provide a detailed analysis of HC-MPNNs' expressive power through Weisfeiler-Leman tests and logical expressiveness, which significantly adds to the theoretical foundation of GNNs on hypergraphs.
- **State-of-the-Art Performance**: Empirical results demonstrate that HC-MPNNs outperform existing models for both inductive and transductive link prediction tasks across multiple datasets, showcasing substantial improvements.

**Weaknesses:**

- **Computational Complexity**: The paper acknowledges that the proposed HC-MPNNs may be computationally intensive, particularly for large hypergraphs. An in-depth analysis of the computational costs compared to baseline models is needed to assess scalability better.
- **Limited Scope**: The model is focused solely on link prediction. Exploring its applicability to other tasks, such as node classification or hypergraph-based query answering, would provide more versatility and impact.
- **Assumptions on Data Structure**: The approach assumes that hypergraph structures are available and complete, which may not always be feasible in real-world scenarios. The limitations of applying this method to noisy or partially observed hypergraphs are not discussed.

**Questions:**

- **Expressiveness Limitations**: Could the authors provide more details on situations or graph structures where the expressive power of HC-MPNNs might fall short, even with the enhanced encoding?
- **Impact of Hypergraph Structure**: How does the structure or density of a hypergraph affect the model’s performance, especially in cases of sparse or highly connected hypergraphs?
- **Generalization to Noisy Data**: How robust is HC-MPNN to noisy or incomplete hypergraph data, and are there mechanisms in place to handle such cases?
- **Positional Encoding Variants**: The ablation studies show sinusoidal positional encoding as the best choice. Could the authors elaborate on why this is effective in hypergraph contexts compared to other embeddings?
- **Use of Meta-Paths**: Given that meta-paths are important structures in relational (non-hypergraph) graphs [1, 2, 3], could there be advantages to incorporating them in hypergraph contexts?
- **Scalability of the Model**: Is the model feasible to apply in very large-scale hypergraphs (e.g., millions of nodes)?

[1] Seongjun, Y. et al., Graph Transformer Networks: Learning meta-path graphs to improve GNNs, 2022

[2] Ferrini, F. et al., Meta-Path Learning for Multi-relational Graph Neural Networks, LOG 2023

[3]Fu, Xinyu, et al. "Magnn: Metapath aggregated graph neural network for heterogeneous graph embedding." Proceedings of the web conference 2020. 2020.

---

> ### Author Response · Authors · 2024-11-18
>
> >“W1. **Computational Complexity**: The paper acknowledges that the proposed HC-MPNNs may be computationally intensive, particularly for large hypergraphs. An in-depth analysis of the computational costs compared to baseline models is needed to assess scalability better.”
>
> We have provided an in-depth analysis of computational complexity in **Appendix J** of the original manuscript.  We also provided the execution time for baseline models and HCNets in **Table 19** of the original manuscript.
>
>
> >“W2. **Limited Scope**: The model is focused solely on link prediction. Exploring its applicability to other tasks, such as node classification or hypergraph-based query answering, would provide more versatility and impact.”
>
> We acknowledge that exploring applicability to other tasks would be beneficial, but it is out of the scope of the current submission.
>
> >“W3. **Assumptions on Data Structure**: The approach assumes that hypergraph structures are available and complete, which may not always be feasible in real-world scenarios. The limitations of applying this method to noisy or partially observed hypergraphs are not discussed.”
>
> We kindly disagree with the reviewer as the proposed approach does not assume that the hypergraphs are complete. In fact, the task of link prediction is exactly to discover these missing links in partially observed hypergraphs.
>
> >“Q1. **Expressiveness Limitations**: Could the authors provide more details on situations or graph structures where the expressive power of HC-MPNNs might fall short, even with the enhanced encoding?”
>
> As shown in our paper, the expressive power of HC-MPNNs is upper bounded by the hypergraph relational 1-WL test, and hence, whenever this test cannot distinguish two nodes, HC-MPNNs cannot distinguish them either. For instance, consider a relational hypergraph $G = (V,E,R)$ where $V = \\{x,u’,u,v\\}, E = \\{r(x,u’)\\}, R = \\{r\\}$. Then even HC-MPNNs cannot distinguish between $r(u,v)$ and $r(u’,v)$, where it is clear that $u$ and $u’$ are not isomorphic.
>
> >“Q2. **Impact of Hypergraph Structure**: How does the structure or density of a hypergraph affect the model’s performance, especially in cases of sparse or highly connected hypergraphs?”
>
> In the experimental section, we have shown that HCNets excel on all datasets, especially in the inductive experiments. To further analyze the impact on the structure and density, we present the average degree of (training) datasets in the following table.
>
>
>
> | Dataset        | WP-IND | JF-IND | MFB-IND | FB-AUTO | WikiPeople |
> |----------------|--------|--------|---------|---------|------------|
> | Average degree | 1.03   | 1.36   | 104.5   | 2.16    | 6.06       |
>
>
> Observe that even though MFB-IND is a very dense hypergraph, our HCNets can still manage to double the metrics compared to existing models. We also show that under sparse hypergraph settings, HCNets still achieve competitive performance. We will present a detailed discussion on the impacts of the structure and density of hypergraphs in the camera-ready version.
>
>
> >“Q3. **Generalization to Noisy Data**: How robust is HC-MPNN to noisy or incomplete hypergraph data, and are there mechanisms in place to handle such cases?”
>
>
> Yes, there are a few mechanisms to handle noisy and incomplete hypergraph data, similar to the ones for normal graphs. One potential mechanism is DropEdge [1], i.e., randomly dropping (hyper-)edges during training, which helps alleviate overfitting issues on a small portion of noisy hyper-edges. We believe it is an important future work to study the robustness of HC-MPNNs.
>
>
> >“Q4. **Positional Encoding Variants**: The ablation studies show sinusoidal positional encoding as the best choice. Could the authors elaborate on why this is effective in hypergraph contexts compared to other embeddings?”
>
> We observe that sinusoidal encoding performs much better in experiments and this can be attributed to its ability to encode relative positions: it ensures that the node $u$ gets a different position in $r(u,v)$ than in $s(u,v,w)$ since even though the node appears in the first arity position in both facts, there is a difference between being in the first arity position of a binary fact vs a ternary fact. This difference is captured by sinusoidal encodings and so is the relative distance between $u$ and $v$ in the respective facts.

---

> ### Author Response · Authors · 2024-11-18
>
> > “Q5. **Use of Meta-Paths**: Given that meta-paths are important structures in relational (non-hypergraph) graphs [1, 2, 3], could there be advantages to incorporating them in hypergraph contexts?”
>
> Thank you for raising this point! Indeed, we believe that incorporating additional structures such as meta-paths would be beneficial for the task of link prediction with relational hypergraphs, and strategies presented in the provided literature could be adapted into a hypergraph context. This is beyond the scope of the current paper, but it is an interesting direction for future studies.
>
> > “Q6. **Scalability of the Model**: Is the model feasible to apply in very large-scale hypergraphs (e.g., millions of nodes)?”
>
> Scalability is generally a concern for inductive link prediction since link prediction between a given pair of nodes relies heavily on the structural properties of these nodes (due to the lack of node features) which necessitates strong encoders that go beyond the power of $1$-WL. This is more dramatic for relational hypergraphs since the prediction now relies on the structural properties of $k$ nodes and any model will suffer from scalability issues if $k$ becomes large. With that being said, our approach remains feasible for the benchmark datasets, but we think it is important for future work to scale up these models for larger datasets, much like it has been done for classical GNNs [2, 3].
>
> **Potential strategies.** There are several strategies to alleviate the scalability issues. One potential strategy is to take $k$-hop subgraph samplings of source nodes and potential targets to limit the size of the input hypergraphs, thus allowing HC-MPNNs to run on larger data. Furthermore, to resolve this practically, we have included custom implementation via Triton kernel in the submitted codebase to account for the message-passing process on relational hypergraphs, which on average halved the training times and dramatically reduced the space usage of the algorithm (5 times reduction on average). The idea is to not materialize all the messages explicitly as in PyTorch geometric, but directly write the neighboring features into the corresponding memory addresses. Compared with materializing all hyperedge messages which takes $O(k|E|)$ where $k$ is the maximum arity, computing with Triton kernel only is $O(|V|)$ in memory. This will enable fast and scalable message passing on relational hypergraphs, both on HR-MPNNs and HC-MPNNs.
>
> [1] Yu et al. DropEdge: Towards Deep Graph Convolutional Networks on Node Classification. ICLR 2020
>
> [2] Hamilton et al. Inductive representation learning on large graphs. NIPS 2017
>
> [3] Zhu et al. A*net: A scalable path-based reasoning approach for knowledge graphs. NeurIPS 2023

---

> ### Author Response · Authors · 2024-11-25
>
> We thank you for your time and for your initial review that allowed us to strengthen the paper. Since the author-reviewer discussion period is closing soon, we would highly appreciate your feedback on our response. We are keen to take advantage of the author-reviewer discussion period to clarify any additional concerns.

---

> > ### Comment · Reviewer_mmja · 2024-11-25
> >
> > I thank the authors for addressing my concerns. I will raise my score.

---

> > > ### Author Response · Authors · 2024-11-25
> > >
> > > Thank you for going through our rebuttal and for raising your score. We appreciate all the feedback.

---

### Official Review · Reviewer_tZJo · 2024-11-04

**Soundness:** 3
**Presentation:** 3
**Contribution:** 3
**Rating:** 6
**Confidence:** 4

**Summary:**

The authors first give an understanding of the expressive power and limitations of HR-MPNNs and propose the HC-MPNN structure based on HR-MPNN. Compared with current HR-MPNN models, the HC-MPNN structured model HCNet carries out a novel query-specific node initialization method. With the Weisfeiler-Leman test, the author proves that this model can better model isomorphic nodes since it maintains a better expressiveness than the HR-MPNN structured models in the relational hypergraphs.

**Strengths:**

S1. Clear and well-organized presentation of the whole paper.

S2. Solid proof of the theoretical expressiveness of the proposed method.

S3. Available code.

S4. Transductive experiments have been added.

**Weaknesses:**

Unfortunately for the authors, I am the reviewer who held a negative opinion on this paper for NeurIPS 2024. Here, I summarize my previous negative feedback:

**Major Concerns:**

C1. The theoretical proof in this paper is unrelated to the inductive setting. Previously, I hoped the authors would compare more transductive settings. This time, **Table 2 has addressed this concern quite well**. Thus, I decided to raise my score.

C2. I previously argued that hyper-relational graphs and relational hypergraphs can be converted with small tricks, which is why many prior studies conducted experiments on datasets like WD50K, WIKIPEOPLE, JF17K, and FBAuto. Therefore, I also suggested that the authors consider experimenting on other hyper-relational graphs. However, the authors hold a different view, and the explanation provided in lines 145-153 is not sufficient to alleviate my concern. Thus, I will retain my suggestion for this submission and discuss it further with other reviewers.

**Minor Concerns:**

C3. Many of the methods presented in Table 1 were published in or before 2020, which may imply that the inductive setting is not currently a major research focus. The authors may need to emphasize the significance of the inductive setting more in the paper.

**Questions:**

I would like the authors to continue discussing the disagreement over C2, explaining why they insist that relational hypergraphs and hyper-relational graphs have irreconcilable differences. In my view, even if their structures are not identical, small modifications could allow format conversion. The benefits of doing so include:

	1.	Making the proposed method more generalized and applicable to a broader range of scenarios.
	2.	Aligning with many previous studies in the field.

---

> ### Author Response · Authors · 2024-11-18
>
> > “C1. The theoretical proof in this paper is unrelated to the inductive setting. Previously, I hoped the authors would compare more transductive settings. This time, **Table 2 has addressed this concern quite well.** Thus, I decided to raise my score.”
>
> We thank the reviewer for keeping an open mind, for acknowledging the improvement, and for raising their score.
>
>
> > “C2. I previously argued that hyper-relational graphs and relational hypergraphs can be converted with small tricks, which is why many prior studies conducted experiments on datasets like WD50K, WIKIPEOPLE, JF17K, and FBAuto. Therefore, I also suggested that the authors consider experimenting on other hyper-relational graphs. However, the authors hold a different view, and the explanation provided in lines 145-153 is not sufficient to alleviate my concern. Thus, I will retain my suggestion for this submission and discuss it further with other reviewers.”
>
> **“Small tricks’’ for conversion.** We would like to stress that we are in full agreement with the reviewer here. Indeed, with small tricks, hyper-relational knowledge graphs and relational hypergraphs can be transformed into each other. Specifically, any hyper-relational knowledge graph can be transformed into relational hypergraphs by hashing the main relation and the qualifiers (with some order), whereas any relational hypergraph can be transformed into a hyper-relational knowledge graph by creating a brand new qualifier for each relation at each position. We also agree that HCNet could be applied to a broader range of scenarios following these transformations, and align better with many previous studies.
>
> **Relational hypergraphs are more general.** With that being said, we want to highlight that relational hypergraphs are still a more general set-up compared to hyper-relational knowledge graphs, because the converted hyper-relational knowledge graph from relational hypergraphs, according to the small trick we described above, must satisfy the property that qualifiers can only appear together with their corresponding relation in the main triplet, i.e., transforming $r(u_1,u_2,u_3, u_4)$ to $(u_1, r, u_4, \\{ r_2: u_2, r_3: u_3 \\} )$ will enforce the newly introduced qualifiers $r_2$ and $r_3$ to appear together with each other and with the main relation $r$. As a result, the expressiveness results in the paper also upper bound the hyper-relational knowledge graphs. We will modify the claims appropriately in the main body of the submission to make this point clear.
>
>
> **Requested experiments.** As suggested by the reviewer, we have experimented on JF17K under such transformation, comparing selected baselines with models defined on hyper-relational knowledge graphs, despite the well-known problems with JF17K, e.g., redundant entries and main triplet leakages. In fact, the hyper-relational knowledge graph version of JF17K is exactly transformed from the relational hypergraph version by the described trick above.
>
> It is crucial to highlight the differences in evaluation: in the evaluation of hyper-relational knowledge graphs, only the head and tail entities in the main triplet are corrupted. In our setup, we follow the evaluation convention of relational hypergraphs and corrupt all positions. We report the results here and will include the full table of baselines in the final version.
>
> | JF17K  | MRR    | Hits@1 | Hits@10 | Hits@10 |
> |--------|--------|--------|---------|---------|
> | RAE    | 0.310  | 0.219  | 0.504   | 0.854   |
> | HINGE  | 0.517  | 0.436  | 0.675   | 0.850   |
> | StarE  | 0.542  | 0.454  | 0.685   | 0.856   |
> | HCNet  | 0.540  | 0.440  | 0.730   | 0.922   |
>
>
> The results of HCNet are better than existing baselines, including models designed for relational hypergraphs and models designed for hyper-relational knowledge graphs. In particular, HCNet marginally outperforms StarE according to Hits@10. StarE is one of the state-of-the-art models on link prediction with hyper-relational knowledge graphs, but StarE is a transductive method and is inherently limited to transductive datasets, whereas HCNets do not have such limitations. In this sense, HCNets also lift the capabilities of methods designed for hyper-relational knowledge graphs to the inductive setup, which is substantially more challenging. We thank the reviewer for helping us to enlighten this aspect, and we are happy to include more experiments as time allows.

---

> ### Author Response · Authors · 2024-11-18
>
> > “C3. Many of the methods presented in Table 1 were published in or before 2020, which may imply that the inductive setting is not currently a major research focus. The authors may need to emphasize the significance of the inductive setting more in the paper.”
>
> **Recent baseline.** We indeed considered a rather recent baseline RD-MPNNs in **Table 1**, which was proposed in 2023. To the best of our knowledge, the only existing works on inductive link prediction over relational hypergraphs are G-MPNNs and RD-MPNNs (with slight modification on initialization), and in particular, RD-MPNNs is a model under the HR-MPNN framework and we showed that HCNet also outperforms RD-MPNNs in inductive link prediction.
>
> **Importance of inductive setting.** The importance of the inductive setting is already widely acknowledged in the literature on knowledge graphs [2-4], and the same arguments apply to relational hypergraphs with high-arity relations.  In fact, we believe that this leaves an important gap in the literature. With that being said, we agree with the reviewer that the significance of the inductive setting is currently not made explicit enough in the paper, and we will further highlight this aspect in the camera-ready version.
>
>
>
>
> > “Q1. I would like the authors to continue discussing the disagreement over C2, explaining why they insist that relational hypergraphs and hyper-relational graphs have irreconcilable differences. In my view, even if their structures are not identical, small modifications could allow format conversion. The benefits of doing so include: 1. Making the proposed method more generalized and applicable to a broader range of scenarios. 2. Aligning with many previous studies in the field.”
>
> Please refer to the response of C2 and we are more than happy to discuss further, as we strongly believe that this process has made our paper stronger. We also deeply appreciate the feedback.
>
>
> [1] Barcelo, et al. The logical expressiveness of graph neural networks, ICLR 2020
>
> [2] Teru, K. K., Denis, E. G., and Hamilton, W. L. Inductive relation prediction by subgraph reasoning. In ICML, 2020
>
> [3] Zhu, Z., Zhang, Z., Xhonneux, L.-P., and Tang, J. Neural bellman-ford networks: A general graph neural network framework for link prediction. In NeurIPS, 2021
>
> [4] Zhu et al. A*net: A scalable path-based reasoning approach for knowledge graphs. NeurIPS 2023

---

> ### Author Response · Authors · 2024-11-25
>
> We thank you for your time and for your initial review that allowed us to strengthen the paper. Since the author-reviewer discussion period is closing soon, we would highly appreciate your feedback on our response. We are keen to take advantage of the author-reviewer discussion period to clarify any additional concerns.

---

> ### Comment · Reviewer_tZJo · 2024-11-25
> **ACK**
>
> The authors addressed one of my major concerns. I read the rebuttal carefully and appreciate the authors' clarification. I would like to keep the positive review unchanged.

---

> > ### Author Response · Authors · 2024-11-25
> >
> > Thank you for acknowledging the rebuttal and the clarifications. We found the feedback very helpful and also appreciate your continued positive evaluation.

---

### Official Review · Reviewer_t9zR · 2024-11-09

**Soundness:** 2
**Presentation:** 3
**Contribution:** 2
**Rating:** 6
**Confidence:** 4

**Summary:**

This paper proposes a new method for link prediction in relational hypergraphs, where the task focuses on k-ary relations. It first investigates the expressive power of existing GNNs for relational hypergraphs, and then introduces the hypergraph conditional message passing neural network (HC-MPNN) by utilizing and extending the techniques from conditional message passing neural networks (C-MPNNs) (Huang et al., 2023). The HC-MPNN is further extended to an inductive setting, and show good experimental results.

**Strengths:**

1. The paper is written in a good structure. Although the definition of the relational hypergraph is a little bit complex; the descriptions in the paper is not difficult to understand. The organization of the paper is good. From the theoretic analysis to model improvement, the techniques are quite clear.

2. Extending the C-MPNN and HR-MPNN to HC-MPNN is technically sound. And the extension to inductive link prediction is interesting.

3. Both theoretic analysis and experiments are comprehensive. It has with both inductive and transductive settings, and the analysis of the initialization and positional encoding is also useful.

**Weaknesses:**

1. First, I have a concern about the practicality of the method. From my understanding, the relational hyperedge is just like a sentence, and it has fixed directions and positions for entities. So, why do not people just use Transformers for NLP by regarding the relational hyperedge as a sentence? And the query based approach is just like a BERT-based model? I do not see much benefit from modeling it as a hypergraph. I am actually curious to see the results of BERT on the same datasets if possible.

2. Second, although the paper is technically sound and the analysis is complete/rigorious; but every step in the model is not surprising and a little bit trivial. Without the analysis part, the HC-MPNN is like just a migration of C-MPNN to hypergraphs. The inductive link prediction part is more interesting.

3. I do not fully understand the training. Since the message passing is query dependent, there must be a lot of queries in the training data. How do you select such queries for training? And for each query, the node representation is different, does not it increase the computational complexity a lot?

**Questions:**

1. Does a relational hyperedge have a fixed number of entities? For example, can we have r(e1, e2) and r(e1, e2, e3) at the same time?
2. How to train the model with queries?

---

> ### Author Response · Authors · 2024-11-18
>
> > “W1. First, I have a concern about the practicality of the method. From my understanding, the relational hyperedge is just like a sentence, and it has fixed directions and positions for entities. So, why do not people just use Transformers for NLP by regarding the relational hyperedge as a sentence? And the query based approach is just like a BERT-based model? I do not see much benefit from modeling it as a hypergraph. I am actually curious to see the results of BERT on the same datasets if possible.”
>
> **Limitation of modeling as sentences.** Indeed, in principle, one can model each relational hyperedge as a sentence of entities plus a relation, and at inference time, the user can input the query and rely on a transformer-based model to decode the probability that the queried hyperedge exists. However, these approaches are inherently **transductive**, meaning that the model cannot make predictions on unseen entities and is thus limited, much like traditional relational hypergraph embedding methods. In other words, transformer-based models like BERT still rely on explicitly storing learned entity embeddings, making them essentially a more complex form of transductive embedding methods, which do not apply in the inductive setting.
>
> **The benefit of modeling as a hypergraph.** By explicitly modeling these relational hyperedges as a relational hypergraph and learning over the structure instead of standalone sentences, both HR-MPNNs and HC-MPNNs, can leverage the power of message-passing and generalize over unseen entities and even unseen relational hypergraphs. This is the key idea behind the inductivity of these models.
>
> **Additional transductive experiments.** Following the reviewer’s advice, we conducted additional experiments with BERT on one of the transductive datasets, FB-AUTO. Observe that BERT performs better than earlier transductive embedding models such as RAE but falls behind recent transductive embedding models like HypE. It is also significantly behind the performance of HCNet. We will provide the full experiment results on all datasets in the final version of the paper.
>
>
> | FB-AUTO   | MRR       | Hits@1    | Hits@3    | Hits@10   |
> |-----------|-----------|-----------|-----------|-----------|
> | RAE       | 0.703     | 0.614     | 0.764     | 0.854     |
> | BERT      | 0.776     | 0.735     | 0.802     | 0.850     |
> | HypE      | 0.804     | 0.774     | 0.823     | 0.856     |
> | **HCNet** | **0.871** | **0.842** | **0.892** | **0.922** |
>
>
>
> > “W2. Second, although the paper is technically sound and the analysis is complete/rigorious; but every step in the model is not surprising and a little bit trivial. Without the analysis part, the HC-MPNN is like just a migration of C-MPNN to hypergraphs. The inductive link prediction part is more interesting.”
>
>
> We thank the reviewer for highlighting the inductive link prediction part as well as the soundness and completeness of our analysis. We want to stress that the contribution of this paper is not to develop a specific model on relational hypergraphs but rather to propose a general framework (HC-MPNNs) and conduct an analysis to study its expressive power rigorously. The generality of the proposed framework allows us to study the theoretical properties of a broader range of model architectures following the same setup.
>
> While HC-MPNNs can be viewed as a natural adaptation of C-MPNNs to relational hypergraphs, this adaptation is non-trivial and the presented theory requires substantially different techniques. One crucial ingredient in HC-MPNNs is the positional encoding of the arity positions, which is not present in C-MPNNs. Please note that this choice is not merely conceptual and leads to technical differences. The precise relation between HC-MPNN and C-MPNN is discussed in **Appendix H**, and further supported by **Theorem H.4**. The fundamental take-away is the following: **HC-MPNNs remain more expressive than C-MPNNs even on knowledge graphs.** This holds because HC-MPNNs on knowledge graphs can emulate the power C-MPNNs have on inverse-augmented knowledge graphs. Our theory shows that there is no need for such augmentation if we use a positional encoding, as we do in the paper.

---

> ### Author Response · Authors · 2024-11-18
>
> > “W3. I do not fully understand the training. Since the message passing is query dependent, there must be a lot of queries in the training data. How do you select such queries for training? And for each query, the node representation is different, does not it increase the computational complexity a lot?”
>
> We follow the same convention and select all possible hyperedges in the training relational hypergraph as queries.  More precisely, given a training relational hypergraph $G = (V,E,R,c)$, we pick each hyperedge $r(e_1,..., e_t,… e_k) \in E$ from the given relational hypergraph, and for each position $1 \leq t \leq k$, we treat this chosen hyperedge as positive with label 1, and randomly generate $n$ negative hyperedges $r(e_1,..., e’,… e_k)$ with label 0 where $e’$ is a corrupted entity at position $t$, and $n$ is a hyper-parameter indicating the number of negative samples.
>
> Indeed, HC-MPNNs have higher amortized computational complexity per query than HR-MPNNs, which is thoroughly discussed in **Appendix J** of the original manuscript.  Nevertheless, we would like to highlight that HC-MPNNs and HR-MPNNs have the same computational complexity for a single forward pass.
>
>
> > “Q1. Does a relational hyperedge has a fixed number of entities? For example, can we have r(e1, e2) and r(e1, e2, e3) at the same time?”
>
> If the reviewer is asking whether for a fixed relation $r \in R$, can hyperedges of the form $r(e1,e2)$ and $r(e1,e2,e3)$ both exist, then the answer is no: for each fixed relation $r$, it can only take exactly $\text{arity}(r)$ of entities, which is predefined. Thus, if $\text{arity}(r)$ = 3, then only the hyperedge $r(e1,e2,e3)$ can exist but $r(e1,e2)$ can not.
>
> If the reviewer is asking whether all hyperedges in relational hypergraphs have a fixed number of entities, then the answer is again no: each hyperedge can have a different number of entities, depending on the arity of the relation $r$, determined by the function $\text{arity}$.
>
>
>
> > “Q2. How to train the model with queries?”
>
> Please refer to the response of W3 above.

---

> ### Author Response · Authors · 2024-11-25
>
> We thank you for your time and for your initial review that allowed us to strengthen the paper. Since the author-reviewer discussion period is closing soon, we would highly appreciate your feedback on our response. We are keen to take advantage of the author-reviewer discussion period to clarify any additional concerns.

---

> > ### Comment · Reviewer_t9zR · 2024-11-28
> >
> > Thanks for the response. I am still not convinced of the practicality of the method (e.g. the inductive learning problem can also be solved by adapting the path sentence), but considering the clarification and additional good results, I am willing to raise the score to 6.

---

> > > ### Author Response · Authors · 2024-11-28
> > >
> > > Thank you for acknowledging the additional experiments in our rebuttal, and for raising your score!
> > >
> > > We notice that we were not very explicit in our initial response regarding inductive learning and the limitations of BERT in the inductive setup. Let us elaborate on this further and more concretely:
> > >
> > > 1. **Encoding hyper-edges using BERT**: Since each hyper-edge is of the form $r(u_1,…,u_k)$, they can also be interpreted as paths or sentences, as the reviewer suggests. In other words, BERT learns to encode hyper-edges over a set $V$ of nodes which are all the nodes that are *observed* during the training.
> > > 2. **Test Time**: We are given a fact of the form $r(v_1,…,v_k)$, where $v_1,…,v_k$ are unseen during training. First, observe that we only have access to the **node IDs** in this setup and there is *no meaningful textual information* available regarding the nodes. Given this, the only way to make an inductive prediction is by relying on structural similarities. Specifically, if the model learns from the structural properties of the existing hyper-edges (i.e., their local structure in the graph) then the model can transfer this knowledge to make a prediction on the new hyper-edge $r(v_1,…,v_k)$. Since BERT ignores the structural role of the hyper-edge in the graph, this model cannot make inductive predictions in this setup.
> > >
> > > That being said, BERT-based model can make transductive predictions but we observe that the performance of the BERT-based model is substantially below of HCNets even in this restricted setup. We are happy to elaborate further on this and hope that this addresses the concern of the reviewer.

---

### Author Response · Authors · 2024-11-18
**Global response**

We thank the reviewers for their comments. We respond to each concern in detail in our individual responses, and we provide a summary of our rebuttal based on the feedback from the reviewers:
- **Additional Baseline.**  We add an additional baseline model BERT as an instance of transductive embedding methods (**Reviewer t9zR**).
- **Additional Dataset.** We experimented with an additional transductive dataset  (JF17K) and compared our model with the baseline model including models operating on hyper-relational knowledge graphs. (**Reviewer tZJo**).
- **Extended discussions on Design Choices.** We highlight the delicate design choices behind HC-MPNNs which are crucial in applying conditional message passing on relational hypergraphs. (**Reviewer t9zR, Reviewer dikX**)
- **Implications of Expressiveness Results.** In addition to quantifying the precise expressive power of HC-MPNNs, we also elaborate on the induced expressiveness limitations (**Reviewer mmja, Reviewer dikX**).
- **Scalability.**  We discuss the scalability of HCNets and provide a strategy to apply HCNets on much larger graphs. (**Reviewer mmja, Reviewer dikX**)

We hope that our answers address your concerns, and we are looking forward to a fruitful discussion period.

---

### Author Response · Authors · 2024-11-24

We thank the reviewers for taking their time to review our paper. We have now integrated all the feedback and additional findings from the rebuttal to the paper (highlighted in blue). We would deeply appreciate any additional feedback and looking forward to discuss further.

---

### Meta-Review · Area_Chair_s8qC · 2024-12-18

**Metareview:**

This submission violated the anonymity policy. In the provided link (https://anonymous.4open.science/r/HCNet/README.md), when clicking the blue-colored text, Link Prediction with Relational Hypergraphs, the arXiv version of the paper is shown, which reveals all authors' information: https://arxiv.org/abs/2402.04062

This submission should have been desk-rejected. It was difficult to expect a redirection to the arXiv paper, so this clear violation was not found during the desk-rejecting period.

In addition to some of the reviewers' remaining concerns, I also found some critical issues based on my own reading:

1. **Misunderstandings of the relationship between relational hypergraphs and n-ary relational graphs.**\
The authors argue that "link prediction with relational hypergraphs is a more general problem setup" than that with n-ary relational graphs. However, I'm afraid I have to disagree with it. While a relational hyperedge can be transformed into an n-ary relational fact without loss of information, the reverse transformation is not always lossless. Given a hyperedge $r(u_1,u_2,u_3, u_4)$, it can be transformed into a 4-ary relational fact $\\{r_1:u_1, r_2:u_2, r_3:u_3, r_4:u_4\\}$ without loss of information. However, when we transform $\\{r_1:u_1, r_2:u_2\\}$ and $\\{r_1:u_3, r_2:u_4, r_3:u_5\\}$ into hyperedges, they are transformed into $r’_1(u_1, u_2)$ and $r’_2(u_3,u_4,u_5)$. In this conversion, the information loss occurs: the facts that $u_1$ and $u_3$ are both involved in $r_1$ and that $u_2$ and $u_4$ are both involved in $r_2$ are lost. Therefore, n-ary relational graphs are a more general form than relational hypergraphs.

2. **Mismatch of the concept of the relational hypergraph and benchmark datasets**\
The authors claim to "opt out datasets of hyper-relational knowledge graphs" and to "focus only on the datasets designed for relational hypergraphs". However, the datasets the authors used are extracted from knowledge bases such as Wikidata or Freebase, none of which are originally relational hypergraphs but in the form of n-ary relational graphs or hyper-relational knowledge graphs. While these datasets may have been adopted for evaluating methods targeting relational hypergraphs, their original structures do not perfectly align with the definition of relational hypergraphs. This mismatch renders these datasets unsuitable for evaluating methods designed specifically for relational hypergraphs. The authors should instead conduct experiments on datasets extracted from relational hypergraphs.

3. **Misunderstanding of the literature**\
The authors stated, "We also highlight the evaluation differences as experiments on hyper-relational knowledge graphs only corrupt entities in the main triplets, whereas, in link prediction with relational hypergraphs setting, all entities mentioned at all positions are corrupted". However, this claim is inaccurate. For example, GRAN [1], a model for hyper-relational knowledge graphs, includes an evaluation setting where all entities mentioned at all positions can be corrupted.

4. **Missing literature**\
The manuscript does not include recent baseline methods, such as HJE [2] (Published on February 13, 2024). HJE demonstrates superior performance compared to the proposed method. For instance, regarding MRR on the WikiPeople dataset, HCNet achieves 0.421, whereas HJE achieves 0.450.

5. **Incorrectly reporting baseline's performance**\
This manuscript incorrectly reports ReAlE's performance [3]. For example, on the FB-AUTO dataset, the original ReAlE paper reports an MRR of 0.873, whereas this manuscript reports it as 0.861.

6. **Misunderstanding on JF17K**\
In the rebuttal, the authors included experiments on the JF17K dataset to compare with baseline methods designed for hyper-relational knowledge graphs. However, the JF17K dataset is an n-ary relational graph dataset, not a hyper-relational knowledge graph dataset in its original form. Thus, the authors should instead conduct experiments on datasets such as WD50K to facilitate a valid comparison with baseline methods targeting hyper-relational knowledge graphs.

[1] Wang et al., 2021. Link Prediction on N-ary Relational Facts: A Graph-based Approach. ACL Findings.\
[2] Li et al., 2024. HJE: Joint Convolutional Representation Learning for Knowledge Hypergraph Completion. TKDE.\
[3] Fatemi et al., 2023. Knowledge Hypergraph Embedding Meets Relational Algebra. JMLR.

**Additional Comments On Reviewer Discussion:**

This submission should have been desk-rejected since an embedded hyperlink in the provided link reveals the authors' information. Also, the manuscript has significant issues that should be resolved before publishing, as detailed in the metareview.

---

### Decision · Program_Chairs · 2025-01-22

Reject